# Primary cilia and SHH signaling impairments in human and mouse models of Parkinson's disease

Sebastian Schmidt [1,2,16], Malte D. Luecken [3,16], Dietrich Trümbach [1,4,16], Sina Hembach[1,2], Kristina M. Niedermeier[1,2], Nicole Wenck[1,2], Klaus Pflügler[1,2], Constantin Stautner[1,2], Anika Böttcher [5], Heiko Lickert [5], Ciro Ramirez-Suastegui [3], Ruhel Ahmad[6], Michael J. Ziller [7], Julia C. Fitzgerald[8], Viktoria Ruf[9,10], Wilma D. J. van de Berg[11], Allert J. Jonker[11], Thomas Gasser [8], Beate Winner[12], Jürgen Winkler [13], Daniela M. Vogt Weisenhorn [1,2], Florian Giesert [1] ✉, Fabian J. Theis [3,14] ✉ & Wolfgang Wurst [1,2,10,15] ✉

Parkinson's disease (PD) as a progressive neurodegenerative disorder arises from multiple genetic and environmental factors. However, underlying pathological mechanisms remain poorly understood. Using multiplexed single-cell transcriptomics, we analyze human neural precursor cells (hNPCs) from sporadic PD (sPD) patients. Alterations in gene expression appear in pathways related to primary cilia (PC). Accordingly, in these hiPSC-derived hNPCs and neurons, we observe a shortening of PC. Additionally, we detect a shortening of PC in *PINK1*-deficient human cellular and mouse models of familial PD. Furthermore, in sPD models, the shortening of PC is accompanied by increased Sonic Hedgehog (SHH) signal transduction. Inhibition of this pathway rescues the alterations in PC morphology and mitochondrial dysfunction. Thus, increased SHH activity due to ciliary dysfunction may be required for the development of pathoetiological phenotypes observed in sPD like mitochondrial dysfunction. Inhibiting overactive SHH signaling may be a potential neuroprotective therapy for sPD.

Parkinson's disease (PD) is a neurodegenerative disorder characterized in advanced stages by motor disabilities which are mainly due to the loss of dopaminergic neurons (DAn) in the substantia nigra pars compacta[1]. However, PD is known as a gradual-onset condition with a long-defined preclinical and clinical stage of PD before the manifestation of the typical motor symptoms[2]. The presence of a multitude of symptoms defines the concept of PD being a multisystemic rather than a disease solely affecting DAns. In addition, understanding the early phases in PD progression on the molecular level before the manifestation of the major motor symptoms raises the hope to establish effective neuroprotective therapies[3].

About 15% of PD are associated with heritable familial mutations whereas the vast majority is sporadic with the contribution of genetic and environmental risk factors[4]. Analysis of model systems carrying inherited PD-associated mutations have been harnessed to unravel the molecular and cellular mechanisms such as mitochondrial impairment, autophagy, protein aggregation, proteasomal degradation, and lysosomal dysfunction contributing to disease etiology[5]. Many of them

A full list of affiliations appears at the end of the paper. ✉e-mail: florian.giesert@helmholtz-muenchen.de; fabian.theis@helmholtz-muenchen.de; wurst@helmholtz-muenchen.de

have been established in mouse models of PD without dopaminergic deficits. Still, the relevance of these mechanisms for translational studies was appreciated due to the lack of suitable human models. With the advent of human induced pluripotent stem cells (hiPSCs) an advancement of our knowledge on human physiology is now possible[6]. These cells and their derivatives offer the possibility to elucidate in a systematic fashion the multilayered molecular alterations and cellular signaling networks that underlie the early pathoetiology of PD[7–9].

Thus, to address molecular and cellular mechanisms impaired in early sporadic Parkinson's disease (sPD) we used hiPSC-derived neuronal precursor cells (hNPCs) and dopaminergic neurons (DAns) from sPD patients and control individuals (Ctrls). Neural cells derived from sPD patients exhibited reduced mitochondrial respiration as well as a clear complex I deficiency comparable to postmortem brain tissue. An unbiased molecular characterization of these hNPCs using multiplexed, droplet-based, single-cell RNA sequencing (scRNA-seq) allowed us to identify a multitude of genes associated with primary cilia (PC) function that were dysregulated in our cellular model but also in postmortem tissue of sPD patients. Cellular analysis revealed impairments in PC morphology and PC signaling in sPD - specifically SHH-dependent signal transduction was altered. Notably, inhibition of SHH signaling rescued the observed deficits in PC morphology and mitochondrial respiration. Respective alterations in PC morphology could also be observed in *PINK1*-deficient hNPCs and striatal neurons of a prodromal *Pink1* knock-out (ko) mouse model. These findings suggest that the dysfunction of PC signaling pathways and especially SHH signaling is a molecular pathway underlying PD development.

## Results

### hNPCs and DAns develop a PD-specific mitochondrial dysfunction

In the present study, we used hiPSCs of 7 late-onset sPD patients and 5 Ctrl individuals (with two hiPSC clones per individual) and their neuronal derivatives (hNPCs and DAns) as a cellular model system for sporadic PD. PD patients were extensively clinically examined and defined as having sPD by the absence of known PD-causing familial mutations (PARK 1-18)[10] (Supplementary Data 1). Otherwise, PD patients were not pre-selected in any way. Age- and sex-matched healthy individuals without any neurological disease were chosen as corresponding Ctrls. High passage hiPSCs (>45 passages) and thereof differentiated hNPCs and DAns from these individuals were characterized and showed correct expression of pluripotency markers (NANOG, SOX2, OCT4), differentiation markers (PAX6, NES) and neuronal markers (TUBB3, TH), respectively (Fig. 1a). Since the hNPCs and the thereof differentiated DAns were explicitly derived from high passage hiPSCs an analysis of the copy number variations (CNVs) was performed. The number and size of CNVs were slightly increased compared to those published for low passage cells[10] but did not show significant differences between sPD and Ctrl (Fig. 1b, c). Nor were genes or pathways affected by these CNVs that were relevant for this study (Supplementary Data 2 and 3). Thus, high passage numbers did not have a differential effect on the genomic integrity of cells derived from patients and Ctrls.

To elucidate if DAn differentiation of PD patient derived hiPSCs is affected e.g. already causing their degeneration, numbers of DAns and their neurite morphology was analyzed. Neuronal populations derived from sPD and Ctrl clones contained similar amounts of neurons (positive for RBFOX3/NeuN) and DAns (positive for RBFOX3/NeuN and TH) (Fig. 1d). Also, the neurite morphology regarding the number, length, and branching points of neurites was similar in DAns derived from patients and Ctrls (Fig. 1e).

To validate these cells as a model system for sPD, we first assessed pathological hallmarks that are assumed to contribute to DAn degeneration in sPD patients[11,12] such as alterations in cellular respiration and complex I activity. Mitochondrial respiration was not affected

in the original fibroblasts (Fig. 2a). Also, in hiPSCs (Fig. 2b) reprogrammed thereof no alteration in mitochondrial respiration could be detected, but was significantly reduced in differentiated hNPCs (Fig. 2c) and DAns (Fig. 2d) of sPD patients. Additionally, sPD hNPCs exhibited a clear complex I deficiency (Supplementary Fig. 8a) comparable to postmortem brain tissue[11]. The mitochondrial dysfunction observed in sPD cells was also highly cell-type-specific as it couldn't be observed in hiPSC-derived astrocytes (Fig. 2e).

In a similar powered study analyzing a large cohort of patient derived fibroblasts[13], however, slight mitochondrial dysfunction was observed. This finding could not be replicated in a larger powered study by[14]. Thus, in addition to the observed cell-type specificity of a mitochondrial phenotype a stratification of patients according to deficits in mitochondria also in peripheral tissue seems to be warranted.

These neuronal cells represent a veritable model for sPD as they recapitulate known PD-associated molecular alterations before the final and irreversible degeneration of DAns. As already hNPCs recapitulated these alterations, these hNPCs are a perfect model suitable for high-throughput screening processes.

### Definition of distinct cell populations within the hNPC culture

To further unravel the molecular underpinnings of the cellular phenotype underlying the etiology of multifactorial sPD, we performed a pooled, droplet-based scRNA-seq of hNPCs using a 10xGenomics protocol (Fig. 3a). We demultiplexed pooled libraries[15] from different individuals based on single nucleotide polymorphisms (SNPs), thereby avoiding confounding technical batch effects (see methods). As two clones from the same individual wouldn't be distinguishable using this method, one out of two hNPC clones per individual (5 Ctrl and 7 sPD patients) was selected (Supplementary Data 15) based on features characteristic of hiPSCs such as karyotype and pluripotency. The same hNPC clones were used for all further analysis. After quality control (Supplementary Figs. 1a–c and 12e), we retained a dataset of 30,557 individual cells, which were profiled with about 90.000 reads/cell.

To classify distinct cell subpopulations within the hNPC cultures, all cells were clustered jointly across the 12 individuals thereby producing transcriptionally distinct cell populations with highly consistent expression patterns across individuals (Fig. 3b). We identified and annotated these cellular populations by the expression patterns of specific marker genes (Supplementary Fig. 2; Supplementary Data 4). In total, nine distinct clusters were identified: neural crest stem cells (NCSC – marked by the expression of *LUM*, *LGALS1*, and a high fraction of ribosome[16] and extracellular matrix[17] constituents), glial precursors (marked by the expression of *SOX10* and *S100B*[18]), apoptotic cells (marked by their low mitochondrial fraction, active cell death signaling and nonspecific marker genes), immature neurons (marked by the expression of *DCX* and *MAP2*[19]), and neural stem cells (NSCs) proper all of which expressed the marker gene *SOX2*[20]. This NSC cluster could be subdivided into 4 sub-clusters based on differential expression of respective marker genes. The cluster NSC1a was marked by the expression of *DAAM1*, *FABP7*, *PTCH1*, and *DLL1*, whereas the cluster NSC2a was characterized by the expression of *SHH*, *FOXA2*, and *HES1*. The remaining two clusters NSC1b and NSC2b were characterized by the expression of similar genes as NSC1a and NSC2a, respectively, however, expressed also genes associated with the G2M phase of the cell cycle such as *CENPF*, *AURKB*, and *CDC20*[21]. Pathway analysis superimposed on the respective marker genes revealed that the NSC1 cluster was predominantly marked by the NOTCH and WNT signaling pathway, whereas the NSC2 cluster was predominantly marked by SHH expression and the FOXA2 transcription factor network. Temporal and spatial activation of these pathways plays a role in the proliferation and differentiation of NSCs in vivo, thus subdivision of the NSC population into these 4 clusters is biologically relevant[22,23].

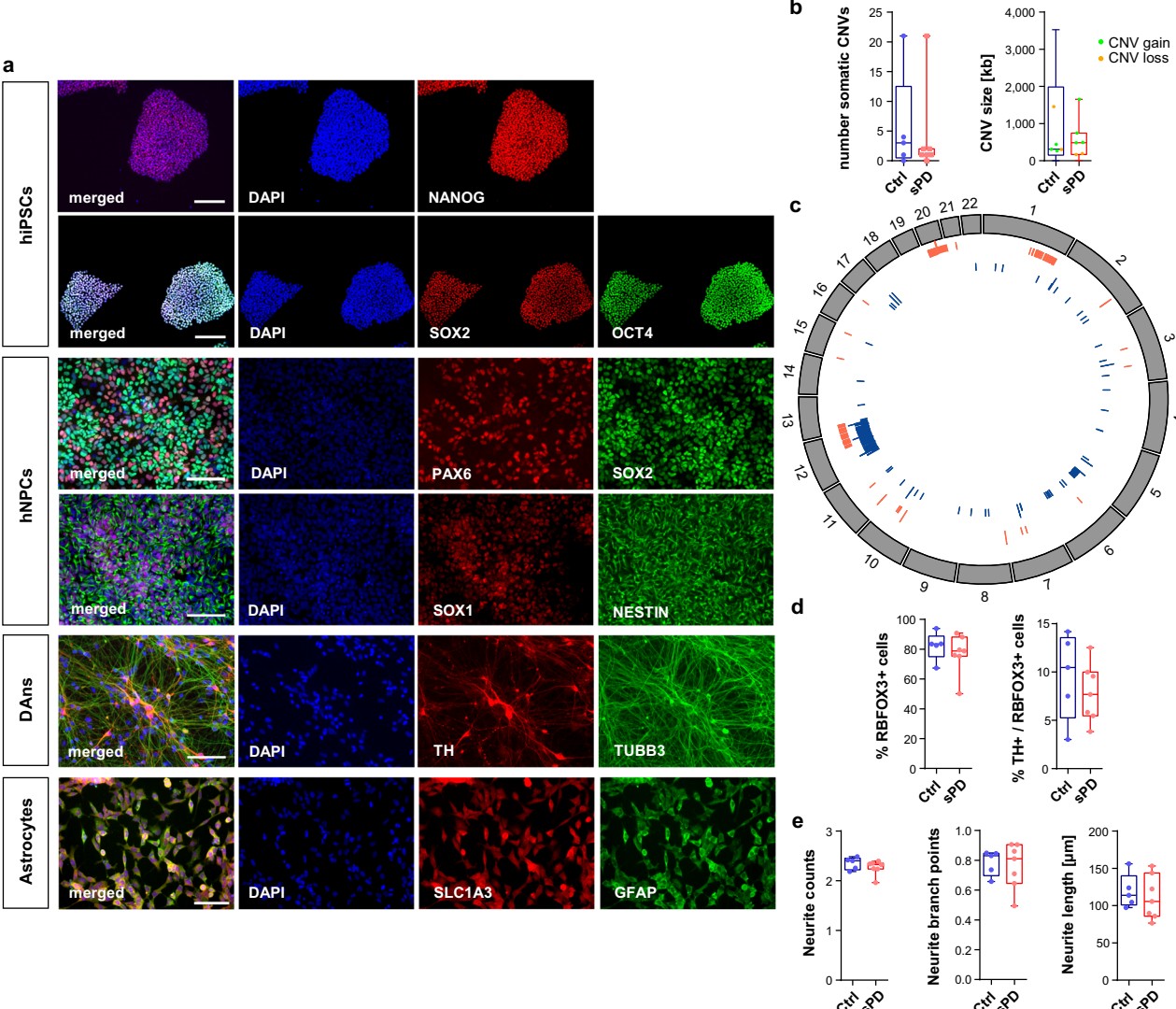

**Fig. 1 | Characterization of hiPSC derived hNPCs, neurons and astrocytes.**
**a** Immunostainings exemplarily shown for O3H-R1-003. hiPSC pluripotency stain-ing for markers OCT4, NANOG, SOX2. Scale bar=200 μm. hNPC staining for markers SOX1, SOX2, NESTIN, PAX6. Scale bar=100 μm. Neuron staining for markers TUBB3 and DAn marker TH. Scale bar=100 μm. Astrocyte staining for markers GFAP and SLC1A3. Scale bar=100 μm. **b** Summary of somatic CNVs identified in hNPC clones by chromosomal microarray analysis shown as total number of somatic CNVs detected per analyzed clone and average length of CNVs (in kb; green=copy number gain; orange=copy number loss) per analyzed clone. *n* = 5 Ctrl and 7 sPD patients. **c** Circos plot showing the genomic distribution of somatic CNVs in Ctrl

(blue) and sPD (red) clones. **d** Quantification of RBFOX3 (synonym: NeuN) positive as well as TH / RBFOX3 double-positive cells in DAn populations. *n* = 5 Ctrl and 7 sPD clones, in triplicates. **e** Characterization of neurite morphologies of DAns. Boxplots show the average number of neurites emerging from TH positive cell bodies, their average number of branch points and their average length. *n* = 5 Ctrl and 7 sPD clones, in triplicates. Boxplots display the median and range from the 25th to 75th percentile. Whiskers extend from the min to max value. Each dot represents one patient. *P*-values were determined by two-sided *t*-test **d** (right), **e**; two-sided Mann–Whitney-U test **b, d** (left). *$p < 0.05$, **$p < 0.01$, ***$p < 0.001$. Source data are provided as a Source Data file.

Further analysis and validation of the clusters showed that neither of these clusters was dominated by either gender, age, or individual donors (Fig. 3c, d; Supplementary Fig. 1f), nor was there an altered contribution of Ctrl or sPD cells to clusters (Supplementary Fig. 1d, e). However, one Ctrl individual was slightly overrepresented in the NCSC cluster. Removal of this respective individual did not change the overall results. Therefore, we kept this respective individual as part of the analysis.

We further validated the different clusters by applying an RNA velocity analysis. RNA velocity predicts the future state of individual cells based on observed splicing kinetics. Splicing kinetics reveal gene-specific expression dynamics that can be projected onto exist-ing low-dimensional embeddings indicating the direction of the gene expression changes and thus developmental trajectories. As cell cycle

processes may mask such developmental trajectories, we removed cell cycle effects from the data. As expected, this led to the NSC population not being subdivided into four but instead two clusters which could clearly be distinguished according to the expression of the marker genes of NSC1 and NSC2 (Supplementary Fig. 1g). Root cell analysis using the RNA velocity vectors indicated one bioinformati-cally determined developmental point of origin (root) within the NSC1a population from which two developmental trajectories arose (Fig. 3e). Whereas the NSC1 fate clearly had its endpoint in the population of immature neurons, the NSC2 fate was within the NSC2 population itself, however, also exhibiting a clear endpoint. Thus, the NSC1a cluster contains the root population underlying neuronal dif-ferentiation and RNA velocity analysis validated our initial subdivi-sions into the described clusters.

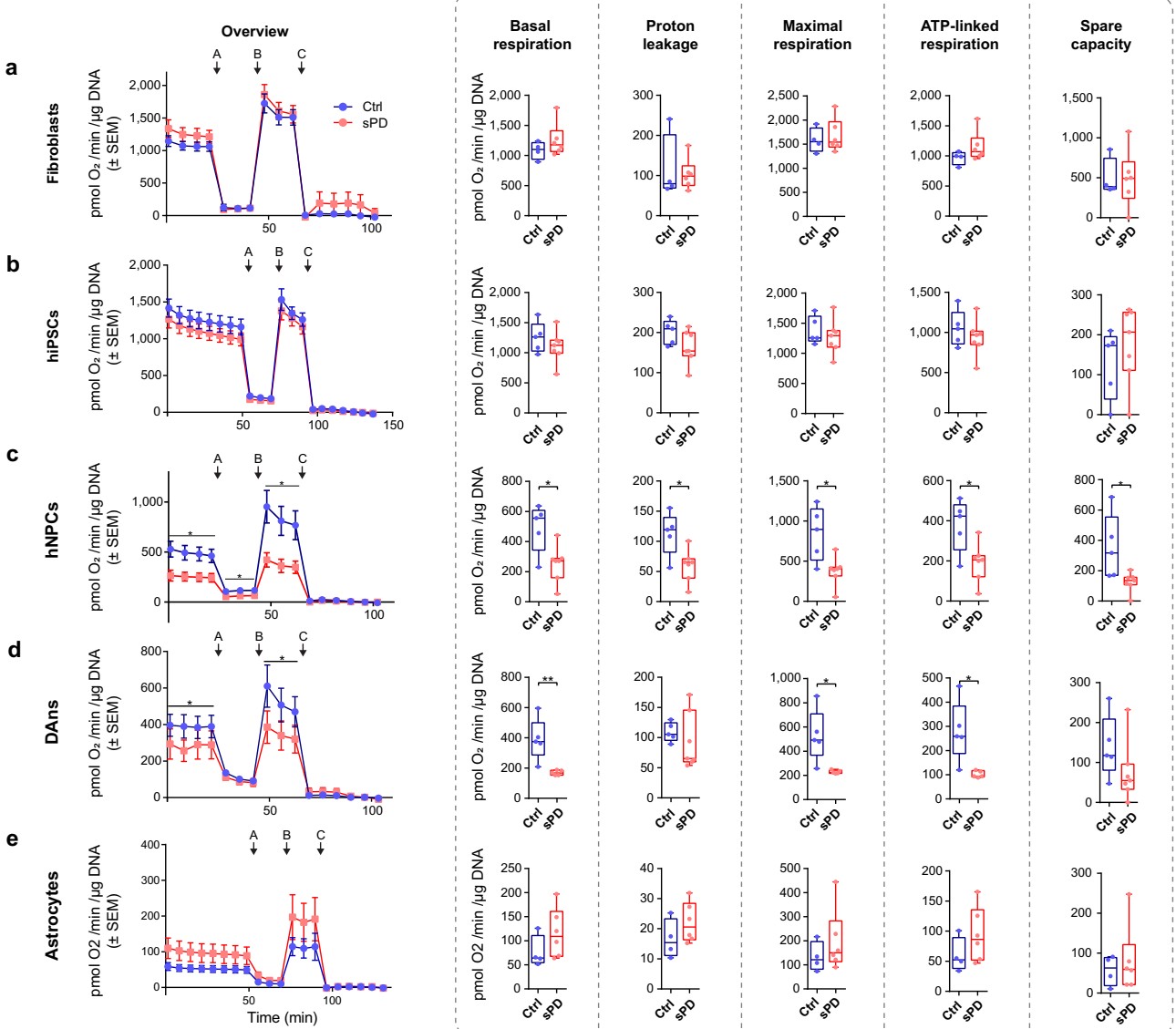

**Fig. 2 | Mitochondrial dysfunction develops in sPD upon dopaminergic neuron differentiation. a** Mitochondrial stress test performed in fibroblasts derived from $n = 4$ Ctrl and 7 sPD patients (in triplicates), **(b)** hiPSCs, **(c)** hNPCs, **(d)** DAns derived from 5 Ctrl and 7 sPD patients (in triplicates), or **(e)** astrocytes derived from 4 Ctrl and 6 sPD patients (in triplicates) using a Seahorse XFe96 Extracellular Flux Analyzer. Injected were (A) Oligomycin (1 µg/ml), (B) FCCP (0.5 µM), and (C) Rotenone (5 µM)/Antimycin A (2 µM). Measurement progression is shown with means±standard error of the mean (SEM). Boxplots display means of serial measurements for basal and maximal respiration (max), proton leakage as well as the difference between basal respiration and proton leak for ATP linked respiration (ATPl) and the difference between basal and maximal respiration for spare capacity (spare). Boxplots display the median and range from the 25th to 75th percentile. Whiskers extend from the min to max value. Each dot represents one patient. *P*-values were determined by two-sided *t*-test **a** (basal $p = 0.2972$; proton $p = 0.7397$; max $p = 0.6578$; ATPl $p = 0.1979$; spare $p = 0.9869$), **b** (basal $p = 0.3359$; proton $p = 0.1252$; max $p = 0.4998$; ATPl $p = 0.3951$; spare $p = 0.3994$), **c** (basal $p = 0.0142$; proton $p = 0.0150$; max $p = 0.0101$; ATPl $p = 0.0164$; spare $p = 0.0224$), **d** (basal $p = 0.0076$; proton $p = 0.4913$; max $p = 0.0146$; ATPl $p = 0.0129$; spare $p = 0.1838$); two-sided Mann–Whitney-U test **e** (basal $p = 0.1714$; proton $p = 0.2571$; max $p = 0.3524$; ATPl $p = 0.3524$; spare $p = 0.9999$). $^*p < 0.05$, $^{**}p < 0.01$, $^{***}p < 0.001$. Source data are provided as a Source Data file.

## Analysis of differentially expressed genes in sPD hNPCs

Subsequently, the clusters before cell cycle regression and the root population identified via RNA-velocity analysis were used to identify PD-associated gene expression alterations and to assess qualitative differences in cluster-specific pathological responses. We were able to identify 13,132 unique differentially expressed genes (DEGs) between Ctrl and sPD cells across any of the 9 clusters and the root cell population. Among them 8,592 protein-coding genes which were distributed over all clusters (Supplementary Fig. 3b; Supplementary Data 5). DEGs were robustly detected at different levels of expression. We selected 6 down- and 4 upregulated DEGs (*GLI1, GLI2, GLI3, FOXA2, BBS5, GET4, LINGO2, PITX3, SLC1A2,* and *SRCAP*)

and validated them by quantitative PCR with reverse transcription (RT–qPCR) in hNPC cultures (see below and Supplementary Fig. 3c).

Analysis of DEGs in the root population revealed that already at this early developmental stage gene sets associated with "Parkinson's Disease" and the "mitochondrial electron transport chain" were significantly overrepresented in sPD lines (Fig. 3f; Supplementary Fig. 3a). This is already validated by the mitochondrial dysfunction observed in hNPCs and DAns (Fig. 2c, d; Supplementary Fig. 8a). The heterogeneous response of clusters to the PD state is visualized by the heat map of the top 10 DEGs per cluster based on the average log fold-change (Fig. 4a).

Remarkable was the high number of repressed genes in the NSC clusters e.g. in NSC1a and NSC2a irrespective of the chosen significance

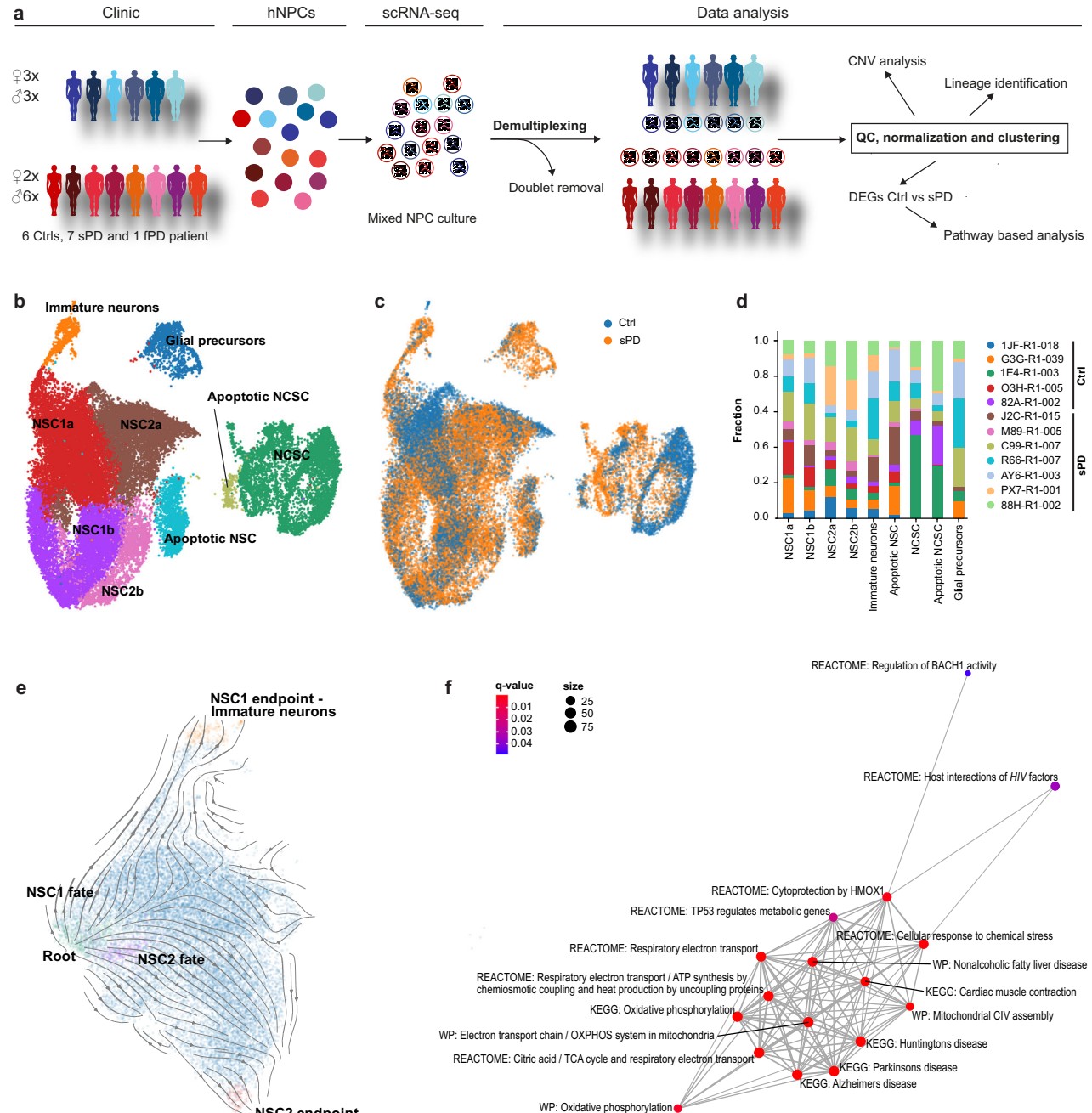

**Fig. 3 | scRNA-seq profiling and clustering. a** Experimental and bioinformatic workflow for multiplexing of 14 hNPC clones derived from 13 individuals. QC, quality control. **b** UMAP visualization of 30,557 annotated cells from 5 Ctrl and 7 sPD hNPC clones within the clusters NSC: neural stem cells, immature neurons, glial precursors, NCSC: neural crest stem cells and apoptotic NSC / NCSC. **c** UMAP visualization with annotations for the disease condition (Ctrl, sPD). **d** Contribution of individual patients to clusters. **e** RNA velocity analysis revealed two NSC trajectories on immature neurons (NSC1 fate) and NSC populations (NSC2 fate) as well as one developmental point of origin (root) after cell cycle effects were regressed out. **f** Detail of the network of enriched canonical pathways for DEGs of the root population with edges weighted by the ratio of overlapping gene sets. FDR corrected *p*-values are represented by *q*-values. Complete network is presented in Supplementary Fig. 3a.

threshold (NSC1a – 89%, NSC2a – 89%; *q* < 0.05 and NSC1a – 64%, NSC2a – 81%; *q* < 0.01 and | FC | > 20%) (Fig. 4b; Supplementary Fig. 4a, c, d). This repression was not caused by a general DNA hypermethylation in hNPCs (Supplementary Fig. 4b). Furthermore, the over-representation of downregulated genes was not as pronounced in other clusters e.g. in the immature neuron cluster (54%; *q* < 0.05 and 41%; *q* < 0.01 and | FC | > 20%) (Supplementary Fig. 4a, e), which supports the notion of a heterogeneous differential response to the PD state per cluster.

Overall, our results suggest that all cell populations were affected by sPD pathology at the transcriptional level. Single-cell resolution is thereby critical as changes in gene expression can be dynamic and opposed across cell identities.

**Dysregulated genes point towards primary cilia dysfunction**
To determine disease-associated cellular processes we performed a pathway enrichment analysis based on all DEGs of each cluster using multiple pathway databases (KEGG, WikiPathway (WP), and

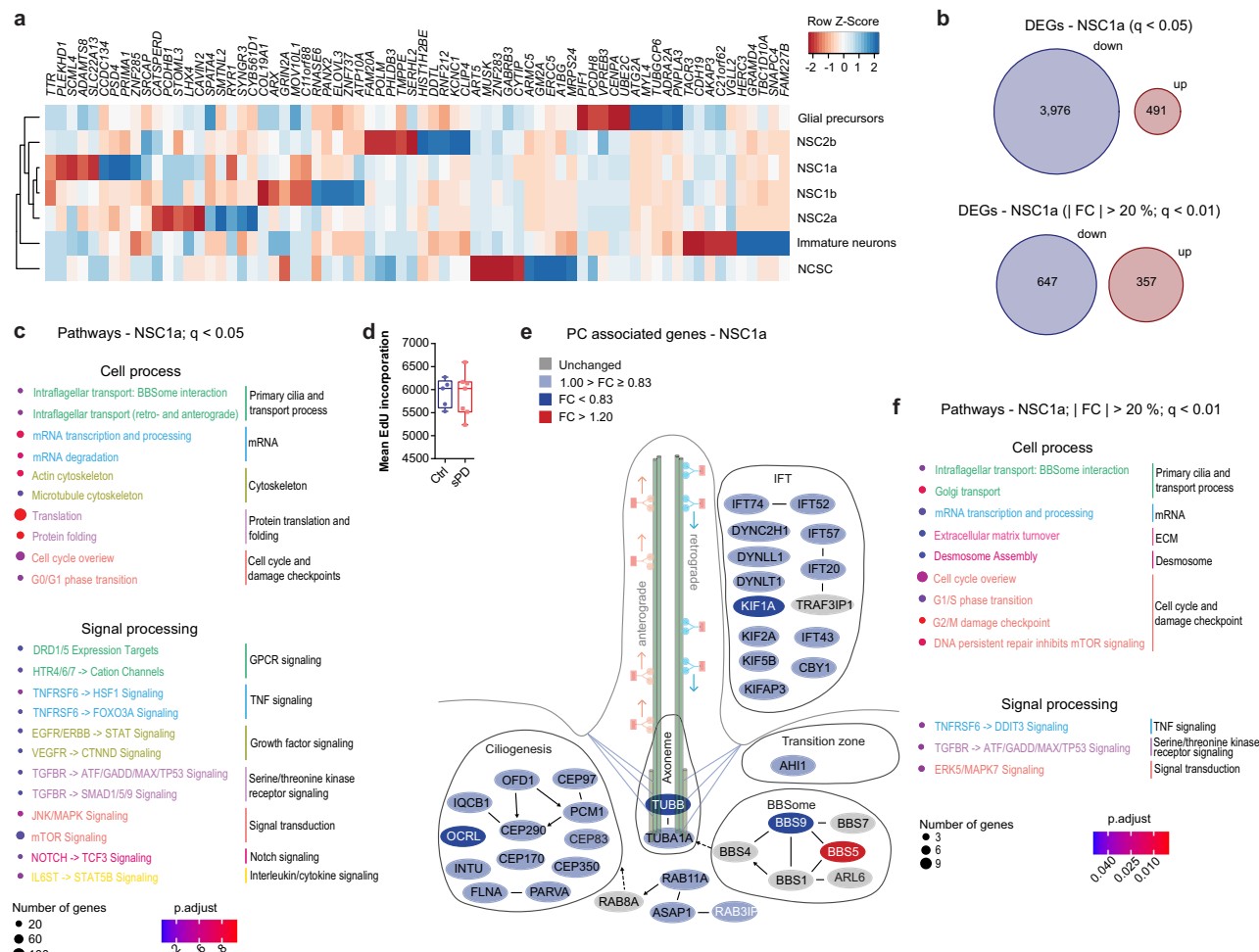

**Fig. 4 | Cluster specific gene expression changes in sPD pathology. a** Heatmap showing log2 transformed fold changes (FC) with columns scaled by z-score for the top ten up- and downregulated DEGs of each cell cluster. Hierarchical clustering of cell types represented by the dendrogram reveals their similarity across DEGs. **b** Number of all and most (| FC | > 20%, $q < 0.01$) up- and downregulated DEGs in NSC1a. **c** Enriched pathways of the categories Cell process and Signal processing analyzed using Pathway Studio for the DEGs of NSC1a ($q < 0.05$). *P*-values were determined by one-sided Fisher's exact tests. FDR corrected *p*-values are represented by *q*-values. **d** Cell proliferation rate of hNPCs determined by EdU incorporation after 2 h of EdU treatment. $n = 5$ Ctrl and 7 sPD clones, in triplicates. *P*-values were determined by two-sided *t*-test. **e** Visualization of the "Intraflagellar

transport: BBSome interaction" pathway with manual annotations. Up- (red) and downregulated (blue) genes are sorted into functional categories intraflagellar transport (IFT), transition zone, BBSome, axoneme, and ciliogenesis. Color intensity is proportional to fold change. **f** Enriched pathways of the categories Cell process and Signal processing analyzed using Pathway Studio for the DEGs of NSC1a (| FC | > 20%; $q < 0.01$). *P*-values were determined by one-sided Fisher's exact tests. FDR corrected *p*-values are represented by *q*-values. Boxplots display the median and range from the 25th to 75th percentile. Whiskers extend from the min to max value. Each dot represents one patient. #$p < 0.1$, *$p < 0.05$, **$p < 0.01$, ***$p < 0.001$. Source data are provided as a Source Data file.

PathwayStudio)[24]. An overview of the top dysregulated pathways per cluster is shown in Table 1 (for full lists see Supplementary Data 6–8). Enriched pathways of the NSC clusters could be grouped mainly into the cellular process categories - cell cycle, oxidative phosphorylation (OXPHOS), PD and neurodegenerative diseases, protein translation and folding, mRNA turnover, cytoskeleton, and primary cilia (PC) - as well as into the signaling categories - receptor tyrosine kinase (RTK), G protein-coupled receptor (GPCR), tumor necrosis factor (TNF), and serine/threonine kinase receptor signaling (Fig. 4c; Supplementary Fig. 6).

The high proportion of cell cycle-associated processes, specifically in the b clusters, which were associated with the G2/M phase of the cell cycle, was, however, not surprising due to the cycling nature of the hNPCs. Still, the dysregulation of these pathways did not negatively impact the cell proliferation rate of sPD hNPCs (Fig. 4d).

It is also not surprising that mitochondrial pathways associated especially with the OXPHOS were found to be affected in sPD as they are assumed to be a prominent pathological culprit contributing to

sPD etiology[5,11,12,25]. The dysregulation of OXPHOS pathways, as well as the enrichment of PD-associated terms further strengthens the suitability of these hNPCs as a veritable model for sPD. Fold changes of PD and OXPHOS associated genes for the NSC and immature neuron clusters are also visualized by a heatmap (Supplementary Fig. 5). Indeed as mentioned before, we validated alterations in mitochondrial function by showing reduced mitochondrial respiration as well as reduced complex I activity in sPD hNPCs (Fig. 2c; Supplementary Fig. 8a).

Especially interesting, however, was the finding of pathways associated with PC in the NSCa clusters, especially NSC1a. Enrichment of cilia pathways mainly in the NSCa clusters, which were associated with the G1/S phase of the cell cycle, is not surprising. PC are hair-like cellular organelles that extend from the cell surface and are thought to function as signaling units[26,27]. They contain 9 pairs of microtubules (axoneme) which are surrounded by the cell membrane on most sides, but at the ciliary base are connected to the mother centriole also referred to as the basal body. Thus, PC formation is tightly coupled to

## Table 1 | Top 8 pathways significantly enriched in sPD

| Cluster | Database | Pathway | q-value |
|---|---|---|---|
| NSC1a | KEGG | Ribosome | 4.8E-33 |
| | | Oxidative phosphorylation | 1.0E-19 |
| | | Parkinson disease | 1.1E-18 |
| | | Huntington disease | 8.7E-15 |
| | | Prion disease | 2.0E-13 |
| | | Protein processing in endoplasmic reticulum | 3.3E-13 |
| | | Amyotrophic lateral sclerosis | 6.7E-12 |
| | | Non-alcoholic fatty liver disease | 8.8E-12 |
| | WikiPathways | Cytoplasmic ribosomal proteins | 8.1E-42 |
| | | Electron transport chain: OXPHOS | 2.0E-15 |
| | | VEGFA-VEGFR2 signaling pathway | 1.6E-12 |
| | | Translation factors | 3.1E-11 |
| | | Nonalcoholic fatty liver disease | 1.4E-09 |
| | | Oxidative phosphorylation | 1.2E-06 |
| | | Ciliary landscape | 1.6E-05 |
| | | Exercise-induced circadian regulation | 1.0E-04 |
| | PathwayStudio | Translation | 2.5E-29 |
| | | Protein Folding | 9.4E-13 |
| | | S/G2 Phase Transition | 6.3E-12 |
| | | APC/C-CDC20 Complex | 9.6E-12 |
| | | G1/S Phase Transition | 1.3E-11 |
| | | APC/C-FZR1 Complex | 7.7E-11 |
| | | Golgi Transport | 9.7E-10 |
| NSC1b | KEGG | Ribosome | 1.6E-24 |
| | | Coronavirus disease - COVID-19 | 2.9E-15 |
| | | Oxidative phosphorylation | 3.5E-12 |
| | | Thermogenesis | 3.6E-12 |
| | | Huntington disease | 5.4E-11 |
| | | Endocytosis | 5.4E-11 |
| | | Ubiquitin mediated proteolysis | 6.7E-10 |
| | | Parkinson disease | 1.2E-09 |
| | WikiPathways | Cytoplasmic ribosomal proteins | 1.9E-39 |
| | | VEGFA-VEGFR2 signaling pathway | 2.4E-13 |
| | | Electron transport chain: OXPHOS | 3.8E-11 |
| | | Cohesin complex - CdL syndrome | 6.6E-07 |
| | | Translation factors | 3.9E-06 |
| | | Nonalcoholic fatty liver disease | 3.9E-06 |
| | | Cell cycle | 1.0E-05 |
| | | Pathogenic *Escherichia coli* infection | 2.1E-05 |
| | PathwayStudio | Translation | 1.6E-21 |
| | | APC/C-CDC20 Complex | 1.9E-17 |
| | | Kinetochore Assembly | 1.4E-15 |
| | | G2/M Phase Transition | 1.8E-14 |
| | | APC/C-FZR1 Complex | 8.8E-14 |

## Table 1 (continued) | Top 8 pathways significantly enriched in sPD

| Cluster | Database | Pathway | q-value |
|---|---|---|---|
| | | S/G2 Phase Transition | 7.1E-13 |
| | | G1/S Phase Transition | 9.5E-13 |
| NSC2a | KEGG | Ribosome | 2.8E-26 |
| | | Coronavirus disease - COVID-19 | 2.2E-14 |
| | | Adherens junction | 2.4E-10 |
| | | Oxidative phosphorylation | 1.2E-09 |
| | | Protein processing in endoplasmic reticulum | 1.3E-08 |
| | | Thermogenesis | 4.0E-08 |
| | | Parkinson disease | 1.0E-07 |
| | | Pathways of neurodegeneration | 1.8E-07 |
| | WikiPathways | Cytoplasmic ribosomal proteins | 5.5E-40 |
| | | VEGFA-VEGFR2 signaling pathway | 4.0E-14 |
| | | Nonalcoholic fatty liver disease | 2.9E-05 |
| | | Electron transport chain: OXPHOS | 2.9E-05 |
| | | Androgen receptor signaling pathway | 8.0E-05 |
| | | EGF/EGFR signaling pathway | 2.1E-04 |
| | | Ciliary landscape | 2.1E-04 |
| | | Translation factors | 3.4E-04 |
| | PathwayStudio | Translation | 2.3E-27 |
| | | JNK/MAPK Signaling | 7.5E-08 |
| | | Golgi to Endosome Transport | 1.5E-07 |
| | | Actin Cytoskeleton | 1.5E-07 |
| | | Ras-GAP Regulation Signaling | 4.2E-07 |
| | | Golgi Transport | 5.6E-07 |
| | | InsulinR -> ELK/SRF/SREBF Signaling | 6.0E-07 |
| NSC2b | KEGG | Ribosome | 8.1E-45 |
| | | Coronavirus disease - COVID-19 | 7.4E-30 |
| | | Oxidative phosphorylation | 8.9E-11 |
| | | Diabetic cardiomyopathy | 7.1E-09 |
| | | Parkinson disease | 2.4E-08 |
| | | Cell cycle | 2.7E-08 |
| | | Prion disease | 5.2E-08 |
| | | Alzheimer disease | 2.3E-06 |
| | WikiPathways | Cytoplasmic ribosomal proteins | 5.5E-63 |
| | | Electron transport chain: OXPHOS | 4.9E-08 |
| | | VEGFA-VEGFR2 signaling pathway | 6.6E-08 |
| | | Cell cycle | 2.3E-05 |
| | | Translation factors | 1.0E-04 |
| | | Cohesin complex - CdL syndrome | 2.2E-04 |
| | | Nonalcoholic fatty liver disease | 2.6E-04 |
| | | Regulation of sister chromatid separation at the metaphase-anaphase transition | 1.2E-03 |

**Table 1 (continued) | Top 8 pathways significantly enriched in sPD**

| Cluster | Database | Pathway | q-value |
|---|---|---|---|
| | PathwayStudio | Translation | 8.9E-41 |
| | | Kinetochore Assembly | 1.6E-10 |
| | | APC/C-CDC20 Complex | 1.9E-10 |
| | | Spindle Assembly | 4.0E-08 |
| | | Centriole Duplication and Separation | 2.2E-07 |
| | | G2/M Phase Transition | 4.8E-07 |
| | | S/G2 Phase Transition | 1.9E-05 |
| Apoptotic NSC | KEGG | Ribosome | 6.6E-67 |
| | | Coronavirus disease - COVID-19 | 1.2E-52 |
| | WikiPathways | Cytoplasmic ribosomal proteins | 4.9E-90 |
| | PathwayStudio | Translation | 7.0E-69 |
| | | rRNA Transcription and Processing | 2.1E-04 |
| | | HIF1 Signaling | 3.1E-02 |
| | | CNR1/2 -> Vascular Motility | 7.9E-02 |
| | | AMPK Signaling | 7.9E-02 |
| | | HTR1 -> Vascular Motility | 8.0E-02 |
| | | Sialophorin -> CTNNB/MYC/TP53 Signaling | 8.0E-02 |
| | | VEGFR -> ATF/CREB/ELK-SRF Signaling | 8.1E-02 |
| Immature neurons | KEGG | Ribosome | 1.5E-50 |
| | | Coronavirus disease - COVID-19 | 3.5E-41 |
| | WikiPathways | Cytoplasmic ribosomal proteins | 3.0E-65 |
| | PathwayStudio | Translation | 4.1E-53 |
| | | rRNA Transcription and Processing | 3.5E-03 |
| | | HIF1 Signaling | 1.2E-01 |
| | | Frizzled Receptors -> ARRB1/ARRB2 non-Canonical Signaling | 1.2E-01 |
| | | NOTCH -> TCF3 Signaling | 1.2E-01 |
| | | TLR4 -> IRF Signaling | 1.2E-01 |
| | | CHRNA7 -> CREB Signaling | 1.2E-01 |
| | | HTR7 -> IL6 Production | 1.2E-01 |
| NCSC | KEGG | *Herpes simplex virus 1* infection | 1.4E-20 |
| | | Biosynthesis of cofactors | 8.6E-03 |
| | | Autophagy - animal | 1.2E-02 |
| | | Lysosome | 1.3E-02 |
| | | Ribosome | 1.4E-02 |
| | | Nicotinate and nicotinamide metabolism | 1.4E-02 |
| | | Other glycan degradation | 3.8E-02 |
| | | Hedgehog signaling pathway | 4.3E-02 |
| | WikiPathways | Genes related to primary cilium development (based on CRISPR) | 0.00164 |
| | | Ciliary landscape | 0.00861 |
| | | Ciliopathies | 0.00861 |
| | PathwayStudio | mRNA Transcription and Processing | 3.1E-07 |
| | | tRNA Transcription and Processing | 3.9E-05 |

**Table 1 (continued) | Top 8 pathways significantly enriched in sPD**

| Cluster | Database | Pathway | q-value |
|---|---|---|---|
| | | Golgi to Endosome Transport | 5.0E-05 |
| | | Intraflagellar Transport: Anterograde | 5.1E-05 |
| | | Intraflagellar Transport: Retrograde | 2.3E-04 |
| | | Single-Strand Nucleotide Excision | 2.7E-04 |
| | | Golgi Transport | 3.4E-04 |
| Apoptotic NCSC | KEGG | | |
| | WikiPathways | | |
| | PathwayStudio | mRNA Degradation | 7.7E-05 |
| | | tRNA Transcription and Processing | 7.7E-05 |
| | | PTPRC -> STAT6 Signaling | 1.6E-03 |
| | | IL13R -> STAT6 Signaling | 3.7E-03 |
| | | IL4R -> STAT Signaling | 3.7E-03 |
| | | Golgi to Endosome Transport | 4.8E-03 |
| | | Mitochondrial Fusion and Fission | 4.9E-03 |
| | | TNFRSF1A -> STAT Signaling | 5.8E-03 |
| Glial precursor | KEGG | Autophagy - animal | 1.4E-05 |
| | | Insulin signaling pathway | 3.0E-04 |
| | | Nucleocytoplasmic transport | 1.5E-03 |
| | | AMPK signaling pathway | 1.5E-03 |
| | | Viral life cycle - *HIV-1* | 1.7E-03 |
| | | Neurotrophin signaling pathway | 1.7E-03 |
| | | FoxO signaling pathway | 1.7E-03 |
| | | Chronic myeloid leukemia | 2.2E-03 |
| | WikiPathways | Insulin signaling | 7.3E-07 |
| | | ANGPTL8 regulatory pathway | 7.3E-07 |
| | | Glycosylation and related congenital defects | 7.3E-07 |
| | | Genes related to primary cilium development (based on CRISPR) | 5.2E-06 |
| | | Cytoplasmic ribosomal proteins | 8.6E-06 |
| | | EGF/EGFR signaling pathway | 7.7E-05 |
| | | Neurodegeneration with brain iron accumulation (NBIA) subtypes pathway | 8.4E-05 |
| | | Joubert syndrome | 6.5E-04 |
| | PathwayStudio | Golgi to Endosome Transport | 1.3E-10 |
| | | mRNA Transcription and Processing | 7.8E-10 |
| | | Nuclear Envelope in Cell Division | 5.0E-08 |
| | | Protein Nuclear Import and Export | 1.2E-07 |
| | | mRNA Degradation | 1.2E-06 |
| | | tRNA Transcription and Processing | 2.2E-06 |
| | | G2/M Phase Transition | 3.8E-06 |

Pathway enrichment analysis based on all DEGs of each cluster using multiple pathway databases (KEGG, WikiPathway and PathwayStudio).

the cell cycle, as centrioles are needed for chromosome segregation during mitosis resulting in PC disassembly.

Particularly two cilia-associated pathways - the microtubule-based intraflagellar transport (IFT) and IFT interaction with the BBSome - seemed to be affected in sPD within the NSC1a cluster (Fig. 4c). A dysregulation of these pathways goes along with the enrichment of significantly altered G protein-coupled receptors (GPCR) signaling pathways, e.g. serotonin signaling (HTR4/6/7 signaling)[28] and dopamine signaling (DRD1/5 signaling)[29], both of which are implicated in PD (Fig. 4c; Supplementary Data 6–8). This is not surprising, as the BBSome is a protein trafficking complex that recognizes ciliary targeting sequences of transmembrane proteins such as class A and B GPCRs or RTKs and mediates their binding to IFT complexes, thus their trafficking to the ciliary membrane[30,31]. Intriguingly, GPCRs are normally localized to neural PC[32] and ciliary export of activated GPCRs and downstream components for signal transduction is also modulated by the BBSome[33]. This suggests that the observed dysregulation of GPCR and RTK pathways was a consequence of ciliary dysfunction. This hypothesis is further supported by the dysregulation of downstream signaling components (e.g. *GNAS*, *ADCY2*, *ADCY9*, and several *MAPKs*) rather than the GPCRs themselves, such as serotonin and dopamine receptors (Fig. 4c; Supplementary Fig. 6; Supplementary Data 5–8).

In sum, we found a high proportion of cilia-associated genes in the NSC1a cluster being dysregulated. These genes could be allocated into different categories according to their functions and localization within the cilia such as intraflagellar transport (IFT: e.g. *IFT74*, *IFT52*, *IFT20*, *DYNC2H1*), transition zone (e.g. *AHI1*), axoneme (*TUBA1A*, *TUBB*), and ciliogenesis (e.g. *INTU*, *CEP83*, *OCRL*) (Fig. 4e). Of note was again the overrepresentation of downregulated genes in the BBSome and other cilia associated categories with one exception the *BBS5*, which was also validated via qPCR in hNPC cultures (Supplementary Fig. 3c).

As many DEGs, especially in the NSC1a cluster, were only slightly downregulated in sPD, we additionally performed a pathway enrichment analysis for DEGs with larger fold changes (| FC | > 20 %; $q < 0.01$) to elucidate the impact of strongly dysregulated genes on the aforementioned cell and signaling processes. After DEG thresholding, only 9 PathwayStudio terms were enriched including several cell cycle associated processes. Intriguingly under this stringent condition, cilia-associated pathways such as the "Intraflagellar transport: BBSome interaction" pathway (Fig. 4f) remained significant.

Taken together, the computational analysis of the scRNA-seq data of hNPCs derived from sPD patients points, amongst others, to an impaired cilia formation and function.

## Altered ciliary morphology and disrupted SHH signal transduction in neural cells derived from sPD patients

To functionally validate our finding that PC formation is disturbed in sPD patients we determined PC length in mitotic, untreated hiPSCs, hNPCs, and post-mitotic neurons, as well as astrocytes, derived thereof (Fig. 5; Supplementary Fig. 7a and 12c). With the same fraction (7–55%) of ciliated cells, sPD hNPC (Kolmogorov-Smirnov test (ks) $p = 0.0007$; linear mixed effects model (lm) $p = 0.04$) and neurons (ks $p = 2.2 \times 10^{-16}$; lm $p = 0.009$), but not hiPSCs (ks $p = 0.15$; lm $p = 0.48$) and astrocytes (ks $p = 0.31$; lm $p = 0.94$) exhibited significantly shorter PC as seen in distribution analysis. Ciliary length is closely associated with the cell cycle, thus, it might be influenced by differences in the proliferation rates. Yet, sPD patient- and healthy control-derived hNPCs exhibited similar proliferation rates (Fig. 4d). PC are known as hubs for various signaling pathways, among which SHH signaling is very prominent. Thus, we next investigated whether an impaired ciliary function is reflected in alterations of SHH signaling[32,34]. For SHH signaling, three transcription factors are known to mediate signal transduction namely GLI1, GLI2, and GLI3[35]. GLI1 acts as a transcriptional activator whose expression is low in hNPCs[36,37]. In contrast, GLI2

and GLI3 exhibit an evolutionarily conserved transcriptional duality - acting as activators or repressors depending on post-translational processing. They are thought to be the primary mediators of SHH signaling that regulate the expression of target genes, including *PTCH1* and the *GLIs* themselves[36,38,39]. Therefore, to investigate the effect of altered PC morphology and hence molecular network alterations on SHH signal transduction in sPD we focused on GLI3 expression, processing, and translocation. SHH functions through its receptor PTCH1. The binding of SHH inhibits PTCH1, which allows the translocation of SMO to the PC and subsequently prevents the degradation of full-length GLI3 (GLI3-FL) to GLI3-repressor (GLI3-R)[32,40]. GLI3-FL is thought to be neutral or functions as a weak transcriptional activator, whereas the degradation product GLI3-R acts as a repressor. In addition, proteolytic processing of GLI protein depends partly on phosphorylation by cAMP-dependent PKA which is a key regulatory component of SHH signaling and downregulated in sPD in the NSC1a cluster (Supplementary Data 5)[40–42]. Additionally, predicted GLI3-FL and GLI3-R target genes were significantly overrepresented (~ 22.7%; Fisher's Exact Test $p = 3.31 \times 10^{-13}$) in the DEGs of the NSC1a cluster (Fig. 6a). From these DEGs, we specifically validated the dysregulation of the SHH transcription factors *GLI1,2,3* and other GLI3 target genes such as *FOXA2*, *GBP1*, and *SHOX2* (Fig. 6b)[43].

The cytoplasmic GLI3 pool was unaffected, however, we showed that GLI3-FL and GLI3-R protein levels were reduced in the nucleus of sPD hNPCs, indicating impaired GLI3 processing (Fig. 6c; Supplementary Fig. 9). Similarly, also nuclear GLI3-R levels in DAns were reduced in sPD (Fig. 6d).

In contrast to the reduced levels of GLI3-FL and GLI3-R in the nucleus, levels of ciliary GLI3 (FL and R) and SMO were not altered in sPD hNPCs (Supplementary Fig. 8b, c). Together with the alterations in the BBSome and intraflagellar transport pathway as seen in the scRNA-seq analysis, this suggests that in sPD hNPCs and DAns SHH induced and ciliary mediated GLI3 processing is impaired.

Following we used cyclopamine to modulate SHH signaling and to determine its effect on the phenotypes observed in sPD hNPCs and DAns, such as shortened PC length and impaired mitochondrial respiration. Cyclopamine is a cell-permeable steroidal alkaloid that specifically and directly interacts with SMO thereby repressing SHH-signaling[44,45]. Upon inhibition of SHH signal transduction by cyclopamine, GLI3 processing was restored in sPD hNPCs to similar levels as in Ctrls (Fig. 7a). As expected, also the expression of GLI targets such as *PTCH1*, *GLI3*, *FOXA2*, *GBP1*, and *SHOX2* was affected. Expression of *PTCH1*, *FOXA2*, and *SHOX2* was repressed, while expression of *GLI3* and *GBP1* was enhanced. Under this condition, the differences between Ctrl and sPD hNPCs were completely abolished (Fig. 7b). In addition, by inhibiting SHH-signaling we were able to eliminate the differences in PC length (Fig. 7c; Supplementary Fig. 8d and 12a, b) without interfering with the cell proliferation rate (Fig. 7d) and to rescue the deficit in mitochondrial respiration. Upon inhibition with cyclopamine, both basal and maximal mitochondrial respiration were increased in sPD hNPCs to levels comparable to Ctrl (Fig. 7e; Supplementary Fig. 12b). Intriguingly, repression of SHH signaling seemed to mainly affect sPD hNPCs as ciliary length and mitochondrial respiration were less altered in Ctrls.

In sum, these findings indicate that in sPD hNPCs SHH signaling is increased resulting in altered ciliary morphology and in deficits in mitochondrial respiration, which is known to be involved in the pathoetiology of sPD. Thus, disrupted ciliary function accompanied by enhanced SHH signal transduction may represent an early hub in sPD etiology.

## Ciliary alterations in postmortem tissue of sPD patients

A previously published meta-analysis of transcriptomes of postmortem substantia nigra tissues derived from 83 sPD patients and 70 Ctrl patients identified 946 genes to be dysregulated in sPD[46]. Of note

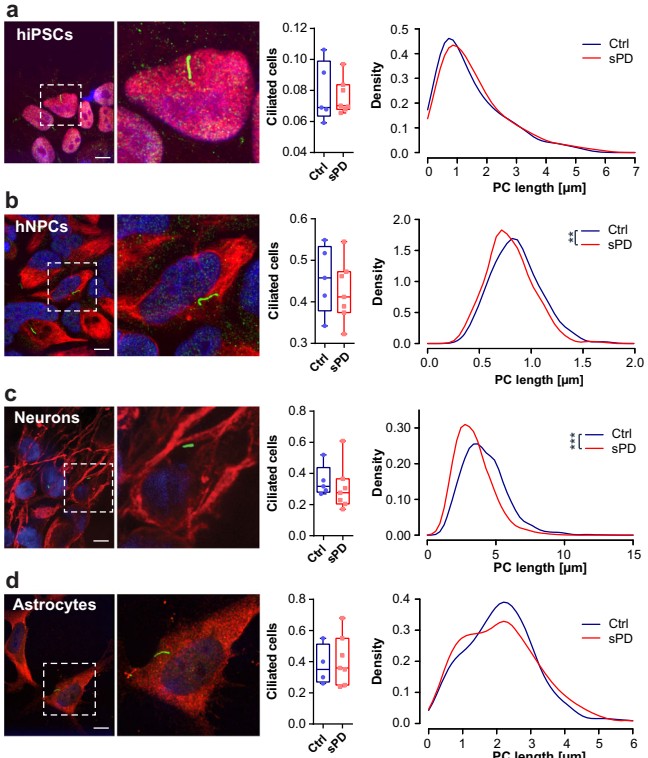

**Fig. 5 | Primary cilia morphology in sPD. a** (right) Density plot illustrating the distribution of PC length (in μm) in hiPSCs. (middle) Fraction of ciliated cells. PC length was measured in immunostainings (left) of hiPSCs positive for anti-SOX2 (red) and anti-ARL13B (green). $n = 37$-$79$ cilia per clone. **b** (right) Density plot illustrating the distribution of PC length (in μm) in hNPCs. (middle) Fraction of ciliated cells. PC length was measured in immunostainings (left) of hNPCs positive for anti-NES and anti-ARL13B. $n = 74$-$123$ cilia per clone. **c** (right) Density plot illustrating distribution of PC length (in μm) in neurons. (middle) Fraction of ciliated cells. PC length was measured in immunostainings (left) of neurons positive for anti-TUBB3 (red) and anti-ARL13B (green). $n = 131$-$385$ cilia per clone. **d** (right) Density plot illustrating the distribution of PC length (in μm) in astrocytes. (middle) Fraction of ciliated cells. PC length was measured in immunostainings (left) of astrocytes positive for anti-GFAP (red) and anti-ARL13B (green). $n = 100$ cilia per clone. All experiments were performed in triplicates, $n = 5$ Ctrl and 7 sPD. Immunostainings are exemplarily shown for G3G-R1-039 with scale bars=10 μm. Boxplots display the median and range from the 25th to 75th percentile. Whiskers extend from the min to max value. Each dot represents one patient. $P$-values were determined by two-sided $t$-test **a** (middle), **b** (middle), **c** (middle), **d** (middle); two-sided Kolmogorov-Smirnov test (ks) + linear mixed effects model (lm) **a** (right - ks $p = 0.15$; lm $p = 0.48$), **b** (right - ks $p = 0.0007$; lm $p = 0.04$), **c** (right - ks $p = 2.2 \times 10^{-16}$; lm $p = 0.009$), **d** (right - ks $p = 0.31$; lm $p = 0.94$). $^\#p < 0.1$, $^*p < 0.05$, $^{**}p < 0.01$, $^{***}p < 0.001$. Source data are provided as a Source Data file.

was the high proportion of repressed genes also in these datasets: approximately 72% of DEGs were downregulated in sPD patients (Fig. 8a; Supplementary Fig. 12f). This is remarkably similar to the high number of repressed genes identified by us in the neural stem cell clusters NSC1a and NSC2a (88% and 93% of DEGs).

Predicted GLI3 target genes were also significantly over-represented (~29.8%; Fisher's Exact Test $p = 1.24 \times 10^{-5}$) in the DEGs identified by[46] (Fig. 8b). This is remarkably similar to our observations in the NSC1a cluster (~28.7%; Fisher's Exact Test $p = 1.67 \times 10^{-14}$) and demonstrates the relevance of altered ciliary SHH signal transduction to PD progression also in postmortem material of patients.

Furthermore, pathway analysis showed that DEGs were also significantly enriched for cell process pathways associated with ciliary function such as "Intraflagellar Transport: Retrograde and Anterograde" ($q = 0.007$) (Fig. 8c, Supplementary Data 10), and a high

proportion of cytoskeleton and cell cycle-associated processes was evident. Similar to our cellular model, several genes associated with ciliary categories intraflagellar transport (IFT: e.g. *IFT57*, *DYNC2LI1*), BBSome (e.g. *BBS4*, *BBS7*), axoneme (*TUBB*), and ciliogenesis (*OCRL*) were downregulated in sPD (Fig. 8d).

Overall, the transcriptome data from postmortem sPD patients support our in vitro finding in diseased patients and verify the hypothesis that ciliary dysfunction is a hallmark of sPD.

To validate the effect of the dysregulation of cilia associated pathways on ciliary morphology in postmortem brain tissue of sPD patients, we analyzed cilia length in striatal tissue of 6 sPD patients (LBD stage 6, AD (Braak & Braak) 1-3, and Thal phase 0-2) and 6 age- and sex-matched Ctrls (LBD stage 0, AD (Braak & Braak) 1-3, Thal phase 0-2) (Supplementary Data 14). Indeed, striatal neurons of sPD patients exhibited a significantly altered distribution of PC lengths (ks $p = 2.2 \times 10^{-16}$; lm $p = 2.5 \times 10^{-13}$). This is due to a prolongation of PCs in neurons, rather than a shortening of PC as observed in patient-derived hNPCs and neurons (Fig. 8e, f; Supplementary Fig. 7b). However, in the occipital cortex of 6 sPD patients (LBD stage 6, AD (Braak & Braak) 1-3, and Thal phase 0-2) and 6 age- and sex-matched Ctrls (LBD stage 0, AD (Braak & Braak) 1-3, Thal phase 0-2) (Supplementary Data 14), neuronal PC morphology was not affected (ks $p = 0.45$; lm $p = 0.62$) (Fig. 8g; Supplementary Fig. 7b).

The obvious discrepancy to the shortened neuronal PCs in hNPCs and neurons derived from hiPSC might be due to the far advanced depletion of dopamine in the striatum of late-stage sPD patients[47]. To validate this assumption, we determined neuronal PC length in the dorsal striatum and cortex of 6-OHDA treated animals representing a toxin-induced PD model. The unilateral injection of 6-OHDA into the mid-forebrain bundle drastically diminished the dopaminergic innervation of the ipsilateral striatum due to the loss of DANs in the substantia nigra (Fig. 8h). This loss of dopaminergic innervation was accompanied by a prolongation of striatal neuronal cilia compared to the unaffected contralateral side (ks $p = 0.001$; lm $p = 0.04$) (Fig. 8i, j; Supplementary Fig. 7c). Furthermore, in the cortex of these mice, which do not receive such a substantial dopaminergic input, neuronal PC morphology was not affected (ks $p = 0.25$; lm $p = 0.73$) (Fig. 8k; Supplementary Fig. 7c). Thus, the elongation of PC in the anterior striatum of advanced sPD patients (LBD stage 6) is highly likely due to the loss of dopaminergic input.

In sum, also at this far advanced stage of PD progression PC of striatal neurons display morphological alterations. The observed elongation at this late stage of the disease, however, highly likely reflects compensation for dopamine depletion rather than early pathological events such as shortened PC which are modeled in patient-derived hNPCs and young neurons.

## Altered ciliary morphology in familial forms of PD

Next, we analyzed whether cilia length is also affected in hNPCs derived from hiPSCs of familial PD (fPD) patients. To do so, we used a *phosphatase and tensin homolog induced kinase 1* (*PINK1*)-deficient hiPSC with a corresponding isogenic Ctrl (see methods). *PINK1* is, after *PARKIN*, the second most common cause for autosomal recessive familial PD[5]. Its function is to regulate proper mitophagy, but also mitochondrial transport[48,49]. Patients carrying *PINK1* mutations develop clinical phenotypes that resemble those suffering from sPD, including slow disease progression[50,51].

To gain a better understanding of altered molecular pathways in a *PINK1*-deficient human cellular model, hNPCs were analyzed using scRNA-seq together with their isogenic Ctrl and the sPD clones and their Ctrls. The *PINK1*-deficient clone and its isogenic Ctrl were differently reprogrammed than the sPD clones. It could not be anticipated that these clones reflect the heterogeneity seen in the sPD clones and their Ctrls and, thus, were clustered separately (Fig. 9a; Supplementary Fig. 11 and 12d, g, h). However, the heterogeneity was

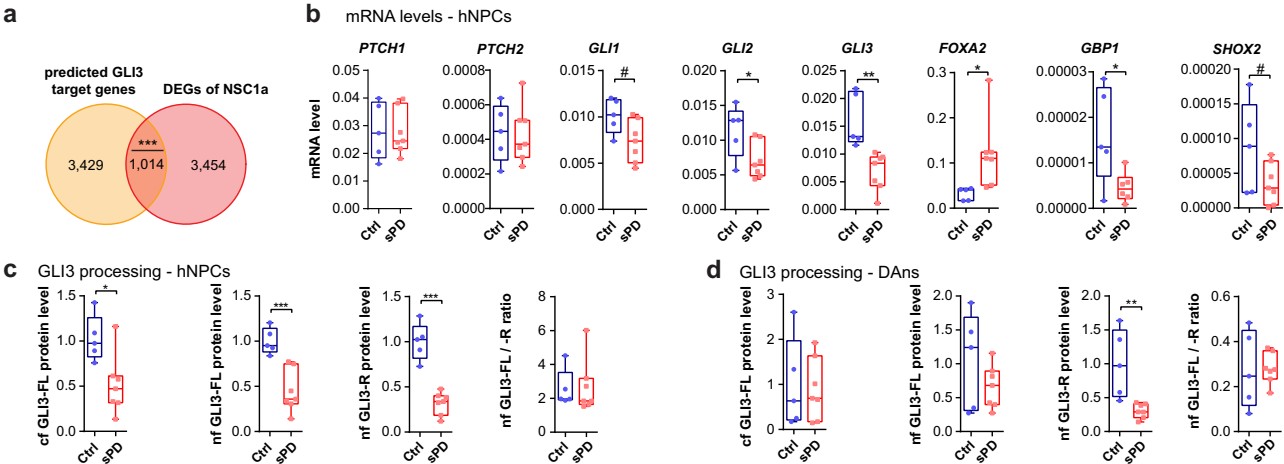

**Fig. 6 | Primary cilia dysfunction in sPD. a** Overlap of genome wide predicted GLI3 target genes using the MatInspector (Genomatix) with all DEGs from NSC1a. **b** Normalized gene expression levels of the SHH receptors *PTCH1*, *PTCH2*, the SHH transcription factors *GLI1*, *GLI2*, *GLI3* and other signaling targets *FOXA2*, *GBP1*, *SHOX2* analyzed on mRNA level by RT-qPCR in hNPCs. **c** Quantification of full length GLI3 (GLI3-FL) and GLI3 transcriptional repressor (GLI3-R) in the cytoplasmic (cf) or nuclear (nf) fraction of protein extracts from hNPCs and (**d**) DANs analyzed by Western blot. Cf and nf GLI3 levels were normalized to levels of ACTB or H1-0, respectively. All experiments were performed in triplicates, $n = 5$ Ctrl and 7 sPD clones. Boxplots display the median and range from the 25th to 75th percentile. Whiskers extend from the min to max value. Each dot represents one patient. *P*-values were determined by two-sided *t*-test **b** (*PTCH1 p* = 0.9180; *PTCH2 p* = 0.9333; *GLI1 p* = 0.0508; *GLI2 p* = 0.0402; *GLI3 p* = 0.0030; *GBP1 p* = 0.0318; *SHOX2 p* = 0.0986), **c** (cf GLI3-FL *p* = 0.0156; nf GLI3-FL *p* = 0.0009; GLI3-R *p* = 0.0001), **d** (cf GLI3-FL *p* = 0.8308; nf GLI3-FL *p* = 0.2297; GLI3-R *p* = 0.0048; nf GLI3-FL/R ratio *p* = 0.9541); two-sided Mann-Whitney-U test **b** (*FOXA2 p* = 0.0303), **c** (nf GLI3-FL/R ratio *p* = 0.9001); one-sided Fisher's Exact Test **a** (*p* = 3.31 × 10⁻¹³). #*p* < 0.1, *p* < 0.05, **p* < 0.01, ***p* < 0.001. Scale bars = 10 µm. Source data are provided as a Source Data file.

remarkably similar with regard to the NSC1 and immature neuron clusters, which were common elements of all hNPC clones analyzed (Fig. 3b; Fig. 9a). These clusters exhibited a similar marker gene expression to their sPD hNPC counterparts (Supplementary Data 4 and 11). Contrary to the sPD clones, which were analyzed in the same library, approximately the same proportion of genes were up and downregulated in the *PINK1*-deficient population in the NSC1 cluster (Fig. 9b). Furthermore, the PC-related cell process pathways "Intra-flagellar Transport: Retrograde and Anterograde" were dysregulated in the NSC1 clusters of *PINK1*-deficient ($q = 0.007$ and $0.009$) hNPCs along with predominantly cell cycle-related processes (Fig. 9c, d; Supplementary Data 12 and 13). The latter has been linked to *PINK1* previously by demonstrating that a deletion increases the proportion of cells in the G2M phase and causes major defects in cell cycle progression[52].

As a result of the alterations in intraflagellar transport the *PINK1*-deficient hNPC populations exhibited significantly shorter PC (ks $p = 7.2 × 10^{-8}$; lm $p = 0.025$), whereas the percentage of ciliated cells did not change (30–55%) (Fig. 9e, f; Supplementary Fig. 10a).

Furthermore, the PC length of striatal neurons (Fig. 9g, h, i; Supplementary Fig. 10b) of *Pink1* ko mice (ks $p = 3.4 × 10^{-12}$; lm $p = 1.6 × 10^{-4}$), including cholinergic neurons (ks $p = 4.6 × 10^{-5}$; lm $p = 1.2 × 10^{-3}$) was reduced. Again, alterations in PC morphology were restricted to receiving brain regions (e.g. striatum) involved in the nigrostriatal circuit, as PC morphology was not affected in the SNpc (ks $p = 0.34$; lm $p = 0.72$) (Fig. 9j; Supplementary Fig. 10b) nor the occipital cortex of these mice (ks $p = 0.34$; lm $p = 0.73$) (Fig. 9k; Supplementary Fig. 10b). These findings support the notion that shortened PC are part of prodromal PD before DAn degeneration occurs, as in the *Pink1* ko PD mouse model no DAn degeneration was observed[53].

## Discussion

In this study, we used sPD and fPD derived hiPSCs and their neuronal derivatives (hNPCs and DANs) to determine PD-associated cellular and molecular alterations. This system offers the advantage of modelling sporadic diseases in patient derived rejuvenated cells[8,9]. Thus, these cells have a patient specific genetic background[54] and do not show any

signs of neurodegeneration yet. We found that a major PD-associated phenotype - mitochondrial impairment - is already present in sPD patient derived hNPCs. In addition, an unbiased molecular characterization of hNPCs using scRNA-seq led to the identification of PC as a key molecular hub affected in sPD. Thus, the analysis of this model derived from sPD patients suggests, that dysfunctional PC accompanied by mitochondrial dysfunction is a very early event in and might even be at the base of sPD pathoetiology.

Concerning alterations in PC biology our analysis reveals that in hNPCs genes responsible for the correct length of PC, shuttling of signaling molecules through the transition zone, and ciliary transport are affected. Also in two recent reports performing either transcriptome analysis in NSCs of postmortem PD patients or hiPSC derived neurons from young-onset PD patients, downregulation of PC-associated pathways has been observed, however, was not further investigated[55,56]. PC exhibit a microenvironment insulated from the cytosol compartment and an enrichment of receptor tyrosine kinase (RTK) and GPCR membrane receptors and their downstream effectors[57]. Thus, the observed dysregulation of these signaling pathways amongst which are the PD relevant serotonergic and dopaminergic signaling via HTR4/6/7 and DRD1/5, respectively[28,29,58], further supports the notion of ciliary dysfunction in early sPD. The alterations in PC-related genes are highly likely to cause alterations in PC morphology (length of PC) and function (altered SHH signaling), both of which we validated in our model systems.

We also identified dysregulated PC-related gene expression in postmortem substantia nigra of late-stage sPD patients based on a previously published meta-analysis[46]. Indeed, also in postmortem tissue of sPD patients in a far advanced disease stage (LBD stage 6) PC morphology was altered, however, elongated. This at the first glance contradictory observation is highly likely based on the characteristic of PC to react very dynamically to their environment in a tissue-specific manner. On one hand, cilia were shown to extend upon dopamine signaling in the renovascular system[59], arteries, and endothelial cells[60]. On the other hand, and fitting to our postmortem analysis, loss of midbrain dopaminergic inputs on striatal neurons induces the elongation of striatal neuronal PC in 6-OHDA or MPTP treated mice and

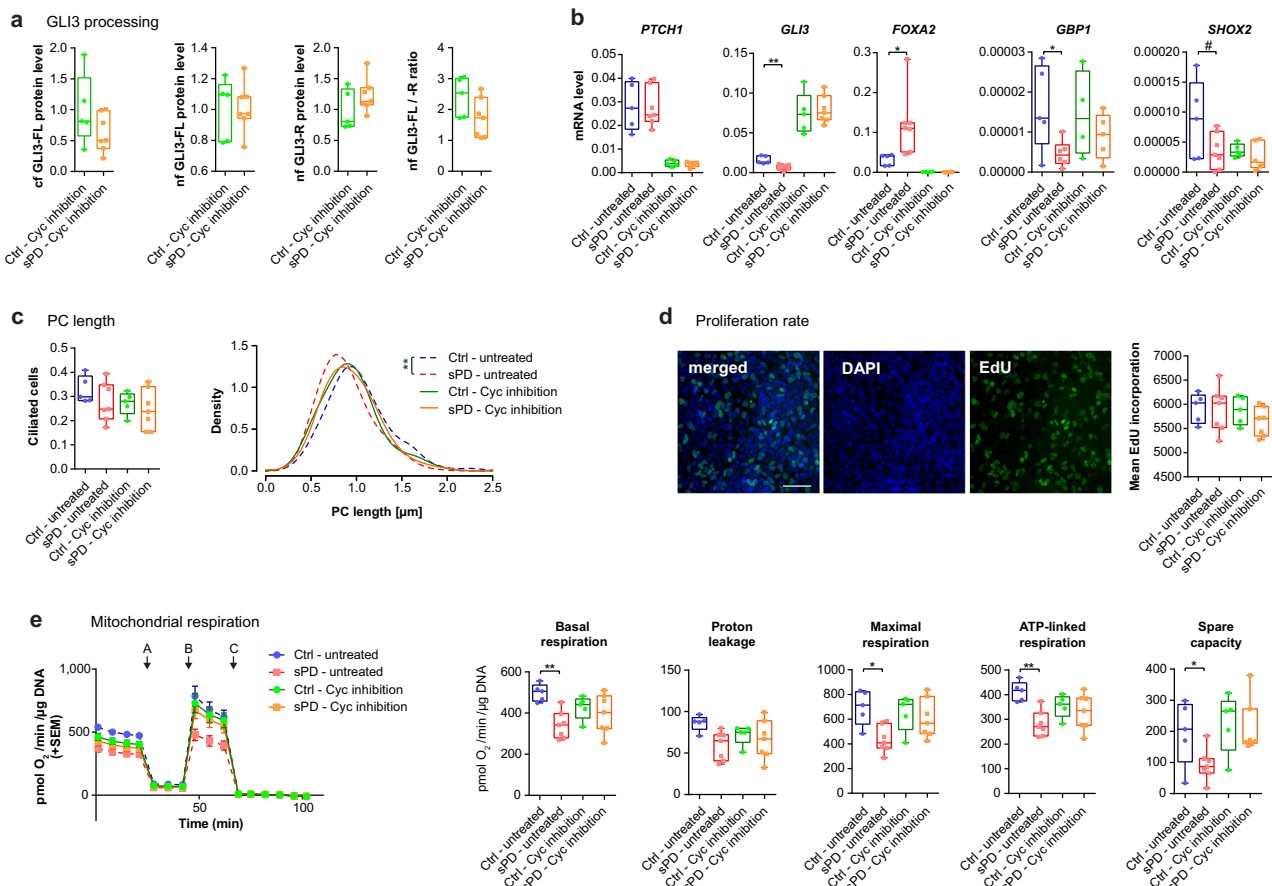

**Fig. 7 | Interfering with SHH signaling rescues PD associated alterations.**
**a** Quantification of full length GLI3 (GLI3-FL) and GLI3 transcriptional repressor (GLI3-R) in the cytoplasmic (cf) or nuclear (nf) fraction of protein extracts from hNPCs upon cyclopamine (Cyc) inhibition (10 μM for 4 days). **b** SHH receptor *PTCH1*, transcription factor *GLI3* and the signaling targets *FOXA2, GBP1, SHOX2* analyzed on mRNA level by RT-qPCR in hNPCs (DMSO ctrl and Cyc treated – 10 μM for 4 days). **c** (right) Density plot illustrating the distribution of PC length (in μm) in a hNPC population (DMSO ctrl and Cyc treated – 10 μM for 4 days). (left) Fraction of ciliated cells. PC length was measured in immunostainings of hNPCs positive for anti-NES and anti-ARL13B. Analyzed were $n = 40$ cilia per clone. **d** Cell proliferation rate of hNPCs (DMSO ctrl and Cyc treated – 10 μM for 4 days) determined by EdU incorporation after 2 h of EdU treatment. **e** Mitochondrial stress test performed in hNPC (DMSO ctrl and Cyc treated – 10 μM for 4 days) using a Seahorse XFe96 Extracellular Flux Analyzer. Injected were (A) Oligomycin (1 μg/ml), (B) FCCP (0.5 μM), (C) Rotenone (5 μM)/Antimycin A (2 μM). Measurement progression is shown with means ± standard error of the mean (SEM). Boxplots display means of

serial measurements for basal and maximal respiration, proton leakage as well as the difference between basal respiration and proton leak for ATP linked respiration and the difference between basal and maximal respiration for spare capacity. All experiments were performed in triplicates, $n = 5$ Ctrl and 7 sPD clones. Boxplots display the median and range from the 25th to 75th percentile. Whiskers extend from the min to max value. Each dot represents one patient. *P*-values were determined by two-sided *t*-test **a** (Cyc cf $p = 0.1688$; nf GLI3-FL $p = 0.9198$; GLI3-R $p = 0.1932$; nf GLI3-FL/R ratio $p = 0.0940$), **b** (Cyc PTCH1 $p = 0.5821$, GLI3 $p = 0.8361$, GBP1 $p = 0.3518$, SHOX2 $p = 0.3658$), **c** (left - untreated $p = 0.2691$; Cyc $p = 0.5563$), **d** (untreated $p = 0.9068$; Cyc $p = 0.2045$), **e** (untreated - basal $p = 0.0009$; proton $p = 0.0095$; max $p = 0.0046$; ATPl $p = 0.0008$; spare $p = 0.0449$) (Cyc - basal $p = 0.5347$; proton $p = 0.6068$; max $p = 0.6061$; ATPl $p = 0.5199$; spare $p = 0.7502$); two-sided Mann–Whitney-U test **b** (Cyc *FOXA2* $p = 0.4762$); two-sided Kolmogorov-Smirnov test (ks) + linear mixed effects model (lm) **c** (right - untreated ks $p = 4.8 \times 10^{-5}$; lm $p = 0.009$) (right - Cyc ks $p = 0.86$; lm $p = 0.80$). #$p < 0.1$, *$p < 0.05$, **$p < 0.01$, ***$p < 0.001$. Source data are provided as a Source Data file.

both in hemiparkinsonian rats and *dopamine receptor D2* (*DRD2*)-null mutant mice[47,61]. This indicates that the elongation of PC in late-stage sPD is a reaction to the changing environment during disease progression, i.e. a compensatory reaction to missing dopaminergic innervation in the adult brain.

Alterations in PC morphology and function in PD can be the consequence of dysfunctional molecular mechanisms at various levels. For example, posttranscriptional changes such as altered phosphorylation of proteins critical for ciliary function (like RAB8 and RAB10) have been identified in other models of fPD such as the *Lrrk2*-mutant mice[62,63], and *Lrrk2* mutations are associated with loss of cilia in vitro and in vivo via altered RAB phosphorylation[62,64]. The present enrichment analysis of the genetic networks in hNPCs of sPD patients and in *PINK1* deficient hNPCs (modeling fPD), however, strongly suggests a dysfunction of the intraflagellar transport, since most of the respective genes were dysregulated in our study, such as the BBSomes and the

IFTs. Also, a recent transcriptome analysis of postmortem tissue[55,56] and a meta-analysis of sPD patient tissue[46] support this notion. Furthermore, PINK1 has already been associated with dysfunctional intracellular transport mechanisms[49]. Parts of this pathway, such as dyneins and kinesins, are also crucial for intraflagellar transport. In addition, our human cellular sPD and fPD models as well as the *Pink1* mouse model exhibit shortening of PC but no loss of cilia, which is similar to the phenotype observed in several mutations of the intraflagellar transport machinery, such as *WDR35/IFT121* or *WDR19/IFT144* mutants[65–67].

Functionally, PC alterations result in modifications of major cilia-linked signaling pathways such as the SHH pathway. In fact, our data clearly show a massive dysregulation of potential SHH target genes including cilia-associated genes. Furthermore, we detect an increase in SHH-signaling as indicated by the reduced level of GLI3-FL and GLI3-R in the nucleus. Similar variations have also been observed in a variety

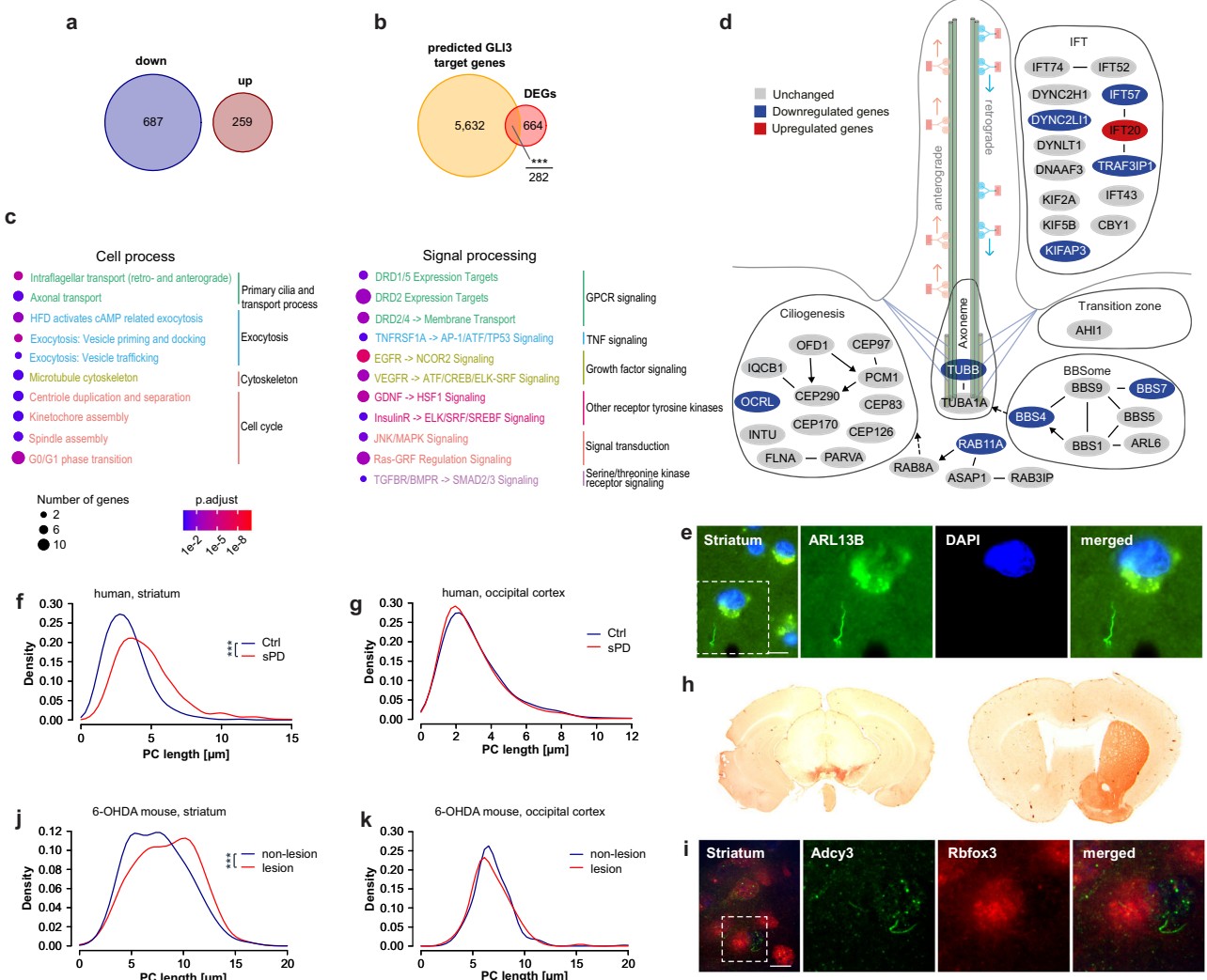

**Fig. 8 | Primary cilia associated genes are dysregulated in postmortem tissue of sPD patients. a** Number of up- and downregulated DEGs. **b** Overlap of genome wide predicted GLI3 target genes using the MatInspector (Genomatix) with DEGs. **c** Enriched pathways of the categories Cell process and Signal processing analyzed using Pathway Studio for DEGs identified in a previously published meta-analysis ($n = 83$ PD patients and 70 Ctrl patients) of substantia nigra transcriptome studies[46]. *P*-values were determined by one-sided Fisher's exact tests. FDR corrected *p*-values are represented by *q*-values. **d** Visualization of cilia associated DEGs. Up- (red) and downregulated (blue) genes are sorted into functional categories intraflagellar transport (IFT), transition zone, BBSome, axoneme, and ciliogenesis. **e** PC immunostaining with anti-ARL13B exemplarily shown for patient GA 2019/20 Ctrl-2. **f** Density plot illustrating PC length (in μm) distribution in human striatum ($n = 79$-133 cilia per individual), and **g** occipital cortex ($n = 75$-105 cilia per individual). PC length was measured in paraffin embedded brain sections

immunostained with anti-ARL13B. Brain sections were obtained from 6 PD patients and 6 Ctrls. **h** Immunostaining with anti-TH (left: substantia nigra; right: striatum) and **i** PC immunostaining with anti-Rbfox3 and anti-Adcy3 exemplarily shown for a 6-OHDA injected mouse. **j** Density plot illustrating neuronal PC length (in μm) distribution in the dorsal striatum ($n = 87$-115 cilia per mouse) and **k** cortex ($n = 71$-111 cilia per mouse) of 6-OHDA treated mice. PC length was measured in 40 μm free floating brain sections of 3 6-OHDA injected mice immunostained with anti-Adcy3 and anti-Rbfox3. Boxplots display the median and range from the 25th to 75th percentile. Whiskers extend from the min to max value. Each dot represents one patient. *P*-values were determined by two-sided Kolmogorov-Smirnov test (ks) + linear mixed effects model (lm) **f** (ks $p = 2.2 \times 10^{-16}$; lm $p = 2.5 \times 10^{-13}$), **g** (ks $p = 0.45$; lm $p = 0.62$), **j** (ks $p = 0.001$; lm $p = 0.04$), **k** (ks $p = 0.25$; lm $p = 0.73$); one-sided Fisher's Exact Test **b** ($p = 1.24 \times 10^{-5}$). $^{*}p < 0.05$, $^{**}p < 0.01$, $^{***}p < 0.001$. Scale bar = 10 μm. Source data are provided as a Source Data file.

of mutants (e.g. *IFT88*, *DYNC2H1*, *KIF3A* knockdown) exhibiting disruption of the ciliary transport machinery[68]. In addition, mouse models exhibiting loss or partial loss of the major ciliary transport machinery protein IFT-A[65] also exhibit an increase in SHH-signaling. Thus, the pattern of SHH-signaling dysregulation again hints towards alterations in intraflagellar transport. Our additional observation that inhibiting SHH signaling leads to an elongation of PC specifically in sPD is a novel finding. In this context, it is interesting to note that amongst the predicted GLI3 target genes dysregulated in sPD patient cells around 20% can be associated with PC (list of cilia-associated genes obtained from the UniProtKB database). These 38 SHH-dependent genes encompass structural genes, genes involved in intraflagellar transport as well as genes regulating ciliary signaling and several of them have been

associated with ciliogenesis and regulation of PC length. Thus, SHH-signaling might not only be regulated by cilia length but possibly also regulates cilia length itself in a feedback loop. A hypothesis that needs to be validated in the future.

Of therapeutic potential is the finding that the inhibition of this overactive SHH-signaling pathway in sPD hNPCs resulted in a rescue of the deficits both in PC morphology and mitochondrial respiration hinting towards a molecular interplay between PC mediated SHH signaling and mitochondrial function/respiration. The exact molecular underpinnings for this interplay between mitochondrial and PC function are still missing and will be the subject of further studies. Yet, reinforcing our findings, it has recently been reported that SHH-inhibition increases cellular (mitochondrial) health in NPCs derived

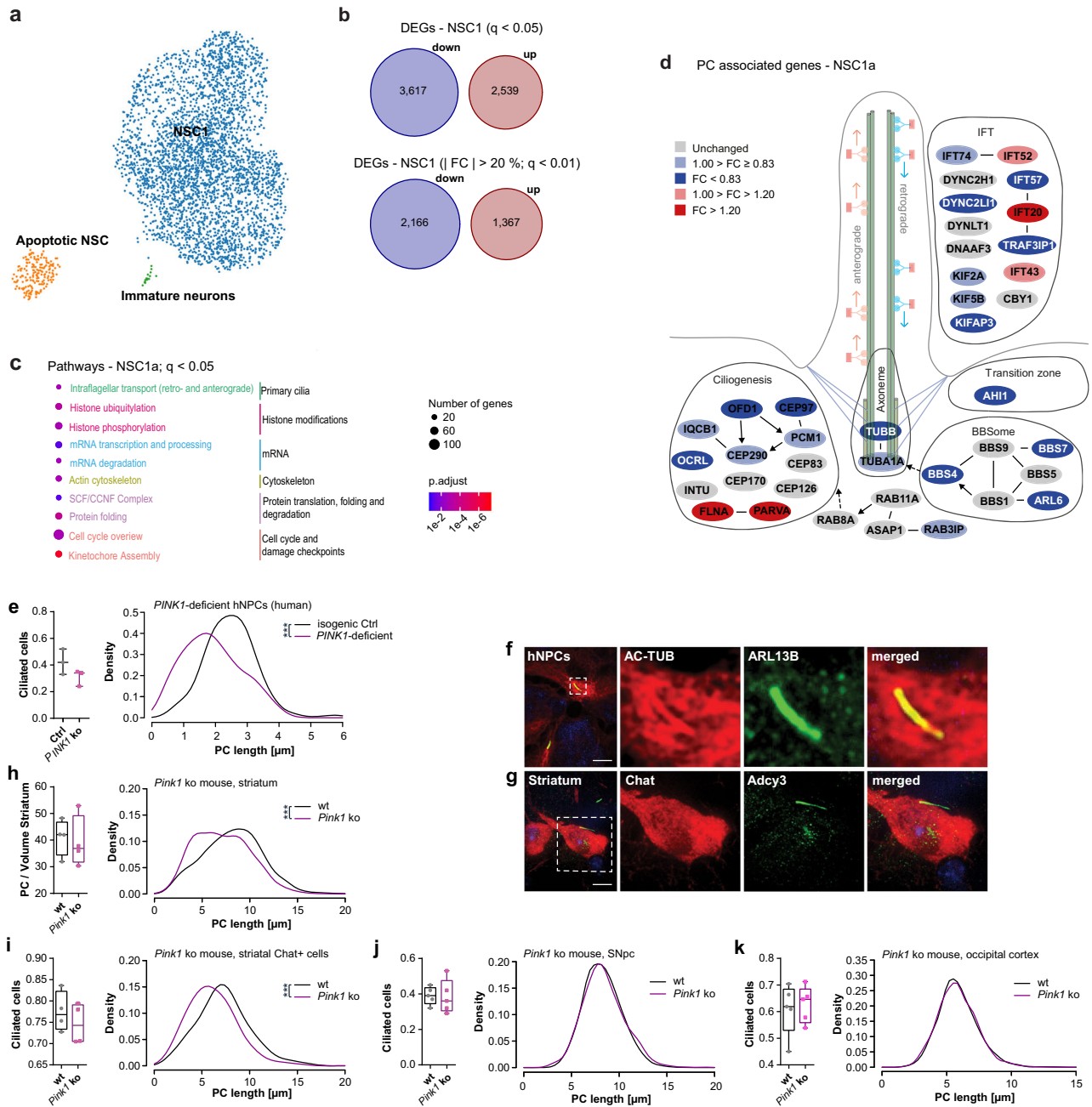

**Fig. 9 | Primary cilia in fPD. a** UMAP visualization of 3671 annotated cells from a *PINK1* ko and isogenic Ctrl hNPC cell line within the clusters NSC: neural stem cells, apoptotic NSC, and immature neurons. **b** Number of all and most (| FC | > 20%; q < 0.01) up- and downregulated DEGs in the NSC1 cluster. FC, fold change. **c** Enriched pathways of the categories Cell process analyzed using Pathway Studio for the DEGs of the NSC1 cluster. *P*-values were determined by one-sided Fisher's exact tests. FDR corrected *p*-values are represented by *q*-values. **d** Visualization of the "Intraflagellar transport: BBSome interaction" pathway with manual annotations. Up- (red) and downregulated (blue) genes are sorted into functional categories intraflagellar transport (IFT), transition zone, BBSome, axoneme, and ciliogenesis. Color intensity is proportional to fold change. **e** (right) Density plot illustrating the distribution of PC length (in μm) in a hNPC population. (left) Fraction of ciliated cells. PC length was measured in immunostainings of *PINK1* ko and isogenic Ctrl hNPCs with anti-ARL13B. *n* = 38-89 cilia per clone and replicate (three independent differentiations). **f** PC immunostaining with anti-AC-TUB and anti-ARL13B exemplarily shown for isogenic Ctrl of *PINK1* ko hNPCs. **g** PC immunostaining with anti-Chat and anti-Adcy3 exemplarily shown for a *Pink1* ko mice. **h** (left) Fraction of ciliated cells (in a striatal volume of 115 μm x 87 μm x 40 μm) and (right)

density plot illustrating the distribution of PC length (in μm) in the mouse dorsal striatum (*n* = 206-256 cilia per mouse) and **i** mouse striatal cholinergic neurons (*n* = 46-100 cilia per mouse). Fraction of ciliated cells and PC length was measured in 40 μm free floating brain sections from 4 *Pink1* ko and 4 wt mice immunostained with anti-Adcy3 and **h** anti-Rbfox3 or **i** anti-Chat. **j** (left) Fraction of ciliated cells and (right) density plot illustrating the distribution of PC length (in μm) in the mouse SNpc (*n* = 172-300 cilia per mouse) and the **k** occipital cortex (*n* = 210-636 cilia per mouse). Fraction of ciliated cells and PC length was measured in 40 μm free floating brain sections from 5 *Pink1* ko and 5 wt mice immunostained with anti-Adcy3 and anti-Rbfox3. Boxplots display the median and range from the 25th to 75th percentile. Whiskers extend from the min to max value. Each dot represents one mouse or replicate. *P*-values were determined by two-sided Mann-Whitney-U test **e** (left - *p* = 0.4000); two-sided *t*-test **h** (left *p* = 0.7697), **i** (left *p* = 0.4236), **j** (left *p* = 0.9034), **k** (left *p* = 0.7527); two-sided Kolmogorov-Smirnov test (ks) + linear mixed effects model (lm) **e** (right - ks *p* = 7.2 × 10⁻⁸; lm *p* = 0.025), **h** (right - ks *p* = 3.4 × 10⁻¹²; lm *p* = 1.6 × 10⁻⁴), **i** (right - ks *p* = 4.6 × 10⁻⁵; lm *p* = 1.2 × 10⁻³), **j** (right - ks *p* = 0.34; lm *p* = 0.72), **k** (right - ks *p* = 0.34; lm *p* = 0.73). *p < 0.05, **p < 0.01, ***p < 0.001. Scale bar = 10 μm. Source data are provided as a Source Data file.

from embryonic stem cells[69] and that in granular precursor cells of the cerebellum SHH stimulation leads to loss of mitochondrial membrane potential and reduced ATP levels[70]. Both observations are indicative for mitochondrial impairments upon SHH-stimulation in NPCs. Mitochondrial impairment is a molecular mechanism known to underlie the pathoetiology of sPD. It is thought to lead to an accumulation of cellular damage induced by reactive oxygen species and low energy levels (ATP levels), thereby facilitating and/or initiating further disease processes[71–73].

Boosting mitochondrial function has been an attractive target for candidate drug therapies, however, so far only with limited success[74]. This may be due to a failure of specifically addressing correct pathways at the right time prior to advanced stages with neuronal loss. In this context, the specific inhibition of the overactive SHH-signaling pathway by SHH-inhibitors (i.e. FDA-approved vismodegib and sonicdegib[75]) early during disease progression, thereby minimizing mitochondrial dysfunction, might become an attractive therapeutic option for halting or slowing down sPD progression. An approach which might attract additional attention since the phenotypes observed in this manuscript are highly cell type specific, thus a targeted therapy might be feasible. As mentioned before, evidence is accumulating that there exists a differential activity of SHH signaling in early versus late stage of the disease. This inhibition of the SHH signaling pathway might indeed be beneficial in early stages of the disease, as opposed to clinical stages where DANs are already lost[76].

In sum, our present findings of PC dysfunction which underlies ciliary mediated SHH-signaling and mitochondrial respiratory deficits represent a common molecular hallmark of early PD, both of sporadic and familial (*PINK1*, *LRRK2*) PD. Further studies into the molecular interplay between these early pathological events and their distinction from late pathological processes will pave the way to new and effective neuroprotective therapies such as the proposed modulation of ciliary SHH-signaling during early stages of sPD.

# Methods

## Ethical compliance
Work with human iPSCs was granted by the local ethics committees (No. 4485 and 4120, FAU Erlangen-Nuernberg, Germany; and No 422-13 and 357/19 S, Technical University Munich, Germany). All participants or their legal guardians gave written informed consent that allows working with and publishing of patient related material. Approved patient identifiers were used to conceal patient identity. All related experiments and methods were performed in accordance with relevant guidelines and regulations. Participants were not compensated.

sPD and Ctrl hiPSC lines were established, characterized and provided by the ForIPS consortium, for more detailed informations regarding the reprogramming and characterization procedures see[10]. hiPSC lines from 7 sPD patients and 5 Ctrls were used, in total 24 hiPSC lines with 2 clones per patient (Supplementary Data 1). To exchange selected sPD lines for research purposes the scientific board of the UKER biobank considers each request. The TALEN induced *PINK1*-deficient line and the isogenic Ctrl were established, characterized and provided by Julia Fitzgerald[77]. To exchange selected fPD lines for research purposes the scientific board of the Neuro-Biobank Tübingen considers each request.

All animal protocols and procedures were conducted with the approval for the ethical treatment of animals by the responsible animal welfare authority of the Regierung von Oberbayern (Government of Upper Bavaria) and considering the 3 R principle.

Ethical approval for the work with human postmortem material was obtained from the Institutional Review Board of TU Munich, Munich, Germany (#419/19 S-SR). All donors or authorized relatives had provided written informed consent. Approved patient identifiers were used to conceal patient identity. Tissue from the anterior striatum and the occipital cortex was obtained from the Neurobiobank Munich (NBM) (www.en.neuropathologie.med.uni-muenchen.de/neurobiobank/neurobiobank/index.html). Brain tissue was fixed in 4 % formalin for 5-120 days (at room temperature), thereafter embedded in paraffin and sliced into 5 μm sections. Participants were not compensated.

## Cell culture fibroblasts
Primary human dermal fibroblasts were cultured in fibroblast growth medium (DMEM/F12-GlutaMAX (Life Technologies, 31331093) supplemented with 10% fetal bovine serum (PAN-Biotech, P30-1402)) at 37 °C, 5 % $CO_2$, 21 % $O_2$. At 60-70% confluency, fibroblasts were detached using 1 ml 0.05% trypsin-EDTA (Life Technologies, 25300096) for 5 min at 37 °C, chopped with a 1000 μl pipette tip and diluted in 5 ml fibroblast growth medium. Fibroblasts were harvested at 200 g for 5 min, resuspended in 1 ml fibroblast growth medium and seeded 1:5 on Geltrex (Thermo Fisher Scientific, A1413302) coated plates. Medium was changed every second day.

## Cell culture hiPSCs
hiPSCs were maintained under feeder-free conditions on Geltrex coating and in mTesR1 medium (StemCell Technologies, 85850) at 37 °C, 5% $CO_2$, 21 % $O_2$, if not indicated otherwise. At 70% confluency, cells were detached using StemMACS Passaging Solution XF (Miltenyi Biotec, 130-104-688) for 6 min at 37 °C. StemMACS Passaging Solution XF was aspirated and hiPSC colonies were harvested in 1 ml mTeSR1 and chopped using a 1000 μl pipette tip. Harvested hiPSCs were diluted and seeded on Geltrex coated plates. hiPSCs at passage 45-65 were used for all analyses. hiPSCs were screened regularly for pluripotency and chromosomal aberrations (Fig. 1a−c).

## Human neural precursor cell differentiation
For reasons of comparability of results obtained with hiPSC distributed by the ForIPS consortium and their derivatives, it was recommended to use the differentiation protocols published by[78]. We did not switch differentiation protocols during the course of the ongoing project because specifically mitochondrial function is heavily dependent on culture conditions.

hiPSCs were differentiated into human small molecule neural progenitor cells (hNPCs) by embryoid body (EB) formation following the protocol described by[78] with minor adaptations. At 70% confluency, hiPSC colonies were detached using 2 mg/ml collagenase type IV (Thermo Fisher Scientific, 17104019) for 40 min at 37 °C, 5% $CO_2$, 21% $O_2$. Detached hiPSC colonies were harvested in knockout serum replacement medium (80% KnockOut DMEM (Life Technologies, 10829018) supplemented with 20% Knockout serum replacement (Life Technologies, 10828028), 10 μM SB431542 (Miltenyi Biotec, 130-106-275), 0.5 μM purmorphamine (Tocris Bioscience, 4551), 1 μM dorsomorphin (Tocris Bioscience, 3093/10), 3 μM CHIR99021 (Tocris Bioscience, 4423/10), 4.44 nM FGF-8b (Miltenyi Biotec, 130-095-741), 1% non-essential amino acids (Life Technologies, 11140035), 1% L-glutamine (Life Technologies, 5030024), 150 μM ascorbic acid 2-phosphate (Sigma, A8960-5G) and 0.02% beta-mercaptoethanol (Life Technologies, 31350010)). hiPSCs were cultivated for two days on an orbital shaker at 37 °C, 5% $CO_2$, 21% $O_2$ and 80 rpm with daily media changes. At day 3, EBs were transferred into neuronal precursor medium (50% DMEM/F12-GlutaMAX, 50% Neurobasal (Life Technologies, 21103049) supplemented with 1% B27 (minus vitamin A, 12587010, Life Technologies), 0.5% N2 (Life Technologies, 17502048), 10 μM SB431542, 0.5 μM purmorphamine, 1 μM dorsomorphin, 3 μM CHIR99021, 4.44 nM FGF-8b, 150 μM ascorbic acid 2-phosphate and 0.02 % beta-mercaptoethanol). EBs were cultivated for 4 days at 37 °C, 7% $CO_2$, 21% $O_2$ and 80 rpm with daily media changes. On day 6, EBs were transferred into neural precursor maintenance medium (neural precursor medium deprived of purmorphamine and CHIR99021). The following day, EBs were seeded unharmed on Geltrex coated plates

and expanded for 3-4 days in neural precursor maintenance medium at 37 °C, 7% $CO_2$, 21% $O_2$ with daily media changes. At day 10-11 hNPCs were dissociated using Accutase (Sigma, A6964-500ML) for 10 min at 37 °C, 7% $CO_2$, 21% $O_2$, chopped with a 1000 μl pipette tip and diluted in 5 ml neural precursor maintenance medium. hNPCs were harvested at 200 g for 5 min, resuspended in 1 ml neural precursor maintenance medium and seeded on Geltrex coated plates. Thereafter, hNPCs were cultivated at 37 °C, 7% $CO_2$, 21% $O_2$ with daily medium changes and passaged with Accutase at 80% confluency. After 2 passages, 0.5 μM purmorphamine was added to the maintenance medium.

## Human dopaminergic neuron differentiation

Human dopaminergic neuron differentiation was performed using 15 μg/mL poly-L-ornithine (Sigma, P3655) and 10 μg/mL laminin (Thermo Fisher Scientific, 23017015) coating and differentiation steps were done precisely as described by[78] using the following growth factors and supplements: 10 ng/mL human BDNF (Miltenyi Biotec, 130-093-811); 10 ng/mL human GDNF (Miltenyi Biotec, 130-096-291); 1 ng/mL human Tgf-beta3 (Miltenyi Biotec, 130-094-008); 100 ng/mL human FGF-8b; 200 μM ascorbic acid 2-phosphate; 1 μM purmorphamine; 500 μM dbcAMP (Sigma, D0627-1G). Human dopaminergic neurons were differentiated at 37 °C, 7 % $CO_2$, 21 % $O_2$ for at least 50 days.

## Human astrocyte differentiation

Human astrocyte differentiation was performed as described by[79] with minor adaptations. hNPCs were passaged with Accutase at 80% confluency and counted using a Neubauer improved cell counting chamber (Carl Roth, PK361). Cells were resuspended in astrocyte differentiation medium (50 % DMEM/F12-GlutaMAX, 50% Neurobasal supplemented with 1% B27 (17504044, Life Technologies), 0.5% N2, 0.5x GlutaMaxx (Thermo Fisher Scientific, 35050061), 0.5x non-essential amino acids, 0.02% beta-mercaptoethanol, 2.5 μg/ml human Insulin (Sigma, I3536-100MG), 5 ng/ml CNTF (Miltenyi Biotec, 130-096-337), 10 ng/ml BMP4 (Miltenyi Biotec, 130-111-165), 10 ng/ml EGF (Miltenyi Biotec, 130-093-825), 8 ng/ml FGF2 (Miltenyi Biotec, 130-093-839), 10 ng/ml Heregulin1/Neuregulin1b (Biomol, 97642.1) containing 10 μM ROCK inhibitor Y27632 (Enzo Life Sciences, ALX-270333-M005) and were seeded on 15 μg/mL poly-L-ornithine and 10 μg/mL laminin coated 6-well plates at a density of 500.000 cells/well. Astrocytes were differentiated in astrocyte differentiation medium at 37 °C, 7% $CO_2$, 21% $O_2$ for at least 35 days with daily medium changes. At 80 % confluency, astrocytes were passaged with Accutase and seeded on 15 μg/mL poly-L-ornithine and 10 μg/mL laminin coated plates.

## Generation of PINK1-deficient hiPSC clones

HiPSCs (previously described and characterized in[78]) were cultured in self-made E8 media on Vitronectin (VTN-N, Gibco, A31804) coated cell culture dishes. The hiPSCs were transfected with a TALEN and a homologous construct for PINK1 exon1 to knockout the PINK1 gene[77]. Briefly, the transfection of healthy hiPSCs was performed using an Amaxa Nucleofector II with the Stem cell Nucleofection Kit (Lonza). The transfected hiPSCs were plated on VTN-N –coated 10 cm dishes in E8 medium containing 10 μM ROCK inhibitor Y27632. Homologous recombined hiPSC colonies were selected with 250 μg/ml G418 (Biochrom, A 2912) or 10 μgml-1 Blasticidin (InvivoGen, ant-bl-5b) in the second round of TALEN transfection and re-plated in 12-well plates. Resistant hiPSC colonies were characterized by sequencing, qRT-PCR and Western blot to confirm successful homozygous gene knockout. TALENs were designed with the online tool TAL Effector Nucleotide Targeter 2.0 (Cornell University). The following RVD sequences were used for the TALEN monomers: HD HD NI NH NH NG NH NI NH HD NH NH NH NH HD and NI NH HD NG HD HD NH NG HD HD NG HD HD NH HD. Colony-PCR after the first TALEN reaction was conducted with the primers pCR8_F1 (5′-TTGATGCCTGGCAGTTCCCT-3′) and pCR8_R1 (5′-

CGAACCGAACAGGCTTATGT-3′). After the second Golden Gate reaction, the colony-PCR was performed with the primers TAL_F1 (5′-TTGGCGTCGGCAAACAGTGG-3′) and TAL_R2 (5′-GGCGACGAGG TGGTCGTTGG-3′).

## Immunocytochemistry and imaging of human cells

Cells were cultured on Geltrex (fibroblasts, hiPSC and hNPCs) or poly-L-ornithine/laminin (neurons and astrocytes) coated glass coverslips for at least 72 h. Cells were fixed with 10% Formalin (Sigma, F5554) for 20 min at 37 °C, washed twice with PBS (Thermo Fisher Scientific, 14190169) and permeabilized/blocked with PBS containing 1 % BSA (Sigma-Aldrich, A7906-500G) and 0.3% Triton X-100 (Sigma-Aldrich, T9284) for 15 min at room temperature. Primary antibodies were diluted in PBS containing 1% BSA and 0.3% Triton X-100 and antibody incubation was performed at 4 °C overnight. Cells were washed twice with PBS and incubated with secondary antibodies diluted in PBS containing 1% BSA and 0.3% Triton X-100 for 2 h at room temperature. Nuclei were stained using a 0.1 μg/ml DAPI (Thermo Fisher Scientific, 62248) - PBS solution for 10 min at room temperature. Cells were washed twice with PBS and coverslips were mounted using Aqua-Poly/Mount (Polysciences Inc., 18606-20). Primary antibodies were diluted as follows: AC-TUB (T6793, Sigma-Aldrich; 1:1000), ARL13B (17711-1-AP, Proteintech; 1:500), GFAP (MAB360, Millipore; 1:250), GLI3 (AF3690, R&D; 1:100), NANOG (AF1997, R&D Systems; 1:200), NES (Ma1110, Thermo Fisher Scientific; 1:250), PAX6 (Ab78545, Abcam; 1:200), PITX3 (38-2850, Invitrogen; 1:300), POU5F1 (2840 S, Cell Signaling; 1:500), RBFOX3 (ab104224, Abcam; 1:800), SLC1A3 (NB100-1869, Novus Biologicals; 1:250), SMO (sc166685, Santa Cruz; 1:500), SOX1 (Ab87775, Abcam; 1:500), SOX2 (sc17320, Santa Cruz; 1:500), TH (P40101, PelFreez; 1:600), TUBB3 (T5076, Sigma-Aldrich; 1:1000). Secondary antibodies were diluted as follows: donkey-anti-goat IgG Alexa 488 (A11055, Thermo Fisher Scientific; 1:500), donkey-anti-mouse IgG Alexa 488 (A21202, Thermo Fisher Scientific; 1:500), donkey-anti-goat IgG Alexa 594 (A11058, Thermo Fisher Scientific; 1:500), donkey-anti-mouse IgG Alexa 594 (A21203, Thermo Fisher Scientific; 1:500), donkey-anti-rabbit IgG Alexa 488 (A21206, Thermo Fisher Scientific; 1:500), donkey-anti-rabbit IgG Alexa 594 (A21207, Thermo Fisher Scientific; 1:500). Fluorescence images were acquired using an Axiovert 200 M (ZEISS) with a 20X / 63X immersion objective or for z-stacks using a spinning disk Axio Observer Z1 confocal microscope (ZEISS) with a 100 X immersion objective. Images were analyzed using Fiji (based on ImageJ version 1.53c)[80] and whenever necessary, z-stacks were collapsed to maximum intensity projections. For hiPSC, hNPCs and astrocytes, at least 30 primary cilia (PC) per cell line and replicate were analyzed blinded from random fields. PC from cells positive for the cilia marker ARL13B and for the hiPSC marker SOX2 or the hNPC marker NES or the astrocyte marker GFAP, respectively, were selected for analysis. The fraction of ciliated neurons, as well as the neuronal cilia length, was assessed using Neurolucida version 2019.2.1 (MBF Bioscience) software and a Zeiss Axioplan 2 (Zeiss) microscope with a 100X immersion objective. PC from cells positive for cilia marker ARL13B and for the neuronal marker TUBB3 and RBFOX3 were selected for analysis. The analysis was performed blinded and an average of >30 PC per cell line were analyzed from random fields.

## Respiratory analysis

Fibroblasts were cultured on Geltrex coating and in fibroblast growth medium at 37 °C, 5% CO2, 21% O2. At 60−70% confluency, fibroblasts were passaged using 0.05% trypsin-EDTA, cell number was determined using a Neubauer improved cell counting chamber and 25,000 cells per well with at least 8 replicates per cell line were seeded on Geltrex coated XF96 cell culture microplates and incubated for 48 h with daily medium changes.

hiPSCs were maintained on Geltrex coating and in mTesR1 medium at 37 °C, 5% CO2, 21% O2. At 70% confluency, hiPSCs colonies were

passaged using Accutase for 6 – 8 min at 37 °C, sheared to single cells by pipetting using a 1000 μl tip and diluted in mTeSR medium containing 10 μM ROCK inhibitor Y27632. hiPSCs were harvested at 100 g for 5 min and resuspended in 1 ml mTeSR medium containing 10 μM ROCK inhibitor Y27632. Cell number was determined and 5000 cells per well with at least 8 replicates per cell line were seeded in mTeSR medium containing 10 μM ROCK inhibitor Y27632 on Geltrex coated XF96 cell culture microplates and incubated for 6 days with daily medium changes.

hNPCs were maintained on Geltrex coating and neural precursor maintenance medium at 37 °C, 7% $CO_2$, 21% $O_2$. At 80% confluency, hNPCs were passaged, 70,000 cells per well with at least 4 replicates per cell line were seeded on Geltrex coated XF96 cell culture microplates and incubated for 48 h with daily medium changes.

DAns were passaged on day 9 of differentiation according to the differentiation protocol, cell number was determined and 5000 cells per well with at least 4 replicates per cell line were seeded on 15 μg/mL poly-L-ornithine and 10 μg/mL laminin coated XF96 cell culture microplates. DAns were matured for at least 30 days at 37 °C, 7% CO2, 21% O2 with medium changes every 2–5 days.

Astrocytes were cultured on 15 μg/mL poly-L-ornithine and 10 μg/mL laminin coating and in astrocyte differentiation medium at 37 °C, 7% $CO_2$, 21% $O_2$. At 70% confluency, astrocytes were passaged, cell number was determined and 50,000 cells per well with at least 4 replicates per cell line were seeded in astrocyte differentiation medium containing 10 μM ROCK inhibitor Y27632 and on 15 μg/mL poly-L-ornithine and 10 μg/mL laminin coated XF96 cell culture microplates and incubated for 6 days with daily medium changes.

The respiratory analysis was performed using an XF96 Analyzer (Seahorse Bioscience) and XF Assay medium (Seahorse Bioscience, 103334-100) supplemented with 5 mM pyruvate (Sigma, P5280-25G). Prior to measurements, cells were washed once with XF Assay medium and at least 4 replicates per patient were incubated with XF Assay medium containing pyruvate for 1 h at 37 °C, 0% $CO_2$, 21% $O_2$. XFe96 Sensor Cartridges (Seahorse Bioscience, 102416100) were hydrated with 200 μl Calibrant (Seahorse Bioscience, 100840000) per well overnight and ports were loaded with (A) 10 μg/mL oligomycin (Sigma, O4876-5MG), (B) 5 μM FCCP (Sigma, C2920-10MG), (C) 50 μM rotenone (Sigma, R8875-1G) and 20 μM antimycin-A (Sigma, A8674-50MG) and (D) 1 M 2-deoxyglucose (Sigma, D8375-25G). After equilibration, respiratory analysis was performed with a total of 4 basal measuring points (mix for 1 min, time delay for 2 min and measure for 3 min). Subsequent to port injections, respiratory analysis was performed with 3 measuring points. Data were visualized and exported using Wave 2.6.1 (Agilent Technologies). Data were normalized to DNA content analyzed on an equally seeded and grown plate (fibroblasts, hiPSCs, hNPCs) or the same plate (DAns, astrocytes) using the Quant-iT PicoGreen dsDNA Assay Kit (Thermo Fisher Scientific, P11496). The copy plate (medium aspirated) or the same plate was stored at −20 °C immediately after the Seahorse run was performed and thawed on ice for 30 min prior to analysis. For the copy plate, cells were lysed in 60 μl RIPA buffer (50 mM Tris-HCL (Sigma, T3253), 150 mM NaCl (Merck, 106404), 1% Triton X-100, 0.5% sodium deoxycholate (Sigma, D6750), 0.1% SDS (Sigma, L3771), 3 mM EDTA (Sigma, EDS-1KG)) per well. For the same plate, 10 μl Proteinase K (20 mg/ml; AppliChem, A3830) was added per well and cells were lysed at 37 °C for 1 h. The assay was performed according to the manufacturer's instructions. DNA concentrations were calculated with a linear regression curve using Lambda DNA standards. For statistical analysis of basal and maximal mitochondrial respiration as well as proton leak the mean values per patient of measuring points 1–4, 8–10 and 5–7, respectively, were used. The last two measuring points were used to subtract non-mitochondrial respiration.

## Analysis of complex I activity

Complex I enzyme activity was measured using the complex I enzymatic activity microplate assay kit (Abcam, ab109721) according to the manufacturer's instructions. Enzymatic kinetics were measured on a SpectraMax M5 (Molecular Devices) for 45 min. Values were normalized to total protein concentrations and average Ctrl levels.

## Isolation of RNA, cDNA synthesis and quantitative real-time PCR

Gene expression was analyzed by RT-qPCR. Total RNA was extracted using RNeasy Plus Mini Kit (Qiagen, 74134) according to the manufacturer's instructions and reverse transcribed to cDNA using the SuperScript VILO cDNA Synthesis Kit (Thermo Fisher Scientific, 11754050). For RT-qPCR 25 ng (278 ng/μl) cDNA was quantitatively amplified on a QuantStudio 7 Flex (Thermo Fisher Scientific) using TaqMan universal PCR MM no Ung (Thermo Fisher Scientific, 4324018) and gene specific TaqMan primers (Thermo Fisher Scientific, 4331182): ACTB (Hs99999903_m1); HPRT1 (Hs99999909_m1); GAPDH (Hs99999905_m1); GLI1 (Hs00171790_m1); GLI2 (Hs01119974_m1); GLI3 (Hs00609233_m1); FOXA2 (Hs05036278_s1); PITX3 (Hs010139-35_g1); BBS5 (Hs00537098_m1); GET4 (Hs00944514_m1); LINGO2 (Hs01102041_s1); SLC1A2 (Hs01102423_m1); SRCAP (Hs00198472_m1); PTCH1 (Hs00181117_m1); PTCH2 (Hs00184804_m1); SHOX2 (Hs00243-203_m1); GBP1 (Hs00977005_m1). Comparative $C_t$ method was used to analyze differences in gene expression, values were normalized to ACTB or BestKeeper[81] calculated from ACTB, GAPDH, HPRT1 expression levels.

## Cytoplasmic/nuclear fraction isolation and immunoblotting

For protein extraction hNPCs or DAns ($5 \times 10^6$) were washed with ice cold PBS and collected by scraping in 100 μl ice cold Hypotonic buffer (20 mM Tris-HCl pH 7.4, 10 mM NaCl, 3 mM MgCl2 (Sigma, 208337-100 G)) containing Protease Inhibitor Cocktail (Roche Diagnostics, 11836170001). Cells were lysed on ice for 15 min, 5 μl 10% $NP_40$ (Sigma, 11754599001) was added and the homogenate was centrifuged at 1,000 g, 4 °C for 10 min. The supernatant contained the cytoplasmic fraction. The nuclear pellet was washed with 500 μl Hypotonic buffer, centrifuged at 1000 g, 4 °C for 10 min and resuspended in 20 μl Cell Extraction Buffer (Thermo Fisher Scientific, FNN0011) containing a Protease Inhibitor Cocktail. Nuclei were lysed on ice for 50 min with vortexing at 10 min intervals. Debris were removed by centrifugation at 14,000 g, 4 °C for 30 min. The supernatant contained the nuclear fraction. Protein concentration was quantified using Pierce BCA Protein Assay Kit (Thermo Fisher Scientific, 23225) according to the manufacturer's instructions. 10 μg protein extract was diluted in RIPA and NuPAGE (Novex, NP0007) and incubated for 5 min at 95 °C. Protein extracts from 5 Ctrl and 7 sPD hNPC lines and Protein Marker VI (AppliChem, A8889) were separated on a Criterion XT Bis-Tris Gel, 4-12% (Bio-Rad, 3450124) using Tris/Glycine Buffer (Bio-Rad, 1610771) and a Criterion Vertical Electrophoresis Cell (Bio-Rad) at 120 V for 70 min. Proteins were blotted on Immobilon – P Membranes (Millipore, IPVH00010) using XT MOPS buffer (Bio-Rad, 1610788) and a Criterion Blotter (Bio-Rad) at 20 V, 4 °C overnight. Membranes were blocked for 1 h with TBS containing 0.01 % Tween (TBST) and 5 % milk. Primary antibodies anti-GLI3 (AF3690, R & D; 1:100), anti-ACTB (ABO145-200, OriGene; 1:2,000), anti-PCNA (ab29, Abcam; 1 μg/ml) or H1-0 (ab11079, Abcam; 1 μg/ml) were incubated in blocking buffer overnight at 4 °C. Membrane was washed with TBST and secondary antibodies rabbit-anti-mouse IgG peroxidase (GTX213112-01, GeneTex; 1:10,000) and rabbit-anti-goat IgG peroxidase (305-035-003, Dianova; 1:10,000) were incubated in blocking buffer at room temperature for 2 h, respectively. Subsequently, the membrane was washed with TBST and incubated with ECL substrate (GE Healthcare, RPN2106OL/AF) for 1 min. Protein bands were visualized using a ChemiDoc Imager

(Bio-Rad) and quantified using Image Lab 6.1 (Bio-Rad). Uncropped blots are supplied in the Source Data file.

### EdU proliferation assay

Cells were cultured on Geltrex coated 96 well plates for at least 72 h. hNPCs were incubated with 10 μM 5-ethynyl-2′-deoxyuridine (EdU) for 2 h at 37 °C, 7% $CO_2$, 21% $O_2$. Cells were fixed, EdU was detected and DNA was stained according to the Click-iT Plus EdU Alexa Fluor 488 Imaging Kit (Thermo Fisher Scientific, C10420) manual. Images were acquired using a Cellinsight NXT platform (Thermo Fisher Scientific) with a 20 × 0.4 NA objective (field size of 45,441 by 454.41 μm) and analyzed using HCS Studio 2.0 (Thermo Fisher Scientific). Nuclear DNA fluorescence intensity (Hoechst dye) was assessed in channel 1 and a nuclear mask was created using the image analysis segmentation algorithm to identify viable cells as valid objects according to their object area and shape. The nuclear mask was used to quantify EdU fluorescence intensity of valid objects in channel 2 with a fixed exposure time. At least 500 valid objects per well and cell line were analyzed. EdU fluorescence intensity was normalized to the nuclear area and mean Ctrl fluorescence intensity, an average fluorescence intensity per cell line was reported.

### Neuron counting and neurite quantification

Neurite outgrowth analysis was performed using a Cellinsight NXT platform with a 20 × 0.4 NA objective (field size of 454.41 by 454.41 μm) and analyzed using HCS Studio 2.0 (Neuronal Profiling Bioapplication). Twenty-five imaging fields were collected per well. Nuclei were identified by DAPI staining, while cell somas and neurites of dopaminergic neurons were identified by TH staining. Cells were classified as neurons if they had a DAPI and RBFOX3 (NEUN) -positive nucleus as well as a TH positive soma. Only neurites longer than 10 μm were included in the analysis.

### DNA methylation analysis

Genomic DNA from hNPC lines was extracted using the QIAamp DNA Mini Kit (Qiagen, 51304) according to the manufacturer's instructions for cultured cells. DNA concentration was measured using Quant-iT™ PicoGreen according to the Infinium HTS Assay Protocol Guide (Illumina). DNA methylation was quantified using the Methylated DNA Quantification Kit (Colorimetric) (Abcam, ab117128) according to the manufacturer's instructions. An amount of 100 ng sample genomic DNA was used per reaction. Absorbance was measured on a SpectraMax M5 at 450 nm after 10 min.

### SNP detection and CNV analysis

Genomic DNA from hNPCs was extracted using the QIAamp DNA Mini Kit according to the manufacturer's instructions for cultured cells. DNA concentration was measured using Quant-iT™ PicoGreen according to the Infinium HTS Assay Protocol Guide (Illumina). SNPs from hNPC lines (200 ng DNA per sample) were detected in duplicates using an Infinium CoreExome-24 v1.3 Kit (20024662, Illumina) and an iScan system (Illumina) according to the Infinium HTS Assay Protocol Guide. The Infinium HTS Assay was performed in the genome analysis center of Helmholtz Zentrum München. Clustering, quality control, and SNP calling was done using GenomeStudio 2.0 (Illumina) as described by[82]. The log2 R ratios (LRR) and B-allele frequencies (BAF) were used to identify copy number variations (CNVs) on autosomes for each cell line using cnvPartition v3.2.1 (Illumina) with regards to aberrations minimally sizing 100 kb. cnvPartition v3.2.1 was run with default settings, including a confidence threshold of 50. All SNPs were converted to the GRCh38 forward strand using a script developed by Robertson and Wrayner [https://www.well.ox.ac.uk/~wrayner/strand/] (downloaded on 12.11.2019). For multiplexed scRNA-seq, SNP files were converted to vcf files using PLINK 2.00 alpha[83]. A custom-made perl script was used to extract exonic SNPs using exon boundaries obtained

from BioMart ([https://www.ensembl.org/info/data/biomart/index.html#biomartdoc]) (downloaded on 12.11.2019).

### Single-cell transcriptome library preparation and sequencing

For multiplexed scRNA-seq analysis 5 Ctrl and 7 sPD lines, as well as one *PINK1* ko with a isogenic Ctrl were used. hNPCs were maintained on Geltrex coated plates containing neural precursor maintenance medium at 37 °C, 7% $CO_2$, 21% $O_2$. At 80% confluency, hNPCs were dissociated using Accutase for 10 min at 37 °C, chopped with a 1000 μl pipette tip and diluted in 5 ml PBS containing 10% BSA (Sigma, SRE0036-250ML). hNPCs were harvested at 200 g for 5 min and resuspended in 1 ml PBS containing 10% BSA. The hNPC solution was filtered through a 35 μm cell strainer (Corning, 352235) and counted using a Neubauer improved cell counting chamber (Carl Roth, PK361). 978 hNPCs from each ForIPS line, 1956 hNPCs from the *PINK1* ko line or isogenic Ctrl were mixed and used for multiplexed droplet single-cell library production. In total 6 libraries with 13,692 hNPCs each were generated in the 10X Genomics Chromium controller according to the manufacturer's instructions and using the Chromium Single Cell 3′ Library & Gel Bead Kit v2 (PN-120237, 10xGenomics) and Chromium i7 Multiplex Kit (PN-120262, 10xGenomics). Libraries were pooled and sequenced on a HiSeq4000 (Illumina) according to the Chromium Single Cell v.2 specifications and with an average read depth of 90,000 aligned reads per cell. Sequencing was performed in the genome analysis center of Helmholtz Zentrum München.

### Sequencing data processing

Sequencing data was processed using Cell Ranger version 2.1.1. Read files were aligned against hg38 from Ensembl release 94 using default parameters. Spliced and unspliced counts for RNA velocity analysis were called using velocyto version 0.17.7 and samtools version 1.7.

We used Demuxlet[15] (retrieved 17th July 2018) and the obtained SNP data to demultiplex the pooled sequencing data. The SNP array data was reformatted to fit the chromosome ordering in the Cell Ranger output using vcf tools version 0.1.15. Demuxlet was run separately per sequencing batch (lane) with potential mixture proportion parameter alpha set to 0 and 0.5 and otherwise default parameters. As demuxlet can both assign cells to the donor they came from and identify doublets, we used this algorithm to also filter out doublets. Thus, only cells called as singlets by demuxlet were used for further processing. The sPD clones and their respective Ctrls could be separated by SNP data, while the *PINK1* ko clone was run on different sequencing lanes as its isogenic Ctrl.

### Single-cell transcriptome analysis

Single cell gene expression data was processed according to a recently published best practices workflow[84] in a python version 3.6.7 and R version 3.5.1 environment. Data was loaded into Scanpy[85] version 1.4.3 commit 0075c62. After sequencing and filtering out empty droplets by Cell Ranger, we were left with 30,626 cells.

### Quality control and pre-processing

After considering the joint distribution of count depth, the number of genes expressed, and MT read fraction per sample, cells with more than 45,000 counts, with fewer than 1000 genes expressed, and with 15% or more reads aligned to mitochondrial genes were filtered out. Furthermore, genes that were measured in fewer than 20 cells were also removed from the dataset. This left a dataset of 30,557 cells and 24,920 genes. Normalization was performed by the pooling method implemented in the computeSumFactors() function in scran version 1.10.2. Normalized expression values were log+1-transformed.

To remove batch effects between donor samples in the dataset we used the python implementation of the matching mutual nearest

neighbors algorithm[86], mnnpy (version 0.1.9.5; retrieved from [https://github.com/chriscainx/mnnpy]) with donor as batch covariate. MNNPY was run on a subset of highly variable genes (HVGs) that were shared by all batches. We selected 6000 HVGs per batch (donor) as implemented by the *highly_variable_genes* method (with *flavor = "cell_ranger"*) in Scanpy. Here, HVGs are selected by binning genes by mean expression and choosing the genes with the highest coefficient of variation per bin. The intersection of the set of HVGs per batch left 516 genes, which we used to find mutual nearest neighbors between batches in the dataset. Upon MNN data integration, we selected 4000 HVGs on the integrated dataset according to previously published best practices[84].

Low dimensional representations of the data were obtained by taking the top 50 principal components (PCs) and calculating a k-nearest neighbor (KNN; k = 15) graph using Euclidean distance on the PC space as implemented in scanpy. The KNN graph was used as the basis on which to run UMAP[87] v0.3.9 for visualization.

### Cell cycle scoring and regression
Cell cycle phases were called for each cell using Scanpy's *score_genes_cell_cycle* function with the gene list from[88]. Scoring is performed by taking the mean expression of the gene sets for S and G2/M phases and subtracting a background gene expression score to obtain an S and G2/M phase score per cell. The background expression score is calculated by taking the mean expression of a random sample of 50 genes which are selected to have similar mean expression as the original gene set. Cells are assigned to S phase, G2/M phase, or G1 phase based on the relative S and G2M scores.

For trajectory and RNA velocity analysis, we used a version of the data with the cell cycle regressed out. For this we used Scanpy's regress_out function to jointly regress out the S G2/M phase scores on the MNN-corrected expression matrix.

### Clustering, sub-clustering, marker gene detection, and cluster annotation
We performed graph-based clustering on the computed KNN graph using the python implementation (version 0.6.1) of the Louvain algorithm[89] in Scanpy. As a first basis, we performed a coarse Louvain clustering at a resolution of 0.2. For each cluster, marker genes were determined by applying Welch's *t*-test (as implemented in Scanpy's *rank_genes_groups* function with default parameters) between the cells in the cluster and all other cells. Differential expression testing for marker gene detection was performed on the log-normalized, non-mnn-corrected expression values as recommended by published best practices[84]. We annotated clusters by gathering evidence from four strategies: overlaps of data-derived marker genes (Supplementary Data 4) with marker genes from the literature, literature-derived marker gene expression scores in our data, Gene Ontology[90] biological process term enrichment tests on data-derived marker genes via GProfiler (python-gprofiler version 1.2.0)[91], and cell cycle phases of cluster cells. Literature derived marker gene sets were obtained from[16–23]. To gain a higher-resolution view on the cellular populations in the data, we sub-clustered larger clusters at a resolution of 0.2 (Clusters 0 and 4) using the restrict_to parameter in Scanpy's Louvain implementation. Sub-clusters were annotated in the way described above.

As clustering and sub-clustering frequently revealed a cluster structure that correlated with cell cycle phases, we performed a second Louvain clustering on the data after regressing out cell cycle effects (noCC data). Louvain clustering was performed at a resolution of 0.2 and broadly overlapped with results from the above clustering on original data. The NCSC cluster was sub-clustered at a resolution of 0.3 and 0.1 iteratively to isolate the apoptotic sub-population, which was identified by data-derived marker genes (Supplementary Data 4), and low mitochondrial read fractions. Cluster annotations from the original and the noCC data were harmonized using the original clustering as a basis and adding annotations from the noCC data when the original cluster structure was determined by cell cycle phases. For

example, the cluster Neuronal Stem Cells (G2M-phase) was divided into NSC1 (G2M-phase) and NSC2 (G2M-phase) in this way. Furthermore, the apoptotic sub-cluster labels from the noCC data annotation were kept.

### Compositional data analysis
To test whether there are compositional changes between Ctrl and sPD conditions we utilized a Bayesian approach implemented with scCODA version 0.1.7[92]. We used the following parameters: *formula = " Condition"* and *reference_cell_type = "automatic"*. Briefly, scCODA models cell-type counts with a hierarchical Dirichlet-Multinomial distribution with parameters inferred via Hamiltonian Monte Carlo (HMC) sampling and detects statistically credible changes in cell-type composition through the covariate's posterior inclusion probability, where the covariates' effects are determined by the spike-and-slab prior.

### RNA velocity and trajectory analysis
RNA velocity analysis was performed using scVelo[93] version 0.1.24 on commit e45a65a. Unspliced and spliced read data contained 71% spliced counts, 22% unspliced counts, and 7% ambiguous reads. Merging of unspliced and spliced counts with our processed expression data resulted in a dataset containing 24,956 of the initial 30,557 cells. Unspliced and spliced count data were pre-processed by filtering out genes with fewer than 20 spliced counts and 10 unspliced counts. The resulting data was normalized (total count normalization) and log-transformed via scVelo's *filter_and_normalize* function and smoothed via the *moments* method. To select only genes that were fit to a minimal degree of accuracy by the scVelo model we used a threshold of 0.001 on the per-gene log-likelihood to determine our velocity gene set. This thresholding left 2,439 genes to fit the full dynamical model in scVelo.

RNA velocity trajectories were determined by finding the root and end states shown by the velocity transition matrix. For this computation, we subsetted the data to the main neuronal cell population of interest (clusters *NSC1a*, *NSC1b*, *NSC2a*, *NSC2b*, and *immature neurons*) and determined the terminal states via sCVelo's *terminal_states* function. This function computes root and endpoint probabilities via a spectral decomposition of the velocity-inferred transition matrix. Specifically, the left eigenvectors of this matrix that correspond to a steady-state eigenvector of 1 determine the endpoint probabilities, while the root cell probabilities are obtained by using the transpose of the transition matrix for the same calculation.

We obtained the root cell population by multiplying the root cell probabilities with the KNN graph connectivities to smooth the transitions and then selected cells with a score above 0.4. The resulting root cells were clustered by the *louvain* function at a resolution of 0.3. The dominant root cell cluster (>50% of cells) was used to represent the root cell population and ignore outliers. Root cell sub-clusters with particular fates were identified by sub-clustering the main cluster at a resolution of 0.5 and assigning NSC1 and NSC2 fates based on velocity arrows. Endpoint clusters were obtained in a similar manner, using a threshold of 0.6, without prior KNN-smoothing. After Louvain clustering at a resolution of 0.3, we selected the two clusters that best correspond to the low dimensional embedding for further investigation.

### Transcriptome analysis of *PINK1* ko clones
As quality control (QC) metric distributions were similar between samples, we performed a joint QC across all *PINK1* ko clones. Cells with more than 39,000 counts, with fewer than 1000 genes expressed, and with 15% or more reads aligned to mitochondrial genes were filtered out. Furthermore, genes that were measured in fewer than 20 cells were also removed from the dataset. This resulted in a dataset of 10,240 cells and 20,623 genes, with 3671 *PINK1* ko cells. Normalization was performed jointly across all clones by the pooling method

implemented in the computeSumFactors() function in scran version 1.10.2. Normalized expression values were log+1-transformed.

Cell cycle scoring was performed on a gene set[88] as described above. For the *PINK1* ko cell data, we first regressed out the cell cycle effect and then integrated the two conditions using mnnpy as described for the sPD clones. Here, the intersection of per-batch HVGs between batches left 5,328 HVGs on which the mnn data integration could be performed. Low dimensional embeddings for the *PINK1* ko cell data were computed via a KNN graph (k = 15) calculated using Euclidean distances on the PC-reduced expression space subsetted 4000 HVGs selected by Scanpy's *highly_variable_genes* function as described above.

Clustering by Louvain community detection was performed at a resolution of 0.3 for the *PINK1* ko dataset. Marker genes were obtained by applying Scanpy's Welch's t-test implemented in the *rank_genes_groups* function in the same way as for the sPD clone data. Clusters were annotated by comparison of data-derived marker genes with the previously described literature curated marker gene list.

### Differential expression analysis

Differential expression analysis per cell identity cluster was performed via the Diffxpy package (https://github.com/theislab/diffxpy; version 0.6.1; dev branch). Diffxpy fits a negative binomial model to the raw count data and allows for adding covariates and constraints into the model. In the sPD clone dataset, we fit the model:

$$Y \sim 1 + condition + donor + size\_factors, \qquad (1)$$

for each annotated cluster. Here, *Y* represents the raw gene expression counts, *1* denotes the intercept, *condition* is the disease status (Parkinson vs control), *donor* is the patient from which the cell was obtained, and *size_factors* denote the scran pooling size factors. As the *donor* and *condition* covariates are perfectly confounded, we added the constraint that donor effects sum to 0 per condition (input as '*Donor:condition*' in Diffxpy) to make the model identifiable. As we only have a single sample per condition in the *PINK1* ko cell data, we instead fit the simplified model:

$$Y \sim 1 + condition + size\_factors, \qquad (2)$$

in these dataset. This simplified model did not require additional constraints.

The differential test was performed via a Wald test over the *condition* covariate per gene for all genes expressed in at least 50 cells in the tested cluster. Multiple testing correction was performed via the Benjamini-Hochberg method[94].

### Enrichment and pathway analysis

Analysis were performed in R version 4.0.5 and RStudio Desktop 1.2.1335. DEGs were investigated for enrichment in the curated pathway categories Cell Process and Signal Processing of the Pathway Studio software (Elsevier) version 12.4.0.5 by using a one-sided Fisher's exact test. Subsequently, *p*-value correction for multiple testing with the fdrtool program[95] was performed. The fdrtool function (*statistic* = "p value", *cutoff.method* = "fndr") within the fdrtool v1.2.16 package in R was used to calculate *q*-values. In case of small censored sample size for null model estimation (N.cens<10) *cutoff.method* = "ptc0" was selected. We extended the literature-derived network of the intraflagellar transport BBSome interaction pathway in the category Cell Process from the Pathway Studio by manual literature searches. To further expand the BBSome interaction pathway a list of cilia associated genes were downloaded from the UniProtKB database (www.uniprot.org/keywords/ with the keywords Cilium biogenesis/degradation (KW-0970) and Cilium (KW-0969)) and compared to DEGs of cell clusters.

DEGs were investigated for enrichment in WikiPathway (WP) and KEGG pathways using the *enrichKEGG* or *enrichWP* function of the R package clusterProfiler v4.2.2[96] (one-sided hypergeometric test). *p*-values were adjusted for multiple testing by Benjamini and Hochberg[94]. Enrichment maps were generated using the *treeplot* function of clusterProfiler v4.2.2.

DEGs of the root population were analyzed for enriched gene sets of the category curated canonical pathways by using the *enricher* function of the R package clusterProfiler v4.2.2[96] and the C2.CP collection of the Molecular Signatures Database v7.5.1[97]. p-values were adjusted for multiple testing by Benjamini and Hochberg[94]. Enrichment maps were generated using the *emapplot* function of cluster-Profiler v4.2.2.

### Heatmap and volcano plot

Analysis were performed in R version 4.0.5 and RStudio Desktop 1.2.1335. We represented log2 transformed fold changes of DEGs from each cell cluster by a heatmap. This heatmap was generated by using the *heatmap.2* function within the gplots v3.1.1 package of R. Agglomerative hierarchical clustering by the *hclust* function (method = "complete") was applied to group cell clusters (rows). Log2 transformed fold changes of DEGs (columns) were scaled and represented as z-score.

Based on the complete set of DEGs per cluster volcano plots were produced by plotting log2(fold change) versus −log10(q-value) on the x and y-axis, respectively. The *q*-value represents the differential expression p-value adjusted for multiple testing by Benjamini and Hochberg[94]. Volcano plots were generated using the *EnhancedVolcano* function of the R package EnhancedVolcano v1.12.0 with pCutoff = 0.05 and FCcutoff=0.26.

### Transcription factor target gene prediction

In silico prediction of GLI3 transcription factor binding sites in all human promoter sequences was performed by using the MatInspector program v8.4.1 (Genomatix)[98]. The corresponding position weight matrices V$GLI3.01 and V$GLI3.02 were applied to promoter analysis according to the Matrix Family Library Version 11.1 (February 2019). We used the maximum core similarity of 1.0 to exactly match the highest conserved bases of a matrix in the sequence. The threshold for the matrix similarity was set to 0.9 to minimize false positives.

### Hemiparkinsonian animal models

For the analysis of neuronal PC in 6-hydroxydopamine-HCl (6-OHDA-HCl) lesion mice, three 3-4-month-old homozygous B6.Cg-Tg(Gfap-cre)77.6Mvs/2 J (Jackson Laboratories, 024098) male littermates (breeding of heterozygous parents) were analyzed.

B6.Cg-Tg(Gfap-cre)77.6Mvs/2 J mice received a unilateral injection of 6-OHDA-HCl (Sigma-Aldrich, H4381) into the left medial forebrain bundle. Mice were anesthetized by intraperitoneal injection of Medetomidine (0.5 mg/kg body weight), Midazolam (5 mg/kg body weight) and Fentanyl (0.05 mg/kg body weight). Mice received preemptive injections of Metamizol (200 mg/kg; s.c.) and a local subcutaneous injection of 2% Lidocain. The animals were positioned in a stereotacic frame containing an integrated warming base (51730D, Stoelting). 6-OHDA-HCl (H4381, Sigma-Aldrich, 2 µg/µl) was dissolved in physiological saline containing 0.2% ascorbic acid (Sigma-Aldrich, A4403) and was injected into the left medial forebrain bundle at the stereotactic coordinates: −1.2 mm anteroposterior, +1.0 mm mediolateral, −4.9 mm dorsoventral, relative to the bregma. A volume of 1.5 µl was injected using a microsyringe at a rate of 0.2 µl/min. The microsyringe was removed slowly 3 min after the injection to allow the toxin to diffuse into the tissue. Anesthesia was antagonized by subcutaneous injection of Atipamezol (2.5 mg/kg) and Flumazenil (0.5 mg/kg). Mice were sacrificed 14 days after 6-OHDA injections and brains were isolated and post fixated for at least 24 h in 4% PFA.

### Pink1 ko animal models

For the analysis of neuronal PC in an animal model for genetic PD, five 12.5 ± 1.5 months old male homozygous *Pink1-/-* mice and five male wild type littermate *Pink1 +/+* Ctrl animals were analyzed. *Pink1* KO mice were generated and characterized in our lab as previously described[53] and have been bred on a C57BL/6 J genetic background in the animal facility at the Helmholtz Zentrum München. In brief, in this mouse line, *Pink1* has been knocked out by excision of floxed exons 2 and 3 of the *Pink1* gene. Mice were kept under specific pathogen-free (SPF) conditions on a 12/12-hour dark-light cycle in a temperature (22 – 24 °C) and humidity (50–60%) controlled environment and were provided food and water *ad libitum*. All protocols and procedures were conducted with the approval for the ethical treatment of animals by the responsible animal welfare authority of the Regierung von Oberbayern (Government of Upper Bavaria). Mice were sacrificed with $CO_2$ inhalation at the age of 12.5 months±1.5 months and perfused transcardially with 0.1 M phosphate buffered saline (PBS) (pH 7.4) followed by 4% paraformaldehyde (PFA) (Sigma, P6148).

### Primary cilia analysis in animal models

Fixed brains were excised from the skull and post-fixed in 4% PFA overnight at 4 °C and subsequently transferred to a 30 % (w/v) sucrose (Sigma, S0389) cryoprotectant solution until fully saturated. Brains were then sectioned at −20 °C into 40 μm thick horizontal sections using a freezing cryostat and collected free-floating in a cryoprotectant solution containing 25% ethylene glycol (Roth, 2441.2) and 25% glycerine (Sigma, G5516) in 0.1 M PBS. Sections were stored in cryoprotectant solution at −20 °C until further processing. Immunofluorescent double labeling was performed according to standard protocols. In brief, free-floating brain sections (1 in 6 series) were rinsed in 0.1 M PBS overnight at 4 °C on a platform shaker (100 rpm) to ensure that any residual cryoprotectant solution was washed off. Sections were incubated in blocking buffer containing 0.3% Triton-X-100 in 0.1 M PBS with 2% fetal calf serum (FCS) for 1 h to decrease unspecific background staining. After blocking, brain tissue was incubated in a primary antibody dilution in blocking buffer overnight at 4 °C on a platform shaker. The following primary antibodies were used: rabbit anti-Adcy3 (PA5-35382, Invitrogen; 1:500) in combination with mouse anti-Rbfox3 (MAB377, Millipore; 1:200) or goat anti-Chat (AB144P, Millipore; 1:200). Following primary antibody incubation, brain sections were washed three times in 0.1 M PBS for 10 min each at room temperature on a platform shaker. Brain tissue was then incubated in secondary antibody dilution in 0.3 % Triton-X-100 in 0.1 M PBS for 2 h at room temperature on the platform shaker. Following secondary antibodies were used: Alexa Fluor 488-conjugated donkey anti-rabbit IgG (A21206, Invitrogen; 1:500), Alexa 594-conjugated donkey-anti-mouse IgG (A21203, Thermo Fisher Scientific; 1:500) and Alexa Fluor 594-conjugated donkey anti-goat IgG (A11058, Invitrogen; 1:500). Sections were washed with 0.1 M PBS three times for 10 min each before they were mounted on slides and coverslipped using Aqua-PolyMount. PC of striatal cholinergic neurons (double-stained for Chat and Adcy3), as well as cortical and striatal neurons (double-stained for Rbfox3 and Adcy3) were selected for cilia length quantification using Neurolucida version 2019.2.1 with a Zeiss Axioplan 2 microscope equipped with a 100 X immersion objective. The analysis was performed blinded and an average of 60 – 100 PC were analyzed per mouse from random fields. Additionally, striatal PC stained for Adcy3 were quantified in a three dimensional space (115 μm x 87 μm x 40 μm) with an average of >210 PC per mouse from random fields.

To determine the lesion site in brain slides from 6-OHDA animals, mounted brain sections were stained for TH after PC analysis. The samples were rehydrated in 0.1 M PBS overnight at room temperature, which allowed us to carefully remove the cover slip. Following rehydration, sections were washed once in sodium citrate buffer (0,01 M sodium citrate pH 6.0 (Sigma-Aldrich, C8532)) for 3 min at room temperature and heated (microwave, 900 watt) in a pressure cooker filled with sodium citrate buffer for 30 min. After sections were cool to room temperature, they were washed once in aqua destillata and once in 0.1 M PBS for 5 min each at room temperature. Endogenous peroxidases were suppressed using 0.1 % $H_2O_2$ in PBS for 5 min at room temperature. Sections were washed twice in 0.1 M PBS for 5 min each at room temperature and incubated in blocking buffer (0.3 % Triton-X-100, 1% FCS, 5% milk (170-6404, Biorad), 1% BSA (A3059, Sigma) in in 0.1 M PBS) for 1 h to decrease unspecific background staining. After blocking, brain tissue was incubated in a primary antibody dilution in blocking buffer overnight at 4 °C in a humidity chamber. The following primary antibody was used: rabbit anti-Th (P40101, Pel-Freeze; 1:200). Following primary antibody incubation, brain sections were washed three times in 0.1 M PBS for 10 min each at room temperature. Brain tissue was then incubated in secondary antibody dilution in 0.3% Triton-X-100 in 0.1 M PBS for 1 h at room temperature in a humidity chamber. Following secondary antibody was used: Biotin-SP-conjugated goat anti-rabbit IgG (111-065-003, Dianova; 1:250). Sections were washed three times in 0.1 M PBS for 5 min each and incubated in an avidin-biotin peroxidase solution (PK-6100, Vector Laboratories) for 30 min at room temperature in a humidity chamber. Following one wash step in 0.1 M Tris-HCl pH 7.4, sections were incubated in a 3,3'-Diaminobenzidine (DAB) chromogen solution (0.1 M Tris-HCl pH 7,4, 1% DAB (D8001, Sigma), 0.02% $H_2O_2$) for up to 10 min at room temperature in a humidity chamber. Sections were washed with 0.1 M PBS three times for 10 min each before they were cover-slipped using Aqua-PolyMount.

### Primary cilia analysis in postmortem patients

Tissue from the anterior striatum and the occipital cortex was obtained from the NBM (www.en.neuropathologie.med.uni-muenchen.de/neurobiobank/neurobiobank/index.html). Brain tissue was fixed in 4 % formalin for 5-120 days (at room temperature), thereafter embedded in paraffin and sliced into 5 μm sections.

Immunofluorescent double labeling of paraffin embedded sections was performed according to standard protocols. In brief, brain sections were deparaffinized two times in xylol for 15 min each at room temperature and rehydrated in an ethanol dilution series (100%, 96 %, 80%, 70 %, 60 %, 30%, aqua destillata) for 3 min each if not indicated otherwise (100% ethanol, two times 3 min; aqua destillata two times 5 min). Following rehydration, sections were washed once in sodium citrate buffer (0,01 M sodium citrate pH 6.0) for 3 min at room temperature and heated (microwave, 900 watt) in a pressure cooker filled with sodium citrate buffer for 30 min. After sections were cool to room temperature, they were washed once in aqua destillata and once in 0.1 M PBS for 5 min each at room temperature. Sections were incubated in blocking buffer (0.3% Triton-X-100, 1% FCS, 5% milk (170-6404, Biorad), 1% BSA (A3059, Sigma) in in 0.1 M PBS) for 1 h to decrease unspecific background staining. After blocking, brain tissue was incubated in a primary antibody dilution in blocking buffer overnight at 4 °C in a humidity chamber. The following primary antibody was used: rabbit anti-ARL13B (17711-1-AP, Proteintech; 1:500). Following primary antibody incubation, brain sections were washed three times in 0.1 M PBS for 10 min each at room temperature. Brain tissue was then incubated in secondary antibody dilution in 0.3% Triton-X-100 in 0.1 M PBS for 1 h at room temperature in a humidity chamber. Following secondary antibody was used: Alexa Fluor 488-conjugated donkey anti-rabbit IgG (A21206, Invitrogen; 1:500). Sections were washed with 0.1 M PBS three times for 10 min each before they were incubated in DAPI solution (0.1 μg/ml in PBS) for 10 min. Following DAPI staining, brain sections were washed three times in 0.1 M PBS for 10 min each at room temperature and coverslipped using Aqua-PolyMount. DAPI staining was used to distinguish between neurons and non-neuronal cells. PC of cortical and striatal neurons (a large, less intense nuclei in close proximity to an ARL13B stained PC) were

selected for cilia length quantification using Neurolucida version 2019.2.1with a Zeiss Axioplan 2 microscope equipped with a 100X immersion objective. The analysis was performed blinded and an average of 100 PC were analyzed per patient from random fields.

## Statistics and reproducibility

No statistical methods were used to predetermine the sample size. Every analysis was performed thrice using independently collected material from 5 Ctrl and 7 sPD patient derived cell clones e.g. from three different passages (fibroblasts and hiPSCs) or three independent differentiation approaches (hNPCs, DAns, astrocytes). Each data point represents the average of these three replicates for each individual. If two clones per patient were used for experiments, first the average value of these clones per individual and replicate was calculated. Average values per patients were used for any further statistical analysis performed using GraphPad Prism 6. For two-group comparisons and in case of normal distribution, unpaired, two-tailed *t*-test was applied. In the case of non-Gaussian distribution, two-tailed Mann–Whitney-U test was applied. For comparison of multiple groups, two-way ANOVA with Tukey's Post-hoc test was performed. Data displaying a measurement progression are shown with mean ± standard error of the mean (SEM). Boxplots are displayed from min to max values with all data points shown. If not stated otherwise, R version 4.0.5 and RStudio Desktop 1.2.1335 was used for further analysis. Distribution plots were generated using the *sm.density.compare* function (R package "sm: Smoothing Methods for Nonparametric Regression and Density Estimation" version 2.2-5.6[99]). *P*-values for the distribution were calculated by applying a two-sided two-sample Kolmogorov-Smirnov (ks) test (*ks.test* function in R). Additionally, a linear mixed effects model (lm) was fit using the *lmer* function (R package "lme4" Version 1.1-26), where unique cells were included but nested within donors (formula: *PC Length ~ disease state* + 1 | *disease state:Patients*; REML = FALSE). P values for lm were calculated using the Anova function (R package "car" Version 3.0-10). Venn diagrams were generated using the *draw.pairwise.venn* function (R package "VennDiagram" Version 1.6.20[100]). *P*-values or *q*-values below 0.05 are considered significant. Not-significant differences are not indicated.

## Reporting summary

Further information on research design is available in the Nature Research Reporting Summary linked to this article.

## Data availability

All data produced in this study (Supplementary Data 15) are archived internally. Source Data are provided as a Source Data file which is also deposited in Zenodo as record 6677636 [101]. The scRNA-seq data were deposited in the NCBI Gene Expression Omnibus (GEO) under accession number GEO: GSE176160. For further requests please contact the corresponding author (W.W.). Following databases were used within this study: Pathway Studio Web (Elsevier) Mammal Database Version 12.4.0.5 [https://www.pathwaystudio.com] (downloaded on 03.2022), KEGG pathways [https://www.genome.jp/kegg/pathway.html] (downloaded on 03.2022), WikiPathway pathways [https://www.wikipathways.org/index.php/WikiPathways] (downloaded on 03.2022), C2.CP collection of the Molecular Signatures Database Version 7.5.1 [https://www.gsea-msigdb.org/gsea/msigdb/] (downloaded on 03.2022), BioMart (https://www.ensembl.org/info/data/biomart/index.html#biomartdoc) (downloaded on 11.2019), Ensembl hg38 release 94 [http://oct2018.archive.ensembl.org/Homo_sapiens/Info/Index] (downloaded on 02.2019), Matrix Family Library Version 11.1 [https://www.genomatix.de/online_help/help_gems/mat_lib_111.html] (downloaded on 02.2019). Source data are provided with this paper.

## Code availability

Analysis code is freely available at [https://github.com/theislab/ipsc_ipd_analysis][102] [https://doi.org/10.5281/zenodo.6656506].

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

## Acknowledgements

We thank Bernt Popp and André Reis (University Erlangen) for performing whole exome sequencing in hiPSCs; Markus J. Riemenschneider (University of Regensburg) for providing material; F. Marxreiter (University Erlangen) for providing clinical information; Michaela Farrell, Sonja Plötz, Annerose Kurz-Drexler, Tanja Orschmann, Susanne Badeke, Sebastian Lacher, Ipek Eroglu, Lisa Hornsteiner for excellent technical assistance; Alessandra Moretti (TUM) for discussions and comments; Volker Bergen (Helmholtz Zentrum München) for discussions about root cell analysis; Martin Jastroch (Stockholm University) for discussions about mitochondrial stress tests.

## Author contributions

This study was designed by W.W., F.G., D.M.V.W., F.J.T., H.L., C.S., S.S., coordinated by W.W., F.G., D.M.V.W. and S.S. and supervised by D.M.V.W., F.G. and W.W. J.W. and B.W. provided the sPD hiPSC clones. J.C.F. and T.G. provided and initially characterized the *PINK1* ko hNPCs. R.A. and M.J.Z. analyzed neurite morphology. N.W. and S.S. performed the PC analysis in hNPCs. S.H. and S.S. performed the PC analysis in

human neurons. K.P. and S.S. performed the PC analysis in human astrocytes. S.S. and K.M.N. performed the PC analysis in *Pink1* ko mice. A.B. and S.S. generated the scRNA-seq libraries. S.S. and D.T. analyzed SNPs and CNVs. M.D.L. and C.R.S. performed the computational analysis of the scRNA-seq under supervision of F.J.T. Clusters were annotated by S.S., D.T., M.D.L. D.T. and S.S. performed enrichment and pathway analysis. Human postmortem material was provided and processed by V.R., W.D.J.vdB., A.J.J. Remaining experiments were performed by S.S. W.W., S.S., F.G., D.M.V.W., M.D.L., D.T., J.C.F, S.H., and K.N. wrote the manuscript.

## Funding

This work was supported in part by the Bavarian Ministry of Science and the Arts in the framework of the ForIPS consortium (C.S., W.W., J.W., B.W.), the ForInter consortium (B.W., J.W., F.J.T.), the German Science Foundation Collaborative Research Center (CRC) 870 (to W.W. - K.M.N.), by the German Federal Ministry of Education and Research (BMBF) through the Integrated Network MitoPD (Mitochondrial endophenotypes of Morbus Parkinson), under the auspices of the e:Med Programme (grant 031A430E to W.W. and 031A430A to T.G.), through the Joint Project HIT-Tau (High Throughput Approaches for the Individualized Therapy of Tau-Related Diseases – TP2: Grant 01EK1605C (to W.W. - D.T.), by the AMPro project - 'Aging and Metabolic Programming'-, by the Incubator grant # ZT-I-0007 sparse2big (to F.J.T. – M.D.L.), by the Chan Zuckerberg Initiative DAF (advised fund of Silicon Valley Community Foundation, 182835) (to F.J.T. – M.D.L.), as well as by 'ExNet-0041-Phase2-3 (SyNergy-HMGU)' through the Initiative and Network Fund of the Helmholtz Association (to H.L., F.G., M.D.L., F.J.T. and W.W.). Open Access funding enabled and organized by Projekt DEAL.

## Competing interests

The authors declare no competing interests.

## Additional information

[1]Institute of Developmental Genetics, Helmholtz Zentrum München, Ingolstädter Landstraße 1, 85764 Neuherberg, Germany. [2]Chair of Developmental Genetics, Munich School of Life Sciences Weihenstephan, Technical University of Munich, Alte Akademie 8, 85354 Freising, Germany. [3]Institute of Computational Biology, Helmholtz Zentrum München, Ingolstädter Landstraße 1, 85764 Neuherberg, Germany. [4]Institute of Metabolism and Cell Death, Helmholtz Zentrum München, Ingolstädter Landstraße 1, 85764 Neuherberg, Germany. [5]Institute of Diabetes and Regeneration Research, Helmholtz Zentrum München, Ingolstädter Landstraße 1, 85764 Neuherberg, Germany. [6]Max Planck Institute of Psychiatry, Munich 80804, Germany. [7]Department of Psychiatry, University of Münster, 48149 Münster, Germany. [8]Department of Neurodegenerative Diseases, Center of Neurology and Hertie Institute for Clinical Brain Research, University of Tübingen, Hoppe-Seyler-Straße 3, 72076 Tübingen, Germany. [9]Center for Neuropathology and Prion Research, Ludwig-Maximilians-Universität Munich, Feodor-Lynen-Str. 23, 81377 Munich, Germany. [10]Munich Cluster of Systems Neurology (SyNergy), Munich, Germany. [11]Section Clinical Neuroanatomy and Biobanking (CNAB), Department of Anatomy and Neurosciences, Amsterdam UMC, Vrije Universiteit Amsterdam, De Boelelaan 1108, 1081HV Amsterdam, The Netherlands. [12]Department of Stem Cell Biology, University Hospital Erlangen, Friedrich-Alexander-Universität Erlangen-Nürnberg (FAU), Glückstrasse 6, 91054 Erlangen, Germany. [13]Department of Molecular Neurology, University Hospital Erlangen, Friedrich-Alexander-Universität Erlangen-Nürnberg (FAU), Schwabachanlage 6, 91054 Erlangen, Germany. [14]Department of Mathematics, Technische Universität München, Boltzmannstraße 3, 85748 Garching bei München, Germany. [15]German Center for Neurodegenerative Diseases (DZNE) site Munich, Feodor-Lynen-Straße 17, 81377 Munich, Germany. [16]These authors contributed equally: Sebastian Schmidt, Malte D. Luecken, Dietrich Trümbach. ✉e-mail: florian.giesert@helmholtz-muenchen.de; fabian.theis@helmholtz-muenchen.de; wurst@helmholtz-muenchen.de

