## [Peer Review File · Nature Communications]

Primary cilia and SHH signaling impairments in patient-derived cellular models of Parkinson's diseaseREVIEWER COMMENTS

Reviewer #1 (Remarks to the Author):

Comments of Schimdt et al.

In this manuscript, Schmidt et al., demonstrate that increased SHH signaling due to ciliary dysfunction is associated with the early pathoetiology of the sporadic PD using scRNA-seq of iPSC derived hNPCs and neural cell types. They further demonstrate that changes in the primary cilia are also observed in the familial forms of PD using PINK1-deficient human and mouse models. Interestingly, when the late stage PD samples are analyzed, they observed opposite changes in the PC (elongation) which they suggest is due to a compensatory reaction to missing dopaminergic innervation in the adult brain. Their results suggest that downregulation of HH signaling can be a therapeutic intervention during the early stages of the PD.

This study is interesting and also provides important scRNA-seq data for the field. However, some of the conclusions are not fully supported by the evidence provided in the current form of the manuscript. Specifically,

- 1) Fig. 1b,c. Authors suggest that the number of CNVs are not significantly different however the regions affected may be different. For example, there seems to be sPD-specific changes in chr 1 and 20. Authors should comment on these genomic regions and whether they involve genes that are relevant to the pathways described in the rest of the manuscript.
- 2) Fig. 2. Labelling of the clusters by control vs. sPD (Fig. S2b) should be shown in the main figure. Furthermore, authors comment that there is no enrichment for control or sPD cells/or different donors in the clusters (lines 153-157), however it is hard to tell from the UMAPs shown. The proportions of control/sPD cells particularly should be reported in the manuscript.
- 3) Fig. 2c. The RNA velocity graph should also be labelled by the conditions (control/sPD) and by the original cluster identities shown in Fig. 2b. This is particularly important to understand which clusters do the cells in the root population correspond to. Lack of this knowledge makes it hard to understand the differential expression analysis presented in the next section.
- 4) Is the data presented in Fig. S3a from all the cells or some specific population (cluster)? This reviewer finds this section hard to follow. Particularly, it is hard to understand on which clusters the differential expression analysis are performed. Authors suggest that each cluster/population show significant changes in the sPD compared to the control cells within each cluster however, however, data is buried in the supplementary tables. Adding a summary table summarizing common changes amongst clusters/and the top 3-4 gene sets specific to each cluster would be helpful.
- 5) It would be helpful to include IF of the cilia in hNPCs in Fig. 4. Importantly, authors own analysis and IFs shown in Fig. 1 suggest that hNPC cultures are heterogenous. Did authors perform co-staining for NPC makers and/or proliferative markers to make sure that they quantified the stem/progenitor population? Please clarify since this may affect directly impact the results. The same comment is also true for Fig. S6)
- 6) Cyclopamine data in Fig. 4i-k should be presented with the respective vehicle treated controls. Were the cell cycle dynamics affected by cyclopamine treatment, since this can directly affect the PC length quantifications?
- 7) This reviewer thinks that the manuscript can benefit from a more thorough analysis of the expression levels of HH pathway genes and additional downstream targets of the pathways in the scRNA-seq.
- 8) Would overactivation of the HH pathway in control hNPCs cause PC length alterations? Alternatively, would overexpression of the Gli3-FL form in the control cells or Gli3R in the sPD cells also recapitulate these findings on the PC length? Finally, authors show that the PC length is affected in both the hNPCs and the iPSC-derived neurons in Fig. 4a-b. I wonder if 1) the dopaminergic neuron differentiation is affected in the sPD cells, and 2) if the HH pathway signaling (and Gli3-FL) is also upregulated in the iPSC-derived neurons from the sPD donors compared to the cells derived from the control donors.

Minor comments:

- 1) Number of the cells used for each analysis from each donor should be added.
- 2) Line 182: I think this should be Supplementary fig. 4b.
- 3) What do the Gray and Black colors refer to in Fig. 3c.
- 4) Line 475: the data shown in Fig. S1bc does not show that the cells have stable karyotype as suggested.

Reviewer #2 (Remarks to the Author):

In this manuscript, the authors combined scRNA Seq and conventional cell biological analyses to show that both human sPD and fPD patient hiPSC-derived NPCs and DAns exhibit shortened primary cilia and altered Shh signaling. They also showed that blocking Shh signaling in sPD NPCs rescued the cilia length and mitochondrial defects. Finally, they showed that the postmortem sPD patients exhibit longer cilia in striatal neurons, and provided evidence that such a surprising phenotype was secondary to prolonged deprivation of DA input. This manuscript proposed a provocative new mechanism for PD pathogenesis and potential direction of treatment, and provided multiple lines of evidence to support their conclusion. Nevertheless, additional data and in-depth discussion are needed to further strengthen their argument.

Major points:

1. As a key piece of evidence for altered Shh signaling in sPD and fPD patient NPCs and postmortem striatum, the authors pointed out that there was an overrepresentation of predicted Gli3 target genes in the DEGs. These should be further analyzed so see whether the expression changes overwhelmingly support the conclusion that Shh signaling is enhanced in all three contexts.
2. Another piece of evidence for enhanced Shh signaling was the increased Gli3FL and Gli3R levels in the nucleus, which was shown in Fig. S6b. There are a few issues with this data. First, the authors need to label the real signal and background bands as both the Gli3FL and Gli3R panels have more than one bands. Second, the legend indicated that nuclear Gli3 proteins are shown, but the loading control is b-actin. It should be a nuclear loading control (e.g. nuclear lamin) instead. Finally, cytoplasmic fraction along with its own loading control should be presented as well.
3. The only example of the transcriptional output of Shh signaling was Foxa2. Additional well recognized outputs, such as Ptch1, Ptch2, Gli1, Hhip1 etc, should be examined to support the conclusion that Shh signaling is altered.
4. It is critical to analyze Shh signaling in more detail in cyclopamine-treated cells, including how total and nuclear Gli3 changes, how much the expression of multiple Shh target genes change, and how these changes compare to untreated and treated control cells.
5. The authors reported that cyclopamine treatment did not change cilia length or mitochondria in control cells. Are there any changes in Shh signaling? Again, Foxa2 expression alone is insufficient.
6. A Shh/Smo agonist (purmorphamine) is included in the culture medium for both NPCs and DAns. A straightforward experiment would be simply decreasing or withdrawing this to see whether an effect can be observed similar to adding cyclopamine.
7. It is surprising that blocking Shh signaling leads to the lengthening of cilia, and only in sPD cells. Did the treatment alter cell cycle specifically in sPD cells? The authors may try to speculate on the possible mechanism underlying such a negative role of Shh on cilia elongation, as no such a role has been reported previous to my best knowledge.

Minor points

1. Page 7, line 10. I would not use the phrase "activation of Ptch1", as Shh binding inhibits, not activates, Ptch1 activity.
2. Previous studies suggested that Shh signaling and the primary cilia were essential for

NSC maintenance, thus the authors may want to discuss the potential side effect of blocking Shh signaling as a potential treatment based on these findings.

Reviewer #3 (Remarks to the Author):

The authors present a well-performed and interesting study, utilising a human NPC model, showing this has mitochondrial respiration deficits. The study eloquently utilises single cell techniques to find a role of SHH in PC shortening (or lengthening) in model and post mortem tissue. The findings associated with PC across multiple models could present an exciting therapeutic target. Some points are raised for clarification and to aid in understanding the merits of the model and PC findings to wider PD.

Could the authors please comment and make clear in the manuscript if each clone of each participant was treated as an individual or if when presenting data such as Seahorse and complex I activity in Figure 1, if these were treated as independent lines. It is not clear if every clone was used in every experiment and therefore how the data is presented and was analysed.

Seahorse experiments, the authors comment that basal and maximal respiration deficits result in reduction in ATP linked respiration, could the authors please include the calculated graphs, showing mitochondrial basal respiration, spare capacity, ATP linked respiration and coupling efficiency, perhaps in supplementary figures? This enables the reader to easily see the significant differences in the various respiration parameters measured.

Were the sPD lines pre-selected in anyway, or would the authors postulate that these same signatures and responses would be present across the sPD population? If so, how does this fit with work published over the past 2 years in fibroblasts showing a large spread of mitochondrial abnormalities across large sPD cohorts? This is a very important question when thinking about PC being a therapeutic target for PD and therefore, it would be useful if the authors could comment in the manuscript.

Based upon Seahorse and complex I, the authors state the NPC's are a suitable model to use for high throughput assays to represent a PD phenotype. Could the authors comment if these metabolic deficiencies are present in the fibroblasts and iPSC stem cells from the same lines, in addition could the authors comment on why these metabolic deficiencies are present in the NPC's and not astrocytes generated from the same lines, what could this mean for the aetiology of PD?

In Figure 4 data, the authors state the data shown is from triplicates, 5 control clones and 7 sPD clones. Is this is then in a reduced cohort of cells from the data in Figure 1 which the authors state is from 10 control clones and 14 sPD? If so, how were those clones selected, is it 1 clones per line rather than 2? For the data in Figure 4, does each dot represent the mean of 3 biological replicates of each clone, could the authors clarify this? In addition, clarification that the 400 cells or PC referred to were across multiple biological repeats. Finally the Seahorse trace shown in Figure4k, needs the vehicle treated control conditions showing on the same plot, and the additional calculations and graphs shown for each parameter to be able to show that the cyc inhibition has had any effect on the respiration state of the cells. This is critical to the PC and SHH signalling effect being critical to make the link to the overall mitochondrial health.

The data and associated explanation provided by the authors for the decrease and increase in PC length between cell model and post mortem is well articulated; however, as this is key to hypothesis of PC involvement in PD pathogenesis proposed here, is it possible for the authors to validate the PC measurements via an alternative approach or staining?

Minor comments

Line 75 make sure Cntrl is defined

Line 78 grammar, allowed "us" to

Reviewer #4 (Remarks to the Author):

In the manuscript "Primary cilia dysfunction in Parkinson's disease etiology" the authors address the question of whether molecular and cellular alterations of primary cilia in PD patient rejuvenated cell lines are a distinctive feature of early PD development. To this end, the authors used single cell transcriptomics to comprehensively investigate the cellular composition of neural precursors differentiated from several sporadic PD lines and to determine the molecular mechanism underlying primary cilia dysfunction. They identify alterations in SHH signaling pathway as being responsible for mitochondrial deficits in the PD lines analyzed. Additionally, ciliary alterations were observed in postmortem tissue of sporadic PD patients as well as in familial PD iPSCs. Although an extremely valuable tool, the use of iPSCs to model neurodegenerative disorders such as PD is still somewhat debated, and new reprogramming methodologies are being developed with the aim of generating human cellular models that retain age and epigenome modifications of PD patients (e.g. direct reprogramming). The presented manuscript could contribute to providing novel insights into this topical issue by identifying ciliary dysfunction as an early event during differentiation that may guide our understanding of the etiology of PD. Overall, the conclusions of the manuscript are well supported by the data, and the manuscript addresses important and as yet unresolved questions in the field. The characterization of primary cilia alteration in postmortem tissue and in familial forms of PD do indeed corroborate the main objective of the manuscript. However, involvement of SHH signaling in primary cilia dysfunction in PD is not novel (Dhekne et al., eLife 2018; Bae et al., Cell Death and Disease 2019, Sobu et al., PNAS 2021). To make the manuscript suitable for publication in a top-tier journal such as Nature Communications, the authors should provide additional experiments to consolidate their findings especially regarding the use of iPSCs as a potential model to study PD pathoetiology. Major comments: 1) In contrast to older PAX6+-inducing protocols, it has now been shown that for correct patterning towards authentic midbrain dopaminergic neuron generation, hPSCs have to pass through an intermediary floor plate progenitor state involving FOXA2+/LMX1+/PAX6- (Fasano et al., Cell Stem Cell 2010; Kriks et al., Nature 2011; Kirkeby et al., Cell Rep 2012; Nolbrant et al., Nat Prot 2017). The expression of PAX6 in NPC culture suggests that the authors have not used a floor plate transition protocol. They should at least clarify the choice of their differentiation protocol in the text. 2) An in-depth analysis of PD and ctrl lines should be carried out when they are still cultured under pluripotent conditions. Mitochondrial stress testing should be performed together with either bulk- or sc-RNAseq to provide a comprehensive characterization at transcriptional, metabolic, and functional resolution of undifferentiated PD cell lines adopted in this study. 3) sc-RNAseq analysis is generally well carried out, but further details on the methods used should be reported (a) and the following new analyses should be performed (b-e): a) Specifically, the authors should clarify and specify for both sPD and fPD cell lines analyzed: - how the data looked before integration - what cutoffs were used to define DE genes and what the rationale was - what resolution was chosen to define cell clusters - number of single cells analyzed. Additionally, was the harmony integration performed based on cell lines used? Did the authors detect any cluster(s) that might indicate cell doublets? b) In Fig. 2b, a barplot showing the percentage of each cluster in each donor line would be useful to interpret reproducibility. This should be accompanied by statistical testing of difference in cluster proportion in PD/C. c) What happened to the NCSC and glial clusters in the velocity analysis? d) The DEG analysis (Fig. 2) would benefit from a more stringent presentation. It needs to be clear which comparisons have been made and if velocity or regular clusters were used. 12,806 genes are detected as DE, which seems like an extremely high number considering that 4-5000 genes are typically detected per cell. Is the analysis correctly controlling for false discoveries? FC cutoffs are stated as " $0.83 > FC > 1.2$ ", which I think would be more clearly written as $|FC| > 20\%$. e) The authors should also provide a quantification panel (e.g. dotplot or heat map or violin plot) of mitochondrial and PD-associated gene expression between PD and ctrl NPC cultures. 4) The comparison with postmortem tissue (Fig. 5) is important and adds relevance to the data presented in this manuscript. A more direct comparison of genes found in postmortem vs author derived dataset would be prudent. What proportion of genes

overlap? Are there any themes among genes that do not agree between datasets? Single cell results from NPC PD lines should also be analyzed in direct comparison to the single cell dataset generated from substantia nigra of human post-mortem brain tissue recently published by Agarwal et al, Nat Comm 2020. 5) The manuscript might benefit from the addition of sc-RNAseq analysis of mature postmitotic DA neurons differentiated from PD lines vs ctrls. This could reconstruct the trajectory of the disease, yielding a greater understanding of the early phases of PD progression. 6) The authors should clarify the cellular composition and molecular signature of fPD lines and provide a direct comparison to sPD NPCs highlighting points of similarity and differences as they do at protein level. Minor comments: 1) Did the identified CNVs (Fig. 1b/c) fall within any interesting PD-related loci? 2) Bae et al., Cell Death and Disease 2019 and Sobu et al., PNAS 2021 should be reported and commented in the Discussion section. 3) Is supplementary Fig. 6a a cytofluorimetric analysis? If so, please provide representative FACS plots for both ctrl and sPD. 4) Provide primary cilia immunostaining with anti-Adcy3 and anti-Rbfox3 also for midbrain dopamine neurons as in Fig. 5i. 5) NSC2 population is missing in the UMAP plot (Fig. 6a) from PINK1 ko and isogenic ctrl hNPC cell line. High resolution clustering analysis could be run. 7) Line 136 page 4 should be FOXA2.

Dear Reviewers,

We thank you all for your valuable comments on our manuscript which helped to improve the manuscript substantially.

Concerning the revised manuscript, we want to inform you that due to administrative updates within the consortium that generated the hiPSC lines, we had to provide different anonymized patient identifiers for the publication of patient-related raw data. Thus, to ensure that only anonymized patient identifiers are stored in all data objects and to improve the reproducibility of the data analysis, we generated a consistent, updated environment with all data analysis packages, removed old patient identifiers from the data objects, and reran the full analysis in this environment. This guarantees that anyone can reproduce the full analysis scripts and generate the same data objects we have from the data uploaded to GEO. As the original analysis used new versions of methods when they became available, re-running the full pipeline in a consistent environment, slightly changed the representation of the data within the limits of the stochasticity of methods (e.g, normalization and differential expression methods require numerical optimization processes that will slightly change exact values of the method outputs). Despite these small fluctuations analysis outcomes of the rerun and the interpretation and validation of the data remained as before. Thus the rerun – which can be considered as a second independent replication of our analysis - confirms and thus strengthens our initial data.

Within the revised manuscript, we highlighted additional passages, which were requested by the reviewers, in yellow. Two small passages which we eliminated due to the restructuring of the manuscript are also indicated (crossed out).

A point-by-point response to each reviewers' comments is indicated below. For better readability of this part of our response, we highlighted each reviewers' comments in a different color, whereas our responses are always kept in black.

Thank you once again for your valuable input and specifically the time taken to help us to improve our manuscript.

Yours

Prof. W. Wurst

REVIEWER COMMENTS

Reviewer #1 (Remarks to the Author):

Comments of Schmidt et al.

In this manuscript, Schmidt et al., demonstrate that increased SHH signaling due to ciliary dysfunction is associated with the early pathoetiology of the sporadic PD using scRNA-seq of iPSC derived hNPCs and neural cell types. They further demonstrate that changes in the primary cilia are also observed in the familial forms of PD using PINK1-deficient human and mouse models. Interestingly, when the late stage PD samples are analyzed, they observed opposite changes in the PC (elongation) which they suggest is due to a compensatory reaction to missing dopaminergic innervation in the adult brain. Their results suggest that downregulation of HH signaling can be a therapeutic intervention during the early stages of the PD.

This study is interesting and also provides important scRNA-seq data for the field. However, some of the conclusions are not fully supported by the evidence provided in the current form of the manuscript.

Specifically,

1) Fig. 1b,c. Authors suggest that the number of CNVs are not significantly different however the regions affected may be different. For example, there seems to be sPD-specific changes in chr 1 and 20. Authors should comment on these genomic regions and whether they involve genes that are relevant to the pathways described in the rest of the manuscript.

Genes that fall into genomic regions affected by CNVs were investigated and analyzed using pathway enrichment analysis (Pathway Studio). We could show that no pathway or genes were affected by CNVs that were of interest for this study. We explicitly included this additional information now in the manuscript on page 3, lines 27-30.

2) Fig. 2. Labeling of the clusters by control vs. sPD (Fig. S2b) should be shown in the main figure. Furthermore, authors comment that there is no enrichment for control or sPD cells/or different donors in the clusters (lines 153-157), however it is hard to tell from the UMAPs shown. The proportions of control/sPD cells particularly should be reported in the manuscript.

Labeling of the clusters by control vs. sPD is now included in the main figures (**Fig. 3c**). In addition, the proportion of donors to clusters as well as control/sPD cells to clusters is now reported in the figures. Now shown in **Fig. 3d** and **Supplementary Fig. 1d**, respectively. To quantify the distribution of control/sPD cells to clusters, we used scCODA¹, which uses a Dirichlet multinomial model to fit the observed counts per cell type per donor, to test for significant changes in compositions in Parkinson patients vs control samples. As expected, no significant differences were detected (**Supplementary Fig. 1e**). We stated this additional information in the manuscript on page 5, lines 22-23.

3) Fig. 2c. The RNA velocity graph should also be labeled by the conditions (control/sPD) and by the original cluster identities shown in Fig. 2b. This is particularly important to understand which clusters do the cells in the root population correspond to. Lack of this knowledge makes it hard to understand the differential expression analysis presented in the next section.

The root population contains a mixture of donors and conditions. Thus, the differential expression analysis is a general comparison between cells that appear to be the root of the RNA velocity trajectory across donors and conditions. Labeling of the RNA velocity graph by the original cluster identity, donors as well as the disease state (control vs. sPD) is now included in the supplementary figures (**Supplementary Fig. 1g**).

4) Is the data presented in Fig. S3a from all the cells or some specific population (cluster)? This reviewer finds this section hard to follow. Particularly, it is hard to understand on which clusters the differential expression analysis are performed. Authors suggest that each cluster/population show significant changes in the sPD compared to the control cells within each cluster, however, data is buried in the supplementary tables. Adding a summary table summarizing common changes amongst clusters/and the top 3-4 gene sets specific to each cluster would be helpful.

The data presented in the main **Fig. 3f** is based on pathway enrichment analysis of DEGs of the root population and is a detail of the complete network that is shown in **Supplementary Fig. 3a**. This is now explicitly stated in the corresponding figure captions. Summary tables of the top 8 gene sets affected per cluster are now included in the main body of the manuscript as **Table 1**.

5) It would be helpful to include IF of the cilia in hNPCs in Fig. 4. Importantly, authors own analysis and IFs shown in Fig. 1 suggest that hNPC cultures are heterogeneous. Did authors perform co-staining for NPC makers and/or proliferative markers to make sure that they quantified the stem/progenitor population? Please clarify since this may affect directly impact the results. The same comment is also true for Fig. S6)

The immunofluorescence staining of hNPCs is now presented in the main **Fig. 5b**. The identity of these NPCs was verified by NES staining and only primary cilia associated with NES positive hNPCs were quantified. The stainings revealed that the large majority of cells (>90%) were NPCs. The cultures used for the experiments in **Supplementary Fig. 6b,c** were alike.

6) Cyclopamine data in Fig. 4i-k should be presented with the respective vehicle treated controls. Were the cell cycle dynamics affected by cyclopamine treatment, since this can directly affect the PC length quantifications?

We show that cyclopamine treatment did not affect the cell proliferation rate. This information we now included in the main **Fig. 7a**. Furthermore, cyclopamine data are now always presented with vehicle-treated controls (DMSO treated) in **Fig. 7**. The only exception is **Fig. 7d**, the quantification of GLI3 processing by western blots. Here the corresponding vehicle-treated control is displayed in **Fig. 6c**. This separation can be justified for semi-quantitative methods such as western blots. We explicitly stated this now in the manuscript on page 9, lines 6-20.

7) This reviewer thinks that the manuscript can benefit from a more thorough analysis of the expression levels of HH pathway genes and additional downstream targets of the pathways in the scRNA-seq.

We now have included the analysis of expression levels of additional HH pathway genes (*PTCH1*, *PTCH2*, *GLI1*, *GLI2*, *GLI3*) and other respective downstream targets (*FOXA2*, *GBP1*, *SHOX2*) analyzed by RT-qPCR in the manuscript (**Fig. 6a and Fig. 7c**). We explicitly stated this now in the manuscript for hNPCs on page 8, lines 35-36; for DANs on page 8, lines 39-40 and for hNPCs after cyclopamine treatment on page 9, lines 11-14.

8) Would overactivation of the HH pathway in control hNPCs cause PC length alterations? Alternatively, would overexpression of the Gli3-FL form in the control cells or Gli3R in the SPD cells also recapitulate these finding on the PC length?

Overactivation of the SHH signaling pathway did not result in increased proliferation rates also with higher concentrations of activators (purmorphamine) both in sPD and Ctrl hNPCs (**ReFig. 1**) as determined by an EdU incorporation assay. Concerning the length of PC upon overactivation of the SHH signaling pathway, we performed additional experiments depicted below. These experiments revealed that by using low concentrations of the SHH-signaling pathway activator PMA, the length of primary cilia was reduced **only** in sPD clones. However, a reduction in PC length could also be observed in Ctrl clones upon strong activation of SHH signaling (see below). This strengthens our hypothesis of an overactivation of SHH signaling in sPD since the overactivation of the SHH pathway in Ctrl induces the same phenotype as observed in sPD cells.

ReFig. 1 | (left) Cell proliferation rate of hNPCs (DMSO ctrl and PMA treated - 1 μM or 2 μM for 4 days) determined by EdU incorporation after 2 h of EdU treatment. Analyzed were n > 200 cells per individual and independent replicate. **(right)** Density plot illustrating the distribution of PC length (in μm) in an hNPC population (DMSO ctrl and PMA treated - 1 μM or 2 μM for 4 days). PC length was measured in immunostainings of hNPCs positive for anti-NES and anti-ARL13B. Analyzed were n > 30 PC per individual and independent replicate. All experiments were performed in triplicates, n = 5 Ctrl and 7 sPD clones.

9) Finally, authors show that the PC length is affected in both the hNPCs and the iPSC-derived neurons in Fig. 4a-b. I wonder if 1) the dopaminergic neuron differentiation is affected in the sPD cells, and 2) if the HH pathway signaling (and Gli3-FL) is also upregulated in the iPSC-derived neurons from the sPD donors compared to the cells derived from the control donors.

Revision of manuscript NCOMMS-21-30277 - Primary cilia dysfunction in Parkinson's disease etiology
Response to reviewer's comments

- 1) Alterations in primary cilia morphology do not affect the differentiation potential of neurons, nor dopaminergic neurons. We quantified the number of neurons as well as dopaminergic neurons differentiated from Ctrl and sPD hiPSCs. Additionally, we assessed the neurite morphology of dopaminergic neurons, which was not altered. Thus, demonstrating that also neuron maturation was not affected by the disease state nor alterations in primary cilia morphology. This is now shown in **Fig. 1d,f**, respectively. We explicitly stated this now in the manuscript on page 4, line 3-8.
- 2) SHH signal transduction is also affected in sPD DAns. We repeated the quantification of cytoplasmic GLI3-FL as well as nuclear GLI3-FL and GLI3-R in these DAns. This is now shown in **Fig. 6d**. We explicitly stated this now in the manuscript on page 8, lines 39-40.

Minor comments:

- 1) Number of the cells used for each analysis from each donor should be added.

The number of cells per donor and replicate is now stated in the corresponding Material and methods sections and for e.g. PC length quantification also in the Figure legends (e.g. **Fig. 5**).

- 2) Line 182: I think this should be Supplementary fig. 4b.

Yes, meant was Supplementary Fig. 4b. This has been corrected.

- 3) What do the Gray and Black colors refer to in Fig. 3c.

Due to the suggested restructuring of the chapter "Dysregulated genes point towards primary cilia dysfunction" and the corresponding Fig. 3 (now **Fig. 4**), this is no longer relevant. Before, non-significantly enriched pathways were colored in gray, significantly in black. Now only significantly enriched pathways are displayed.

- 4) Line 475: the data shown in Fig. S1bc does not show that the cells have stable karyotype as suggested.

The phrase has been changed to "hiPSCs were screened regularly for pluripotency and chromosomal aberrations" which we now also report in the manuscript as **Supplementary Tables 2-3**.

Reviewer #2 (Remarks to the Author):

In this manuscript, the authors combined scRNA Seq and conventional cell biological analyses to show that both human sPD and fPD patient iPSC-derived NPCs and DANs exhibit shortened primary cilia and altered Shh signaling. They also showed that blocking Shh signaling in sPD NPCs rescued the cilia length and mitochondrial defects. Finally, they showed that the postmortem sPD patients exhibit longer cilia in striatal neurons, and provided evidence that such a surprising phenotype was secondary to prolonged deprivation of DA input. This manuscript proposed a provocative new mechanism for PD pathogenesis and potential direction of treatment, and provided multiple lines of evidence to support their conclusion. Nevertheless, additional data and in-depth discussion are needed to further strengthen their argument.

Major points:

1. As a key piece of evidence for altered Shh signaling in sPD and fPD patient NPCs and postmortem striatum, the authors pointed out that there was an overrepresentation of predicted GLI3 target genes in the DEGs. These should be further analyzed so see whether the expression changes overwhelmingly support the conclusion that Shh signaling is enhanced in all three contexts.

Possibly there was a false impression created that in all systems examined, that is sPD, fPD, and postmortem sPD SHH-signaling pathways, and thus GLI3 processing is altered and in all cases the underlying cause of PC alterations. It was not our intention to generate this impression. We instead wanted to point out clearly, that in all three conditions we - and others - observe an alteration in PC, be it on the morphological and/or on the molecular level. Thus,

1. We now point out more clearly that there is a clear distinction between postmortem sPD - and thus PD end states - and our model system of early PD. This is already indicated by their morphology: PCs in postmortem tissues are elongated and not shortened. An involvement of PCs in sPD pathology is also clearly indicated by the meta-analysis of published transcriptome data, which hints at a dysregulation of PC biology. Indeed also a recently published transcriptome analysis² states that cilia and ciliary function might be affected, however, left it uncommented. The evaluation of the biological consequences of these differences is, however, in our view beyond the scope of this manuscript.
2. Concerning the fPD forms, we point out that there also exist alterations in PC morphology (*PINK1* KO iPSCs and *Pink1* KO mice) and ciliogenesis (reduced fraction of ciliated neurons - published data from *Lrrk2* mice³) in the disease state and that our transcriptome analysis of the *PINK1* KO hNPCs also reveals significantly altered cilia associated pathways. However, the overrepresentation of GLI3 targets could not be observed, the reason why we clearly state in the discussion, that cilia alterations in sPD and fPD are present, but that it is highly likely that the underlying molecular alterations are distinct.

Taken together, altered PC biology is affected in both sPD and fPD as well as in early and late PD, even though - and this is important to mention - the molecular basis of these alterations are highly likely to be different. This, we discussed in the manuscript on page 12, lines 26-40.

2. Another piece of evidence for enhanced Shh signaling was the increased Gli3FL and Gli3R levels in the nucleus, which was shown in Fig. S6b. There are a few issues with this data. First, the authors need to label the real signal and background bands as both the Gli3FL and Gli3R panels have more than one bands. Second, the legend indicated that nuclear Gli3 proteins are shown, but the loading control is b-actin. It should be a nuclear loading control (e.g. nuclear lamin) instead. Finally, cytoplasmic fraction along with its own loading control should be presented as well.

The signal bands that were used to quantify GLI3-FL and -R levels are now specifically indicated by an arrow in the Western blots shown in **Supplementary Fig. 9**. Western blots for the nuclear fraction of proteins presented in **Fig. 6**, **Fig. 7**, and **Supplementary Fig. 9** were repeated with a nuclear loading control (PCNA for NPCs, H1-0 for DANs). This didn't change the outcome, it even increased the significance.

3. The only example of the transcriptional output of Shh signaling was *Foxa2*. Additional well recognized outputs, such as *Ptch1*, *Ptch2*, *Gli1*, *Hhip1* etc, should be examined to support the conclusion that Shh signaling is altered.

We now have included the analysis of expression levels of additional HH pathway genes (*PTCH1*, *PTCH2*, *GLI1*, *GLI2*, *GLI3*) and other respective downstream targets (*FOXA2*, *GBP1*, *SHOX2*) analyzed by RT-qPCR in the manuscript (**Fig. 6a** and **Fig. 7c**). We explicitly stated this now in the manuscript for hNPCs on page 8, lines 35-36; and for hNPCs after cyclopamine treatment on page 9, lines 11-14.

4. It is critical to analyze Shh signaling in more detail in cyclopamine-treated cells, including how total and nuclear Gli3 changes, how much the expression of multiple Shh target genes change, and how these changes compare to untreated and treated control cells.

We show that cyclopamine treatment did not affect the cell proliferation rate. This information we now included in the main **Fig. 7a**. Furthermore, cyclopamine data are now always presented with vehicle-treated controls (DMSO treated) in **Fig. 7**. The only exception is **Fig. 7d** - the now also included quantification of GLI3 processing by western blots upon cyclopamine treatment. Here the corresponding vehicle-treated control is displayed in **Fig. 6c**. This separation can be justified for semi-quantitative methods such as western blots. Additionally, expression levels of dysregulated SHH pathway genes (*PTCH1*, *GLI3*) and other respective downstream targets (*FOXA2*, *GBP1*, *SHOX2*) were quantified after cyclopamine treatment (**Fig. 7c**). We explicitly stated this now in the manuscript on page 9, lines 6-20.

5. The authors reported that cyclopamine treatment did not change cilia length or mitochondria in control cells. Are there any changes in Shh signaling? Again, *Foxa2* expression alone is insufficient.

As mentioned in the reply to question 4, we now also included the processing of GLI3 analyzed by western blots after cyclopamine treatment in the main **Fig. 7d**. In line with our previous results, cyclopamine treatment diminished the alterations between Ctrl and sPD regarding SHH signal transduction. This also led to similar expression levels of the SHH pathway genes (*PTCH1*, *GLI3*) and other respective downstream targets (*FOXA2*, *GBP1*, *SHOX2*) analyzed by RT-qPCR. This is now shown in **Fig. 7c**. We explicitly stated this now in the manuscript on page 9, lines 6-20.

6. A Shh/Smo agonist (purmorphamine) is included in the culture medium for both NPCs and DANs. A straightforward experiment would be simply decreasing or withdrawing this to see whether an effect can be observed similar to adding cyclopamine.

We have done this experiment (**ReFig. 2**). After removal of purmorphamine, both bioenergetic and ciliary alterations normalized in sPD cells. However, we did not include this additional information in the manuscript as purmorphamine stimulation is essential for DAN differentiation. Thus, using cyclopamine as an inhibitor of SHH-signal transduction is in our view more straightforward. Still the results, when removing purmorphamine from the culture are fully supporting the data obtained with cyclopamine.

ReFig. 2 | (left) Mitochondrial stress test performed in hNPCs derived from 5 Ctrl and 7 sPD patients (in triplicates) 4 days after PMA removal using a Seahorse XFe96 Extracellular Flux Analyzer. Injected were (A) Oligomycin (1 µg/ml), (B) FCCP (0.5 µM) and (C) Rotenone (5 µM)/Antimycin A (2 µM). Measurement progression is shown with means ± standard error of the mean (SEM). **(right)** Density plot illustrating the distribution of PC length (in µm) in an hNPC population 4 days after PMA removal. PC length was measured in immunostainings of hNPCs positive for anti-NES and anti-ARL13B. Analyzed were n >30 PC per individual and independent replicate. All experiments were performed in triplicates, n = 5 Ctrl and 7 sPD clones.

7. It is surprising that blocking Shh signaling leads to the lengthening of cilia, and only in sPD cells. Did the treatment alter cell cycle specifically in sPD cells? The authors may try to speculate on the possible mechanism underlying such a negative role of Shh on cilia elongation, as no such a role has been reported previous to my best knowledge.

We show that cyclopamine treatment did not affect the cell proliferation rate, thus SHH blocking did not alter cell cycle length and therefore alterations in cilia length are not due to an altered cell cycle. This information we now included in the main **Fig. 7a**. We also stated this information in the manuscript on page 9, line 16.

The fact that cilia influence Shh signaling is well documented in the literature and also discussed in the manuscript (**Fig. 4,5, and 6**) ending with the speculation that in sPD intraflagellar transport might be disrupted. Further speculations upon molecular mechanisms are less straightforward since dozens, and probably hundreds of proteins are required to build a primary cilium and their dependencies on SHH-signaling as well as their precise role in cilia elongation are often far from being understood. Still, it has to be stressed that amongst the predicted GLI3 target genes dysregulated in sPD patients, around 20% are associated with cilia. Some of them could be involved in the control of cilia length upon alterations of SHH-signaling. However, deciphering which ones is beyond the scope of this manuscript. We now have pointed

out such a hitherto unreported role of SHH signaling in cilia elongation and the respective broad speculation about the underlying molecular mechanisms in the manuscript on page 13, lines 9-18.

Minor points

1. Page 7, line 10. I would not use the phrase "activation of Ptch1", as Shh binding inhibits, not activates, Ptch1 activity.

The phrase has been changed to "Binding of SHH inhibits PTCH1, which allows the translocation of SMO to the PC and subsequently prevents the degradation of full-length GLI3 (GLI3-FL) to GLI3-repressor (GLI3-R)". This is now reported on page 8, lines 27-29.

2. Previous studies suggested that Shh signaling and the primary cilia were essential for NSC maintenance, thus the authors may want to discuss the potential side effect of blocking Shh signaling as a potential treatment based on these findings.

It is correct that Shh-signaling and the primary cilia are essential for NPC maintenance, differentiation specifically during development. Thus, it is not surprising that SHH-signaling disruption is associated with several severe neurodevelopmental diseases, such as holoprosencephaly⁴. However, its role in the adult brain is less well understood. It is known from in vivo studies, that proliferation of adult neural stem cells, as their embryonic counterparts depend on SHH and thus contributes to the maintenance of adult neural stem cell niches in the subventricular zone, the dentate gyrus, and olfactory bulb⁵. Also, in cell culture blockage with Cyclopamine is associated with reduced NPC survival, reduced general differentiation, and increased astroglia differentiation⁶. Thus, interfering with SHH signaling as a therapeutic approach might well lead to disruption of neurogenesis, which has been shown to affect different behaviors such as social memory, stress coping and possibly anxiety-related behavior^{7,8}. It has also been hypothesized that it may be involved in the pathoetiology of schizophrenia and autism^{9,10}. Thus, adverse side effects concerning the occurrence of psychiatric symptoms might indeed be an issue when inhibiting Shh-signaling in adults.

There are three SHH inhibitors (vismodegib, sonidegib, and glasdegib) targeting the SMO receptor having been approved by the FDA for treatment of basal cell carcinoma, all of which are right now in clinical trials to evaluate their efficacy and safety in treating medulloblastomas¹¹. Adverse effects of these compounds are reported, such as muscle cramps, but so far to our knowledge, no adverse effect concerning the emergence of psychiatric problems in adults has been reported. Still, due to the high importance of SHH signaling during development a SHH-inhibition in pregnant women or children and teens is definitively not recommended. As an example, it has been reported that in children upon vismodegib treatment a premature growth plate fusion can occur, which is, however, not relevant in adult patients such as PD patients in an early phase of the disease, where growth plates have already fused.

We deliberately did not include this discussion in the manuscript because in our opinion the suggested repurposing of SMO-inhibitors as a treatment for early SPD is yet far from being applicable. Preclinical

Revision of manuscript NCOMMS-21-30277 - Primary cilia dysfunction in Parkinson's disease etiology
Response to reviewer's comments

studies have not yet been initiated at all. However, we still wanted to have this statement in the manuscript, since our observation that SHH-signaling in early SPD might be overactive is a completely new concept in the field of SPD research, and if it holds true, there might already be a potential treatment option.

Reviewer #3 (Remarks to the Author):

The authors present a well-performed and interesting study, utilising a human NPC model, showing this has mitochondrial respiration deficits. The study eloquently utilises single cell techniques to find a role of SHH in PC shortening (or lengthening) in model and post mortem tissue. The findings associated with PC across multiple models could present an exciting therapeutic target. Some points are raised for clarification and to aid in understanding the merits of the model and PC findings to wider PD.

1) Could the authors please comment and make clear in the manuscript if each clone of each participant was treated as an individual or if when presenting data such as Seahorse and complex I activity in Figure 1, if these were treated as independent lines. It is not clear if every clone was used in every experiment and therefore how the data is presented and was analysed.

If more than one clone per patient was used for experiments, the average value per patient was calculated from these clones. Only average values per patient are presented in figures and were used for statistical analysis (t-test, Mann-Whitney Test, ANOVA). Thus, the statistic is always based on the comparison of 5 Ctrl individuals vs 7 SPD patients. This is now explicitly stated in the Methods section "Statistics and reproducibility" on page 29, lines 28-34. A summary of which clone was used for which experiment is also given in **Supplementary Table 15**.

2) Seahorse experiments, the authors comment that basal and maximal respiration deficits result in reduction in ATP linked respiration, could the authors please include the calculated graphs, showing mitochondrial basal respiration, spare capacity, ATP linked respiration and coupling efficiency, perhaps in supplementary figures? This enables the reader to easily see the significant differences in the various respiration parameters measured.

We definitely agree that this would make the data presented for the Seahorse experiments easier and better understandable. Thus, the calculated graphs showing basal, maximal, ATP-linked respiration, as well as proton leakage and spare capacity are now depicted together with the overview graph in its own figure (**Fig. 2**).

4) Based upon Seahorse and complex I, the authors state the NPC's are a suitable model to use for high throughput assays to represent a PD phenotype. Could the authors comment if these metabolic deficiencies are present in the fibroblasts and iPSC stem cells from the same lines, in addition could the authors comment on why these metabolic deficiencies are present in the NPC's and not astrocytes generated from the same lines, what could this mean for the aetiology of PD?

We now also included Seahorse experiments analyzing the original patient-derived fibroblasts and the thereof reprogrammed hiPSCs. This is now shown in **Fig. 2a,b**, respectively. Both the original fibroblasts as well as the hiPSCs did not exhibit alterations in mitochondrial function. These SPD specific alterations only occurred after the differentiation of hiPSCs into hNPCs and further into DANs (**Fig. 2c,d**). This is now stated in the manuscript on page 3, lines 32-36.

Indeed, the metabolic differences, as well as the PC phenotype, are not observed in astrocytes generated from the same lines. Already the above observation, that fibroblasts and hiPSCs from the same lines do not display the metabolic alterations is indicative of a highly cell type-specific phenotype. Intriguingly, the observed metabolic alterations are not found in cell types relying to a large part on glycolytic respiration, whereas neurons - exhibiting the phenotype - rely on oxidative phosphorylation to generate their energy.

During the course of differentiation, a so-called metabolic switch takes place. In between these two stages (hiPSCs) and matured neural cells, the NPCs are thought to take up an intermediate state. Indeed, by applying scRNA-seq to adult neural stem cells it is found that NSCs exist as a continuum through the processes of activation and differentiation and distinct states with distinct molecular profiles can be identified as it is the case for our hNPCs cultures (NSC1a, NSC1b, NSC2a, NSC2b)¹². Not only do distinct "NPC-subpopulations" exist, however, also their respective metabolic status seems to be important for the progression of their neural differentiation, i.e. exit from the NPC state into differentiated neurons as indicated by the importance of ROS levels for this exit¹³. Thus, it is not surprising that the first indication of altered oxidative phosphorylation is found in NPCs. However, the knowledge about molecular mechanisms underlying the observed metabolic switch from hiPSCs to neurons - according to literature taking place in NSCs - are only starting to be understood. Not to speak of the mechanisms on which the maintenance of glycolytic respiration during the course of differentiation into astrocytes is relying on. Future studies investigating RNA-velocity analysis in more detail might be a very promising approach in this respect but beyond the scope of this manuscript.

Concerning the question, of what it could mean for the pathoetiology of sPD, that astrocytes are not showing the phenotypes observed in neurons and NPCs we again can only speculate. Indeed a prominent role of dysfunctional astrocytes in PD etiology is starting to emerge. In our study, we have looked at two very specific molecular and cellular aspects such as mitochondrial respiration and morphology of primary cilia, both not impaired in sPD patient-derived astrocytes. Still, this does not exclude impairment of other astrocytic functions such as glutamate metabolism, Ca²⁺ signaling, fatty acid metabolism, antioxidant production, and inflammation, all of which are known - when dysregulated - to impact sPD etiology¹⁴. Further studies are necessary to specifically analyze astrocytes derived from sPD patients in these respects. Indeed, we are strongly convinced that this alley of research will reveal astrocytic molecular pathways dysregulated which possibly contribute to sPD etiology. In sum, our results do definitively not exclude an astrocytic dysfunction in sPD etiology.

As stated above we now have included Seahorse experiments on fibroblasts and hiPSCs from the same sPD patients of which the NPCs and DA neurons have been derived into our manuscript. However, the discussion about the cell type specificity of changes in mitochondrial respiration and PC length we have not included, since in our opinion this might be still too speculative and biased by our own view on sPD etiology.

3) Were the sPD lines pre-selected in anyway, or would the authors postulate that these same signatures and responses would be present across the sPD population? If so, how does this fit with work published over the past 2 years in fibroblasts showing a large spread of mitochondrial abnormalities across large sPD

cohorts? This is a very important question when thinking about PC being a therapeutic target for PD and therefore, it would be useful if the authors could comment in the manuscript.

sPD Patients were not pre-selected in any way. Patients were only screened for the absence of PD causing mutations, to ensure that only material from sporadic patients was used¹⁵. This is now explicitly stated in the manuscript on page 3, lines 20-22.

In many studies analyzing the mitochondrial respiration of fibroblasts derived from sPD patients in a similar approach as reported in our manuscript, a clear mitochondrial deficit e.g. a reduced basal or maximal respiration or a reduced complex I activity has been described¹⁶. However, the largest study up to date including fibroblasts from 20 PD patients and 19 Ctrl's couldn't recapitulate these findings also due to the large variability among human individuals¹⁷. Nor could these findings be recapitulated by analyzing the fibroblasts derived from sPD patients used in this study (**Fig. 2a**).

Also, a recent study from 2020¹⁸ revealed differences in mitochondrial respiration between fibroblasts derived from healthy individuals and sPD patients, respectively. However, specifically, the latter report clearly indicates that when interpreting the results of mitochondrial respiration, i.e. cellular oxygen consumption rates the experimental set-up has to be considered. They clearly show that cell culturing conditions (i.e. medium composition) and time after medium changes are critical components for detecting mitochondrial respiration deficits (besides the well-known effect of cell passage numbers). Specifically, they showed that these mitochondrial respiration deficits are only detectable when fibroblasts are forced into OXPHOS respiration for several hours. Interestingly, when cultured for longer times under these conditions the difference is not detectable anymore, as well as that there is no difference after 1 hour in OXPHOS inducing medium (see their Suppl. Fig. 3). The latter condition is, however, exactly the condition we have chosen, to detect changes in mitochondrial respiration and not in glycolysis and to analyze the fibroblasts in a native state, without induced metabolic reprogramming. In addition, in our manuscript - due to comparability with our results in NPCs and neurons - we assessed mitochondrial functionality mainly by the analyses of mitochondrial respiration using the Seahorse analyzer. Even though this is a highly versatile analysis it does not evaluate further mitochondrial abnormalities such as changes in mitochondrial morphology, impaired mitophagy, increased mitochondrial ROS-production, changes in mitochondrial membrane potential, etc.. all of which have been implicated in one way or the other in sPD. However, in our opinion, an analysis of all these different parameters would have been beyond the scope of this manuscript, specifically since the focus of the manuscript is on the alterations in cilia morphology.

Thus, our results fit to the recent studies for mitochondrial respiration in fibroblasts derived from sPD patients. And, as we indicated in the manuscript, we hypothesize a quite distinct and cell type-specific presentation of PD phenotypes, a notion which we have now - in respect to potential therapy - included in the discussion of the manuscript on page 13, lines 19-39.

5) In Figure 4 data, the authors state the data shown is from triplicates, 5 control clones and 7 sPD clones. Is this is then in a reduced cohort of cells from the data in Figure 1 which the authors state is from 10 control clones and 14 sPD? If so, how were those clones selected, is it 1 clones per line rather than 2? For

the data in Figure 4, does each dot represent the mean of 3 biological replicates of each clone, could the authors clarify this? In addition, clarification that the 400 cells or PC referred to were across multiple biological repeats. Finally the Seahorse trace shown in Figure4k, needs the vehicle treated control conditions showing on the same plot, and the additional calculations and graphs shown for each parameter to be able to show that the cyc inhibition has had any effect on the respiration state of the cells. This is critical to the PC and SHH signalling effect being critical to make the link to the overall mitochondrial health.

The initial metabolic analysis shown in Fig. 1 (now **Fig. 2**), especially those of hiPSC, hNPCs, and DANs was carried out using 2 clones per individual from 5 Ctrl and 7 SPD patients. As the pooled, droplet-based scRNA-seq analysis wouldn't allow distinguishing between clones from the same individual, which would have the same characteristic SNP pattern, one hNPC clone per individual was selected (**Supplementary Table 12**) based on features characteristic of hiPSCs such as displaying a stable karyotype and normal pluripotency. The same hNPC clones were used for all further analysis (**Supplementary Table 12**). This is now explicitly stated in the manuscript on page 4, lines 18-21.

Every analysis was performed thrice using independently collected material from 5 Ctrl and 7 SPD patient-derived cell clones e.g. from three different passages (fibroblasts, hiPSCs) or three independent differentiation approaches (hNPCs, DANs, astrocytes). Each data point represents the mean of these three replicates for each individual. If two clones per patient were used for experiments, first the mean value of these clones per individual and replicate was calculated. This is now explicitly stated in the Methods section "Statistics and reproducibility" on page 29, lines 36-40, and page 30, lines 1-2.

Similarly, also PC were analyzed in three independently collected materials per individual. At least 30 PC per individual and replicate were analyzed. This is now explicitly stated in the figure caption of **Fig. 5, 7, 9** and the Methods section "Immunocytochemistry and imaging of human cells" on page 17, lines 32-34.

Cyclopamine data are now always presented with vehicle-treated controls (DMSO treated) in **Fig. 7**. The only exception is **Fig. 7d**, the quantification of GLI3 processing by western blots. Here the corresponding vehicle-treated control is displayed in **Fig. 6c**. This separation can be justified for semi-quantitative methods such as western blots. We explicitly stated this now in the manuscript on page 9, lines 6-20.

6) The data and associated explanation provided by the authors for the decrease and increase in PC length between cell model and post mortem is well articulated; however, as this is key to hypothesis of PC involvement in PD pathogenesis proposed here, is it possible for the authors to validate the PC measurements via an alternative approach or staining?

For visualizing the whole PC of human neuronal cells, ARL13B is probably the best and also the most widely used marker. Another regularly used cilia marker is Ac-Tub, which is however not only localized in cilia and thus often leads to false-positive identifications as shown by Lim et al., 2015¹⁹ and as indicated in **ReFig. 3a**. Double stainings of ARL13B together with either AC-TUB (**ReFig. 3a**), SMO (**ReFig. 3b**), or GLI3 (**ReFig. 3c**) also nicely indicates that ARL13B is capable of visualizing the whole cilia from the ciliary base to tip. Both SMO and AC-TUB completely colocalize with ARL13B throughout the whole cilium, whereas

the ciliary tip is indicated by enrichment of GLI3 (ReFig 3c and ²⁰). The ciliary base is highlighted by GLI3 and AC-TUB spreading originating from the basal body/mother centriole, the cellular microtubule-organizing center. Thus it can be assumed that the whole PC has been used for quantifications.

Furthermore, in the case of *PINK1* ko models, shortening of PC was validated via alternative stainings for human iPSCs (ARL13B) and mouse brain regions (Adcy3) (Fig. 9).

ReFig. 3 | Immunostaining with anti-ARL13B together with either (a) anti-AC-TUB, (b) anti-SMO, or (c) anti-GLI3 using hNPCs. Immunostainings are exemplarily shown for iG3G-R1-039.

Minor comments

Line 75 make sure Cntrl is defined

This has now been defined on page 3, lines 20-22.

Line 78 grammar, allowed "us" to

This has been corrected.

Reviewer #4 (Remarks to the Author):

In the manuscript "Primary cilia dysfunction in Parkinson's disease etiology" the authors address the question of whether molecular and cellular alterations of primary cilia in PD patient rejuvenated cell lines are a distinctive feature of early PD development.

To this end, the authors used single cell transcriptomics to comprehensively investigate the cellular composition of neural precursors differentiated from several sporadic PD lines and to determine the molecular mechanism underlying primary cilia dysfunction. They identify alterations in SHH signaling pathway as being responsible for mitochondrial deficits in the PD lines analyzed. Additionally, ciliary alterations were observed in postmortem tissue of sporadic PD patients as well as in familial PD iPSCs. Although an extremely valuable tool, the use of iPSCs to model neurodegenerative disorders such as PD is still somewhat debated, and new reprogramming methodologies are being developed with the aim of generating human cellular models that retain age and epigenome modifications of PD patients (e.g. direct reprogramming). The presented manuscript could contribute to providing novel insights into this topical issue by identifying ciliary dysfunction as an early event during differentiation that may guide our understanding of the etiology of PD. Overall, the conclusions of the manuscript are well supported by the data, and the manuscript addresses important and as yet unresolved questions in the field. The characterization of primary cilia alteration in postmortem tissue and in familial forms of PD do indeed corroborate the main objective of the manuscript. However, involvement of SHH signaling in primary cilia dysfunction in PD is not novel (Dhekne et al., eLife 2018; Bae et al., Cell Death and Disease 2019, Sobu et al., PNAS 2021). To make the manuscript suitable for publication in a top-tier journal such as Nature Communications, the authors should provide additional experiments to consolidate their findings especially regarding the use of iPSCs as a potential model to study PD pathoetiology.

We are thankful for the hint on the already existing knowledge about SHH signaling in primary cilia dysfunction in PD etiology. Still, we think that our manuscript adds further and new aspects to this field of research due to the following reasons.

1. Most of the findings of altered PC biology in PD are based on familial forms of PD, specifically Lrrk2³ associated PD and lately also a-synuclein²¹ associated PD. We, in this manuscript, have now added a third familial form of PD (*PINK1*-associated) to this list. Thus, our new finding, that also in human material derived from *PINK1*-mutation carriers PC alterations can be observed, puts PC further in the focus of research on PD etiology. Most importantly, we here show for the first time PC alterations also in sporadic PD:
2. Changes in PC morphology have not yet been reported, neither in cellular systems derived from patients nor in post-mortem brain tissue derived from sporadic PD patients, the vast majority of PD cases. Still, and this is pointed out very clearly and explicitly in the manuscript (page 12, lines 34-35), PC related molecular pathways have been identified in several transcriptome analyses of sPD human material but never addressed concerning the possible consequences on PC morphology and in signal transduction e.g. SHH-signaling, nor their contribution to PD etiology.

Mostly this finding has been left uncommented^{2,22}. Thus, our validation of the consequences of altered PC biology concerning SHH-signaling in human material of sPD patients is a new and not yet reported finding in the field. Together, with the emerging knowledge of the involvement of PC biology in different forms of fPD (both dominant and recessive forms) - also shown in this manuscript (see point 1) - we think, that we add sufficient new aspects to research on PD etiology deserved to be published in Nature Communication.

3. We also think that our finding that SHH-signaling alterations - induced or as a consequence of mitochondrial impairments and ciliary dysfunction - is a novel finding in the field. Already published manuscripts about the effect of fPD associated mutations on PC function haven't reported an influence on SHH-signaling at all - in the case of α -synuclein mutations - or have even reported an opposing effect - in the case of Lrrk2 mutations. Furthermore, even though an influence of mitochondrial functionality on PC morphology is known from Bae et al. and that SHH transmits neuroprotection in a toxin-induced mouse model of PD, our findings are new since they are based on a human cellular sPD model derived from patients and represent as worked out in the manuscript a model of early PD. The genetic mouse model (MPTP-induced degeneration of DA neurons) is rather representing the late stage of PD and fits to our reported study of PC morphology in sPD patients. Considering that SHH-signaling is highly cell type-specific - as known from the cancer field - it is not surprising that its effect in different stages and on different cell types might differ, e.g. in early sPD an overactivation can be detrimental to mitochondrial health, whereas in late sPD inducing SHH-signaling might indeed protect from cell death. Concerning the observation in the second paper from Bae et al., 2019²³ - that inhibition of mitochondrial activity induces cilia elongation in a immortalized cell line (SH-5Y5 cells) - is also not rebutting our findings. Contrary, it only demonstrates again that different effects on PC in conjunction with mitochondrial deficits can depend on 1) the severity but also more importantly on 2) the underlying causes of these deficits. Bae et al. showed that strong deficits in mitochondrial function due to the complete inhibition of the mitochondrial electron transport chain can cause PC elongation, whereas in this study we report a shortening of PC due to a ciliary mediated alteration in SHH signaling which also resulted in a slightly reduced mitochondrial respiration. These molecular differences are in our view highly relevant in the PD field, specifically when thinking about using SHH-signaling modulations as therapeutic interventions. Activation and inhibition of this pathway have quite distinct consequences when applied during the course of the disease, a notion which we have now stated clearly also at the end of the manuscript (page 14, lines 1-4). Thus, again, we are convinced that our findings are of the highest importance to the field and add a new view on the role of PC in the etiology of sPD.

In sum, the data presented in this manuscript are novel as they show for the first time ciliary alterations in sPD at the cellular level and clearly implicate SHH-signaling in this process. Furthermore, and most importantly, these data for the first time clearly indicate that molecular and cellular pathomechanisms might differ between early and late stages of PD, even concerning the same pathway. This finding is of the highest importance for addressing, finding and timing the application of new potential disease halting

if not preventing therapies, which are urgently needed in the field of neurodegeneration and thus are of highest importance for a broad readership.

Major comments:

1) In contrast to older PAX6⁺-inducing protocols, it has now been shown that for correct patterning towards authentic midbrain dopaminergic neuron generation, hPSCs have to pass through an intermediary floor plate progenitor state involving FOXA2⁺/LMX1⁺/PAX6⁻ (Fasano et al., Cell Stem Cell 2010; Kriks et al., Nature 2011; Kirkeby et al., Cell Rep 2012; Nolbrant et al., Nat Prot 2017). The expression of PAX6 in NPC culture suggests that the authors have not used a floor plate transition protocol. They should at least clarify the choice of their differentiation protocol in the text.

We have used Reinhard's protocol²⁴, in which indeed the cells do not go through an intermediary floor plate progenitor state, due to "evolutionary" reasons of the project. The hiPSCs were generated and distributed by the ForIPS Consortium (Univ. Erlangen). For reasons of comparability of results obtained with these hiPSC cells and their derivatives, it was recommended to use similar differentiation protocols. Thus, we started this project in 2014 using this protocol. We did not switch during the course of the ongoing project because specifically mitochondrial function is heavily dependent on culturing conditions (own unpublished results) and switching to another differentiation protocol during the course of the project would have required repeating all experiments from the start. In addition, since the focus of our analysis were NPCs and to establish an early sPD model - independent of e.g. neuronal circuitries - and therein identify molecular signatures we judged at this time the usage of this differentiation protocol as justified even though not optimal for differentiation of dopaminergic neurons. A reference for the choice of this protocol is now added in the text on page 15, lines 16-19.

2) An in-depth analysis of PD and ctrl lines should be carried out when they are still cultured under pluripotent conditions. Mitochondrial stress testing should be performed together with either bulk- or sc-RNAseq to provide a comprehensive characterization at transcriptional, metabolic, and functional resolution of undifferentiated PD cell lines adopted in this study.

We now also included an analysis of mitochondrial function and PC morphology using the undifferentiated hiPSCs. This is now shown in **Fig. 2b and 5a**, respectively. hiPSCs derived from sPD patients did not exhibit alterations in mitochondrial function nor in PC morphology. These sPD specific alterations only occurred after the differentiation of these hiPSCs into hNPCs and further into DANs (**Fig. 2c,d and Fig. 5b,c**). As we couldn't observe sPD specific alterations in the cellular processes relevant for this paper on the hiPSC level, we did not proceed with performing a transcriptome analysis.

3) sc-RNAseq analysis is generally well carried out, but further details on the methods used should be reported (a) and the following new analyses should be performed (b-e):

- a) Specifically, the authors should clarify and specify for both sPD and fPD cell lines analyzed:
 - how the data looked before integration

A distinct batch-specific structure could be seen before MNN data integration in both the donor cell line data (**ReFig. 4**) and the knockout cell line data (**ReFig. 5**). Specifically, the knockout cell line data was quite distinct before integration, with only a few cells overlapping in the small immature neuron population. To further analyze common sources of variation we thus proceeded with data integration using MNN. This integration method was recently found to preserve nuanced biological variation on the gene level better than other approaches²⁵.

ReFig. 4 | UMAP plot of the SPD cell line data before MNN integration showing distinct donor structure in the embedding.

ReFig. 5 | UMAP plot of the fPD (*PINK1* ko) cell line data before MNN integration.

- what cutoffs were used to define DE genes and what the rationale was

Differential expression was determined by a cutoff of an adjusted p-value threshold of 0.05, conforming to control for type I error that is standard practice in the field. P-values were adjusted via a Benjamini-Hochberg correction to account for the family-wise error rate in multiple testing. This is stated in the Methods chapter "Differential expression analysis" on page 25, lines 15-32.

- what resolution was chosen to define cell clusters

As mentioned in the Methods sub-section titled "Clustering, sub-clustering, marker gene detection, and cluster annotation", we clustered the data by harmonizing clustering efforts on the data with cell-cycle effects (initial resolution of 0.2, subclustering at a resolution of 0.2) and with cell-cycle effects regressed out (initial resolution of 0.2, with sub-clustering at resolutions of 0.3 and 0.1). To give further details on this process, we have now expanded the methods sub-section mentioned above. We also invite the reviewer to explore our data analysis notebooks on the accompanying github repository at github.com/theislab/ipsc_ipd_analysis.

- number of single cells analyzed.

The number of single cells analyzed in the patient cell lines was 30,626 cells. We have added this number to the Methods section titled "Quality control and pre-processing" on page 22, lines 15-16. We have otherwise included the number of single cells that pass QC in the relevant sections of the paper, such as the figure caption of e.g. **Fig 3**.

Additionally, was the harmony integration performed based on cell lines used? Did the authors detect any cluster(s) that might indicate cell doublets?

We used mutual nearest neighbor integration via the MNNpy algorithm in this study using donor of origin as a batch covariate. We have updated the Methods subsection titled "Quality control and preprocessing" to explicitly state this. We did not integrate the fPD (*PINK1* ko) cell lines with the sPD cell lines as we lack a control to assess whether such integration would induce false signals as the *PINK1* ko cell lines were reprogrammed using a different method. Thus, we focused mainly on the sPD cell line data in our manuscript.

We did not identify any cluster in the analysis that indicates cell doublets, as doublet filtering was performed prior to integration via demuxlet. As we pooled cells from all donors and demultiplexed them afterwards, we could identify doublets based on overlapping genetic barcodes. This approach can identify all doublets except those from the same donor, which amounts to 91.67% of doublets in our study. Thus, residual doublets were negligible and were not detected in either the sPD cell lines nor the knockout models. We have now further clarified this doublet filtering approach in the Methods section on page 22, lines 8-9, reading:

“As demuxlet can both assign cells to the donor they came from and identify doublets, we used this algorithm to also filter out doublets. Thus, only cells called as singlets by demuxlet were used for further processing.”

b) In Fig. 2b, a barplot showing the percentage of each cluster in each donor line would be useful to interpret reproducibility. This should be accompanied by statistical testing of difference in cluster proportion in PD/C.

We thank the reviewer for these helpful suggestions to show the robustness of the replicates. We have now added a stacked barplot per sample showing the consistency of cell-type compositions across donors (**Fig. 3d**). To quantify these observations, we used scCODA¹, which uses a Dirichlet multinomial model to fit the observed counts per cell type per donor, to test for significant changes in compositions in PD patients vs Ctrl samples (details added to the Methods section “Compositional data analysis” on page 23, lines 38-40 and page 24, lines 1-5). As expected, no significant differences were detected (**Supplementary Fig. 1e**), which is now also stated in the manuscript on page 5, lines 22-23.

c) What happened to the NCSC and glial clusters in the velocity analysis?

When performing RNA velocity analysis it is important to consider lineages separately as these may reflect different splicing kinetics leading to false interpretation, especially of non-dominant lineages²⁶. In the time point at which we are analyzing the cells, the NCSC, NSC, and glial lineages have diverged. Thus, we focused only on the NSC cluster for this analysis, which is the dominant lineage and the focus of the manuscript. This is detailed specifically in the Methods section titled “RNA velocity and trajectory analysis” on page 24, lines 6-33.

d) The DEG analysis (Fig. 2) would benefit from a more stringent presentation. It needs to be clear which comparisons have been made and if velocity or regular clusters were used. 13,132 genes are detected as DE, which seems like an extremely high number considering that 4–5000 genes are typically detected per cell. Is the analysis correctly controlling for false discoveries? FC cutoffs are stated as “ $0.83 > FC > 1.2$ ”, which I think would be more clearly written as $|FC| > 20\%$.

In order to make these chapters and the corresponding figures more stringent and to increase clarity, we restructured these elements.

Furthermore, as the reviewer suggested, we have now included a more specific description of the tests that were performed to obtain 13,132 differentially expressed genes. On page 5, lines 29-31, we specifically state that these genes were found to be differentially expressed across any of the 9 cell type clusters and the velocity root population. As distinct sets of genes were expressed in individual clusters due to the different developmental stages/lineage fates e.g. NSC1a vs. NCSC, a dysregulation of these different gene results sum up to this high number of total DEGs. We note

that this combined DEG list was just reported for the sake of completeness, but was not used for any downstream analysis.

Detecting a large number of genes to be differentially expressed is not unexpected in scRNA-seq data as one is comparing expression distributions across large groups of cells. Across these groups of cells many more genes are expressed than in any individual cell. For example, 23,664 genes are expressed in at least 50 cells in the NSC1a cluster. Furthermore, as we are testing for differential expression in each of 11 different clusters, it is expected to get a larger number of DEGs than expressed per cell on average. As mentioned in the methods section titled "Differential expression analysis", we corrected for multiple testing using the Benjamini-Hochberg procedure within every cell type cluster, which is common practice in computational biology and indeed comparatively conservative²⁷. Furthermore, the analysis code is now freely available at www.github.com/theislab/ipsc_sPD_analysis.

Finally, as the reviewer suggested, we have clarified the description of our fold change cutoffs in the text to read " $|FC| > 20\%$ ".

e) The authors should also provide a quantification panel (e.g. dotplot or heat map or violin plot) of mitochondrial and PD-associated gene expression between PD and ctrl NPC cultures.

A heatmap summarizing fold changes of PD- or mitochondria-associated genes per cluster is now included in the manuscript as **Supplementary Fig. 5**.

4) The comparison with postmortem tissue (Fig. 5) is important and adds relevance to the data presented in this manuscript. A more direct comparison of genes found in postmortem vs author derived dataset would be prudent. What proportion of genes overlap? Are there any themes among genes that do not agree between datasets? Single cell results from NPC PD lines should also be analyzed in direct comparison to the single cell dataset generated from substantia nigra of human post-mortem brain tissue recently published by Agarwal et al, Nat Comm 2020.

The postmortem transcriptome analyses were performed using substantia nigra pars compacta tissue at a very late stage of disease progression, with most dopaminergic neurons already being degraded. Thus, alterations in gene expression are probably also resulting from that imbalanced comparison (having neurons vs. not). This may not directly reflect the changes observed in our hNPCs which display a very early stage of 1) neuron differentiation but also 2) PD, as our cultures don't show any sign of neurodegeneration yet. Especially the different states of differentiation (precursors vs. adult brain tissue) may cause different and distinct gene expression patterns. Nevertheless, despite these large differences between both datasets, similar patterns could be observed as stated in the manuscript: 1) Overrepresentation of downregulated genes and 2) Significantly enriched SHH target genes within DEGs. Additionally, around 60% of the DEGs reported in the postmortem analysis were also significantly dysregulated in our hNPC analysis with around 50% of the downregulated DEGs being also downregulated in our analysis (**ReFig. 6**). Whereas for the upregulated DEGs, around 50% of the genes were now downregulated in our analysis. This may highlight the relevance of specifically gene repression for sPD pathology, as it is more consistent between datasets than gene activation.

Revision of manuscript NCOMMS-21-30277 - Primary cilia dysfunction in Parkinson's disease etiology
Response to reviewer's comments

ReFig. 6 | Overlap of DEGs between the transcriptome analysis published by Wang et al., Nature Communications 2019 and the analysis described in this manuscript. Values indicate the percentage composition of DEGs identified by Wang et al. in our bulk-like DEGs.

Although the meta-analysis published by Wang et al., Nature Communications. 2019 is not based on scRNA seq data, it is up to date one of the largest postmortem transcriptome datasets for PD, comparing the gene expression between 83 PD patients and 70 controls. In contrast, the dataset published by Agarwal et al., Nature Communications. 2020 contains substantia nigra scRNA seq results, however, only from healthy donors. As we were analyzing neuronal precursor cells for this manuscript, a direct comparison of precursor clusters to clusters from postmortem neurons might not be expedient. These cell populations are not necessarily directly comparable. Instead, we focused on finding common patterns between our transcriptome analysis and published analysis using postmortem midbrain tissue of also sPD patients.

5) The manuscript might benefit from the addition of sc-RNAseq analysis of mature postmitotic DA neurons differentiated from PD lines vs ctrls. This could reconstruct the trajectory of the disease, yielding a greater understanding of the early phases of PD progression.

We do agree with the reviewer that a scRNA-seq of mature postmitotic DANs would be an interesting experiment. Thus, we are currently performing these experiments. However, finishing these experiments within a meaningful timeframe is out of the scope of this manuscript. Thus, we plan to publish them in a follow up manuscript.

6) The authors should clarify the cellular composition and molecular signature of fPD lines and provide a direct comparison to sPD NPCs highlighting points of similarity and differences as they do at protein level.

Given differences in data size, quality, reprogramming, and differentiation protocol, a detailed comparison between sPD and fPD lines is challenging. As the reviewer suggests, we show below the cell type composition of the two fPD line samples (*PINK1* ko and isogenic Ctrl), which are highly similar (**ReFig. 7**). Due to limitations in data size and quality, we were not able to uncover substructures beyond NSC1, immature neurons, and apoptotic NSCs in the fPD lines. Specifically, marker genes for NSC2 cells (*FOXA2*, *SHH*) were not expressed by more than individual outlier cells in the fPD data (**ReFig. 8**). However, NSC1 and immature neuron markers are specifically expressed consistently across both sPD and fPD lines (**ReFig. 8 and Supplementary Fig. 2**) suggesting these are consistent populations.

Revision of manuscript NCOMMS-21-30277 - Primary cilia dysfunction in Parkinson's disease etiology
Response to reviewer's comments

Furthermore, also at the DEG level of fPD lines similarities and differences to the sPD lines can be reported. Similarly, also in *PINK1* ko hNPCs genes associated with the intraflagellar transport machinery were significantly dysregulated. Thus, ciliary anterograde and retrograde transport processes seemed to be affected both in sPD and fPD lines (**Fig. 4c and Fig. 9c**). Contrary, in *PINK1* ko hNPCs, downregulated genes seemed to be not overrepresented (**Fig. 4b and Fig. 9b**).

This suggests again, that although displaying distinct responses to the PD state, sPD and fPD lines share also similarities, especially regarding PC dysfunction, which might be an interesting convergence point in PD etiology.

ReFig. 7 | Stacked bar plot of cell type compositions per sample in fPD cell lines. The y-axis indicates the fraction of cells from a sample (N=2529 *PINK1* KO cells, N=1142 Isogenic Ctrl cells).

ReFig. 8 | UMAP plots of fPD line data colored by marker genes for NSC2 (*FOXA2*, *SHH*) and immature neurons (*DCX*, *MAP2*). N=3671 cells.

Minor comments:

1) Did the identified CNVs (Fig. 1b/c) fall within any interesting PD-related loci?

Genes that fall into genomic regions affected by CNVs were investigated and analyzed using pathway enrichment analysis (Pathway Studio). We could show that no pathway or genes were affected by CNVs that were of interest for this study.

We explicitly stated this now in the manuscript on page 3, lines 27-30.

2) Bae et al., Cell Death and Disease 2019 and Sobu et al., PNAS 2021 should be reported and commented in the Discussion section.

Bae et al., Cell Death and Disease 2019 and Sobu et al., PNAS 2021 are now discussed in the manuscript on page 12, lines 22-23, and on page 12, line 30, respectively.

3) Is supplementary Fig. 6a a cytofluorimetric analysis? If so, please provide representative FACS plots for both ctrl and sPD.

Cell proliferation rates were analyzed using a high content imager; an exemplary picture taken by the HCI to quantify the nuclear fluorescence intensity is now displayed in **Fig. 7a**.

4) Provide primary cilia immunostaining with anti-Adcy3 and anti-Rbfox3 also for midbrain dopamine neurons as in Fig. 5i.

Immunostainings and the quantification of PC length of neurons within the substantia nigra pars compacta are now included in the manuscript in **Fig. 9j**.

5) NSC2 population is missing in the UMAP plot (Fig. 6a) from PINK1 ko and isogenic ctrl hNPC cell line. High resolution clustering analysis could be run.

As mentioned in the answer to point 6) above, NSC2 markers were only expressed by individual outlier cells in the fPD line data (**ReFig. 8**). Subclustering of the NSC1 cluster did not produce meaningful substructures in this cluster. Thus, we conclude that NSC2 cells are either absent or only present in very small quantities which do not separate out from the NSC1 cluster. Subclustering attempts can be seen in the code at https://github.com/theislab/ipsc_ipd_analysis/blob/master/notebooks/11_iPSC_iPD_ko_cell_line_analysis.ipynb.

7) Line 136 page 4 should be FOXA2

This has been corrected.

References

1. Büttner, M., Ostner, J., Müller, C. L., Theis, F. J. & Schubert, B. scCODA is a Bayesian model for compositional single-cell data analysis. *Nature communications* **12**, 6876; 10.1038/s41467-021-27150-6 (2021).
2. Donega, V. *et al.* Transcriptome and proteome profiling of neural stem cells from the human subventricular zone in Parkinson's disease. *acta neuropathol commun* **7**, 84; 10.1186/s40478-019-0736-0 (2019).
3. Dhekne, H. S. *et al.* A pathway for Parkinson's Disease LRRK2 kinase to block primary cilia and Sonic hedgehog signaling in the brain. *eLife* **7**; 10.7554/eLife.40202 (2018).
4. Belgacem, Y. H., Hamilton, A. M., Shim, S., Spencer, K. A. & Borodinsky, L. N. The Many Hats of Sonic Hedgehog Signaling in Nervous System Development and Disease. *Journal of developmental biology* **4**; 10.3390/jdb4040035 (2016).
5. Machold, R. *et al.* Sonic Hedgehog Is Required for Progenitor Cell Maintenance in Telencephalic Stem Cell Niches. *Neuron* **39**, 937–950; 10.1016/S0896-6273(03)00561-0 (2003).
6. Tail, M. *et al.* The Sonic Hedgehog Pathway Modulates Survival, Proliferation, and Differentiation of Neural Progenitor Cells under Inflammatory Stress In Vitro. *Cells* **11**; 10.3390/cells11040736 (2022).
7. Garrett, L. *et al.* Conditional Reduction of Adult Born Doublecortin-Positive Neurons Reversibly Impairs Selective Behaviors. *Frontiers in behavioral neuroscience* **9**, 302; 10.3389/fnbeh.2015.00302 (2015).
8. Wang, J. *et al.* Disruption of Sonic Hedgehog Signaling Accelerates Age-Related Neurogenesis Decline and Abolishes Stroke-Induced Neurogenesis and Leads to Increased Anxiety Behavior in Stroke Mice. *Translational stroke research*; 10.1007/s12975-022-00994-w (2022).
9. Bicker, F., Nardi, L., Maier, J., Vasic, V. & Schmeisser, M. J. Criss-crossing autism spectrum disorder and adult neurogenesis. *Journal of neurochemistry* **159**, 452–478; 10.1111/jnc.15501 (2021).
10. Inta, D., Lang, U. E., Borgwardt, S., Meyer-Lindenberg, A. & Gass, P. Adult neurogenesis in the human striatum: possible implications for psychiatric disorders. *Molecular psychiatry* **21**, 446–447; 10.1038/mp.2016.8 (2016).
11. Gatto, L. *et al.* Molecular Targeted Therapies: Time for a Paradigm Shift in Medulloblastoma Treatment? *Cancers* **14**; 10.3390/cancers14020333 (2022).
12. Dulken, B. W., Leeman, D. S., Boutet, S. C., Hebestreit, K. & Brunet, A. Single-Cell Transcriptomic Analysis Defines Heterogeneity and Transcriptional Dynamics in the Adult Neural Stem Cell Lineage. *Cell reports* **18**, 777–790; 10.1016/j.celrep.2016.12.060 (2017).
13. Adusumilli, V. S. *et al.* ROS Dynamics Delineate Functional States of Hippocampal Neural Stem Cells and Link to Their Activity-Dependent Exit from Quiescence. *Cell stem cell* **28**, 300-314.e6; 10.1016/j.stem.2020.10.019 (2021).
14. Bantle, C. M., Hirst, W. D., Weihofen, A. & Shlevkov, E. Mitochondrial Dysfunction in Astrocytes: A Role in Parkinson's Disease? *Frontiers in cell and developmental biology* **8**, 608026; 10.3389/fcell.2020.608026 (2020).
15. Popp, B. *et al.* Need for high-resolution Genetic Analysis in iPSC: Results and Lessons from the ForIPS Consortium. *Scientific reports* **8**, 17201; 10.1038/s41598-018-35506-0 (2018).
16. Ambrosi, G. *et al.* Bioenergetic and proteolytic defects in fibroblasts from patients with sporadic Parkinson's disease. *Biochimica et biophysica acta* **1842**, 1385–1394; 10.1016/j.bbadis.2014.05.008 (2014).
17. del Hoyo, P. *et al.* Oxidative stress in skin fibroblasts cultures from patients with Parkinson's disease. *BMC neurology* **10**, 95; 10.1186/1471-2377-10-95 (2010).

Revision of manuscript NCOMMS-21-30277 - Primary cilia dysfunction in Parkinson's disease etiology
Response to reviewer's comments

18. Deus, C. M. *et al.* Mitochondrial remodeling in human skin fibroblasts from sporadic male Parkinson's disease patients uncovers metabolic and mitochondrial bioenergetic defects. *Biochimica et biophysica acta. Molecular basis of disease* **1866**, 165615; 10.1016/j.bbadis.2019.165615 (2020).
19. Lim, Y. C., McGlashan, S. R., Cooling, M. T. & Long, D. S. Culture and detection of primary cilia in endothelial cell models. *Cilia* **4**, 11; 10.1186/s13630-015-0020-2 (2015).
20. Lex, R. K. *et al.* GLI transcriptional repression is inert prior to Hedgehog pathway activation. *Nature communications* **13**, 808; 10.1038/s41467-022-28485-4 (2022).
21. Iqbal, A., Baldrighi, M., Murdoch, J. N., Fleming, A. & Wilkinson, C. J. Alpha-synuclein aggregates inhibit ciliogenesis and multiple functions of the centrosome. *Biology open* **9**; 10.1242/bio.054338 (2020).
22. Stern, S. *et al.* Reduced synaptic activity and dysregulated extracellular matrix pathways are common phenotypes in midbrain neurons derived from sporadic and mutation-associated Parkinson's disease patients (2022).
23. Bae, J.-E. *et al.* Primary cilia mediate mitochondrial stress responses to promote dopamine neuron survival in a Parkinson's disease model. *Cell death & disease* **10**, 952; 10.1038/s41419-019-2184-y (2019).
24. Reinhardt, P. *et al.* Derivation and expansion using only small molecules of human neural progenitors for neurodegenerative disease modeling. *PLoS one* **8**, e59252; 10.1371/journal.pone.0059252 (2013).
25. Luecken, M. D. *et al.* Benchmarking atlas-level data integration in single-cell genomics. *Nature methods* **19**, 41–50; 10.1038/s41592-021-01336-8 (2022).
26. Bergen, V., Soldatov, R. A., Kharchenko, P. V. & Theis, F. J. RNA velocity-current challenges and future perspectives. *Molecular systems biology* **17**, e10282; 10.15252/msb.202110282 (2021).
27. Korthauer, K. *et al.* A practical guide to methods controlling false discoveries in computational biology. *Genome biology* **20**, 118; 10.1186/s13059-019-1716-1 (2019).

REVIEWERS' COMMENTS

Reviewer #1 (Remarks to the Author):

Authors have successfully addressed all my comments. I found the single cell genomics analyses much easier to follow and through. I congratulate them for this exciting and interesting study about the role primary cilia and its interplay with the Sonic Hedgehog signalling in the pathophysiologic of PD.

Reviewer #2 (Remarks to the Author):

The authors have successfully addressed my concerns through additional experiments and further discussion. I have only one minor issue related to wording.

Page 11, line 5 and line 15: It is not appropriate to call human patient derived cells "ko", as they contain naturally occurring mutations, not artificially knocked-out versions. It is better to stick with the term "deficient".

Reviewer #3 (Remarks to the Author):

The authors have addressed most of the comments from the reviewers well and this has added significantly to the manuscript and interpretation of the data within. The additional data and interpretation of the data from across multiple models clearly indicate cell type specific and disease stage specific changes in PC in PD which could be related to mitochondrial function. Some small points of additional clarification are still needed.

With regard the number of clones used for each experiment from each line. The authors state in the manuscript now (and response to reviewers) that if more than 1 clone was used, the mean was taken first before putting the data into the control vs patient analysis. It is important for the authors to state how many clones were used for which experiments for each line, in particular between the controls and the patients. In addition, if more than 1 clone was used for some lines and for other lines only 1 clone was used, could the authors please include in the methods section why the clones were selected in this way.

In the response to reviewers the authors cite the largest study in Parkinson's patient fibroblasts to date and indicate their data are in line with this, however the authors do not include 2 papers published in 2019 and 2020 respectively, Milanese et al, 2019 and Carling et al, 2020 which include significantly larger cohorts of sPD lines and controls. The authors should include a comment in the discussion of the article how their findings of cell type specific dysfunction in mitochondria (and PC which they have focused on) fit into the literature on phenotypes observed in larger sPD patient cohorts.

Reviewer #4 (Remarks to the Author):

I enjoyed reading the revised version of this manuscript. All concerns raised have been adequately addressed.